# Predicting resistance to chemotherapy using chromosomal instability signatures

Joe Sneath Thompson[1,2,10], Laura Madrid [2,10], Barbara Hernando [1,10], Carolin M. Sauer [3], Maria Vias[3], Maria Escobar-Rey[1,2], Wing-Kit Leung[2,3], Diego Garcia-Lopez [2], Jamie Huckstep [3], Magdalena Sekowska[3], Karen Hosking[4,5], Mercedes Jimenez-Linan[5,6], Marika A. V. Reinius [3,5,6], Abhipsa Roy[2], Omar Abdulle[2], Justina Pangonyte [3], Harry Dobson [2], Amy E. Cullen [2,3], Dilrini De Silva [2], David Gómez-Sánchez[1,7], Marina Torres [1], Ángel Fernández-Sanromán[1], Deborah Sanders[3], Filipe Correia Martins[3,5,6,8], Ionut-Gabriel Funingana[3,4,5], Giovanni Codacci-Pisanelli[3,4,9], Miguel Quintela-Fandino [1], Florian Markowetz [2,3,4], Jason Yip[2], James D. Brenton [2,3,4,5,6], Anna M. Piskorz [2,3,11] ✉ & Geoff Macintyre [1,2,11] ✉

Chemotherapies are often given without precision biomarkers, exposing patients to toxic side effects without guaranteed benefit. Here we present chromosomal instability signature biomarkers that identify resistance to platinum-, taxane- and anthracycline-based treatments using a single genomic test. In retrospectively emulated randomized-control biomarker clinical trials using real-world cohorts ($n = 840$), predicted resistant patients had elevated treatment failure risk for taxane (hazard ratio (HR) of 7.44) and anthracycline (HR of 1.88) in ovarian, taxane (HR of 3.98) and anthracycline (HR of 3.69) in metastatic breast and taxane (HR of 5.46) in metastatic prostate. Nonrandomized emulations showed predictive capacity for platinum resistance in ovarian (HR of 1.46) and anthracycline in sarcoma (HR of 3.59). We demonstrate feasibility using whole-genome sequencing, capture-panel sequencing and cell-free DNA. Our findings highlight the clinical value of chromosomal instability signatures in predicting resistance to chemotherapies across multiple cancer types, with the potential to transform the one-size-fits-all chemotherapy approach into precise, tailored treatment.

Cytotoxic chemotherapies exploit the defective properties of a cancer cell, such as impaired DNA repair mechanisms, to preferentially drive cancer cells to programmed cell death[1]. Chemotherapies also have detrimental effects on healthy cells, potentially causing severe side effects despite administration alongside modern-day supportive care[2]. Many of these agents were approved for clinical use before the adoption of therapy selection biomarkers, which is in contrast to new targeted therapies that increasingly require the presence of companion diagnostic tests to guide treatment selection[3]. Identifying inherent resistance to

these agents could allow patients to avoid unnecessary side effects and receive an alternative therapy, ultimately improving overall health outcomes. Furthermore, precision use of cytotoxics could reduce healthcare costs by lowering expenditure on ineffective cancer therapies and additional medical interventions for treatment-related complications.

Before adoption in the clinic, therapy selection biomarkers must undergo clinical performance and utility evaluation[4]. Ideally, these evaluations are done using a prospective randomized-control biomarker trial. In these cases, treatment is randomized across each of

the biomarker positive and negative groups and the performance is evaluated within each group[5]. This 'phase 3 biomarker trial' design, while determining both the predictive performance and clinical utility of the biomarker, usually requires a large (and sometimes impractical) number of patients. As such, alternative designs with smaller patient numbers can be used, which trade off the capacity to assess either predictive performance or clinical utility. For instance, enrichment designs only randomize across one biomarker arm, eliminating the ability to assess utility in the other arm[6]. Single treatment arm or 'phase 2 biomarker trials', where all patients receive the experimental treatment and differences in outcome are compared between biomarker positive and negative groups, requires the smallest number of patients, but does not truly determine whether the biomarker is predictive or prognostic[7].

Considering the often prohibitive financial costs and burden of patient suffering associated with running a randomized-control trial (RCT), there has been a shift toward retrospective analysis of existing trial data for biomarker evaluations[8]. For new therapies, real-world data[9] have also been used to emulate trials[10] to seek approvals[11,12]. As chemotherapies are ubiquitous in cancer treatment, there is an exciting opportunity to combine existing methodologies of real-world emulation with accepted biomarker trial designs to emulate phase 2 and phase 3 biomarker trials.

We have recently developed a class of biomarker, chromosomal instability (CIN) signatures, that has the potential to predict therapy response[13,14]. As the full spectrum of signatures can be quantified in a tumor using a single genomic test, we hypothesize that CIN signatures might be used to predict resistance to multiple chemotherapies at diagnosis. Here, we present three CIN signature-based biomarkers for predicting resistance to treatment with platinum-based chemotherapies, taxanes and anthracyclines. We use real-world cohorts totaling 840 patients to emulate phase 2 and phase 3 biomarker trials for clinical performance evaluation. Finally, we show the feasibility of calling these biomarkers on targeted-capture gene panel sequencing of tumor tissue and shallow whole-genome sequencing (sWGS) of plasma.

## Results

### CIN signatures as biomarkers of chemotherapy resistance

The presence of CIN in a tumor has long been recognized to contribute to chemotherapy sensitivity[15]. The opposite also holds true, where tumors with stabilizing genomes tend to be resistant[16]. In a recent proof-of-principle study, we showed that CIN signatures can further refine treatment response prediction, identifying patients resistant to platinum-based chemotherapies[13]. Here, we aimed to reformulate and extend these biomarkers as predictors of chemotherapy resistance across multiple tumor types.

First, we constructed a biomarker of resistance to platinum treatment. Initially, we apply our CIN signature framework to determine tumors with and without detectable CIN. Those tumors without CIN are classified as resistant. For tumors with CIN, similar to our original study[13], we use a ratio of two signatures of impaired homologous recombination (IHR), where CX2 > CX3 indicates resistance (Fig. 1a). This is based on observations that CX2 represents a type of IHR that does not confer sensitivity to platinum-based chemotherapies, whereas CX3 does[13]. This sensitivity is based on a principle of synthetic lethality, where cells with CX3-based IHR cannot tolerate the damage introduced by platinum treatment. A critical step to enable application of this classifier to multiple tumor types is the computation of a robust scaling model for the signatures. Here, we compute a multitumor-type scaling using all *BRCA1* and *BRCA2* mutant cases present in the The Cancer Genome Atlas (TCGA) cohort (Extended Data Fig. 1a). This scaling shows improved overall survival prediction for patients treated with platinum-based chemotherapies when applied to an esophageal cohort from the original study[13] (Fig. 1b).

Next, we built a biomarker of taxane resistance using IHR signature CX5, which was previously shown to be correlated with paclitaxel

response in vitro[13], again under the principle of synthetic lethality (Supplementary Table 1). As with the platinum biomarker, samples without CIN are considered resistant. For samples with CIN, we sought an optimal threshold on CX5 signature activity to classify a sample as resistant (Fig. 1c). To do this, we used 287 cell lines treated with paclitaxel and explored a range of signature values to find the optimal activity threshold to separate cells based on the area under the dose response curve (AUC) density (Fig. 1d and Supplementary Fig. 1). To ensure multitumor-type applicability, we scaled signature activities across the whole TCGA, resulting in an optimal threshold of $z$ score-scaled signature activity of CX5 < 0 to classify a tumor sample as resistant (Fig. 1d and Extended Data Fig. 1b).

Finally, as anthracyclines are widely used across multiple tumor types, we sought a biomarker of resistance to anthracycline treatment. Like many other genotoxic chemotherapies, anthracyclines can cause DNA damage resulting in extrachromosomal DNA encapsulated in micronuclei[17]. When micronuclei rupture and release their contents into the cytoplasm, this can trigger the activation of cGAS–STING signaling, resulting in proinflammatory signaling through type I interferon[18]. It has also been established that such immune system activation is crucial for the success of anthracycline treatments[19]. However, how tumors resist anthracycline treatment is less well known. Tumors exposed to chronic cGAS–STING activation have been shown to undergo a switch to noncanonical NF-κB signaling, ultimately promoting metastasis and immune evasion[20]. Therefore, it is possible that tumors resistant to anthracyclines may tolerate the ongoing formation of micronuclei via this switching mechanism. This switching mechanism is seen as an important bottleneck during tumor evolution[18] and may represent a vital distinction between anthracycline sensitive and resistant tumors. As the amplified DNA commonly found in micronuclei can be incorporated back into the genome as homogeneously staining regions[21], it may be possible to identify tumors that have survived this evolutionary bottleneck from their genomes. CIN signatures CX8, CX9 and CX13 represent focal amplifications linked to extrachromosomal DNA[13,14]. We therefore hypothesized they could be used to identify micronuclei-tolerant, and thus anthracycline-resistant, tumors. Indeed, we found the presence of these signatures in ovarian cell lines to be associated with reduced micronuclei formation capacity, suggesting the presence of tolerance mechanisms (Supplementary Note 1).

Therefore, we sought to construct a predictor where the presence of any of these three signatures would indicate resistance to anthracycline treatment (Fig. 1e). To estimate optimal thresholds for these signatures, we used a cohort of 8 patient-derived ovarian cancer organoids and 15 primary tumor spheroids isolated from ascites and treated with the anthracycline doxorubicin in vitro (Fig. 1f, Extended Data Fig. 1c and Supplementary Table 2). Signature activity was computed from sWGS of the organoids and spheroids before treatment and treatment response was estimated using the half-maximum inhibitory concentration ($IC_{50}$). Samples were ranked by $IC_{50}$ and labeled as sensitive or resistant based on the expected number of sensitive samples given the observed sensitivity to first-line platinum treatment in the donor patients (platinum-resistant patients are expected to have an 18% response rate to doxorubicin monotherapy[22–28], whereas sensitive patients have a 28% response rate[29]). A threshold of 0.01 for CX8, and 0.009 for CX9 and CX13 showed optimal classification of the patient-derived models (Fig. 1f and Supplementary Note 1).

### Pilot study in ovarian cancer

To test the performance of these resistance biomarkers in a clinic-like setting, we wanted to see whether we could use real-world data. To determine the feasibility of this strategy, we assembled a cohort of patients with high-grade serous ovarian cancer (HGSOC). HGSOC is an ideal test bed as all three chemotherapies (platinum based, taxanes and anthracyclines) are routinely used to treat patients. Here, we aimed to emulate three different phase 2 single-arm biomarker trials using

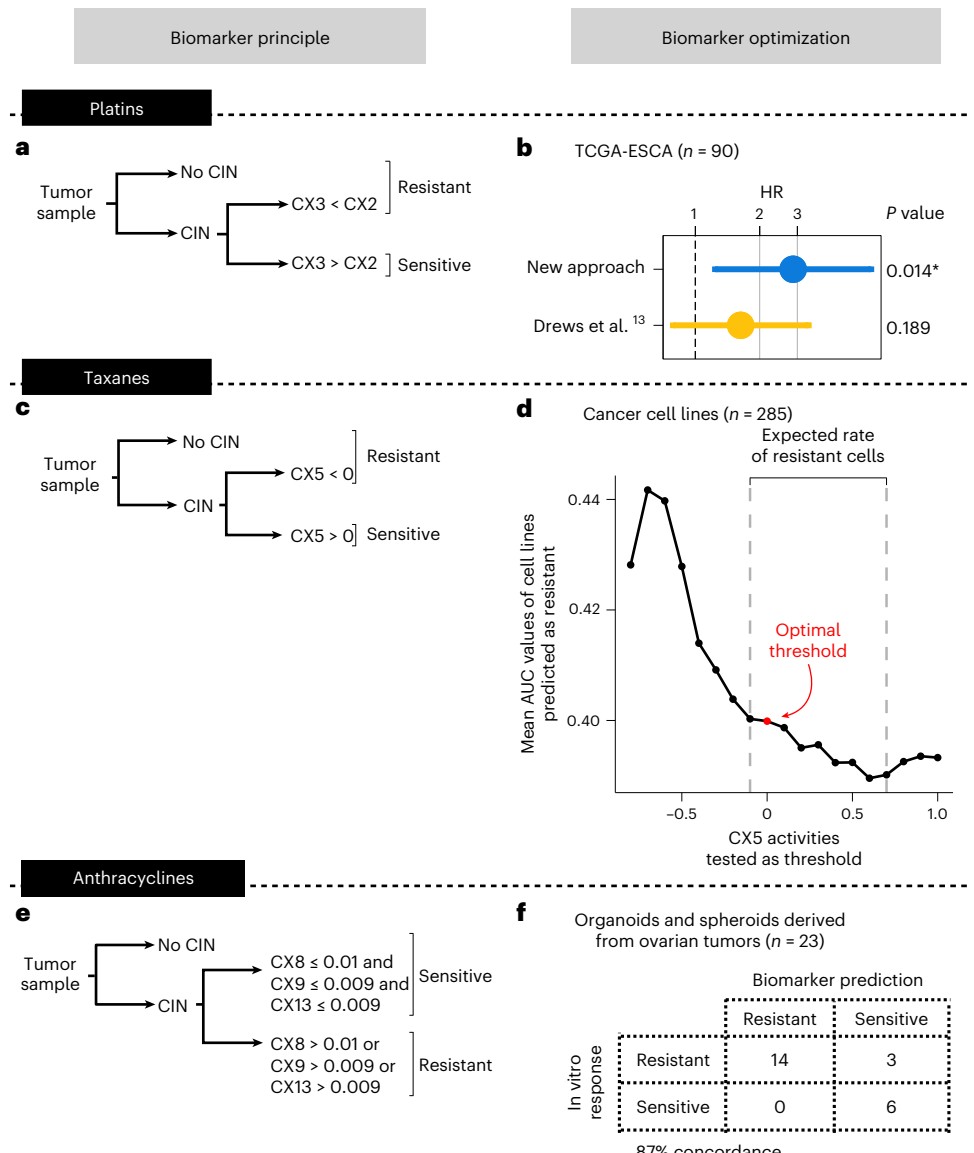

**Fig. 1 | Optimization of biomarkers to predict chemotherapy resistance pan-cancer. a**, The workflow for using CIN signatures as biomarkers for predicting resistance to platinum-based chemotherapies. **b**, Cox proportional-hazards regression models showing overall survival in TCGA esophageal adenocarcinoma (TCGA-ESCA) patients ($n = 90$) classified as predicted or sensitive to platinum-based chemotherapy after applying the classifier from **a**. The dots and error bars represent the HR and its 95% CI, respectively. The Cox proportional-hazards models correct for stage and age at diagnosis. The asterisk denotes a significant result at a level of 0.05. **c**, The workflow for using CIN signatures as biomarkers for predicting resistance to taxanes. **d**, A dot plot showing the mean AUC of cell lines predicted as resistant ($y$ axis) using a range of signature activities for thresholding ($x$ axis). A total of 285 cell lines having high-quality paclitaxel response data were included in the analysis. The red dot denotes the activity value selected as the optimal threshold. The dashed lines show the lower and upper CX5 activity thresholds that match the expected rate of cells as resistant (30% to 60%). **e**, The workflow for using CIN signatures as biomarkers for predicting resistance to anthracyclines. **f**, A contingency table showing the agreement between the observed and the predicted response of patient-derived models to doxorubicin in vitro. Samples with at least one of the three amplification-related signatures (CX8/CX9/CX13) showing an activity higher than the optimal threshold were predicted as resistant.

50 patients from the OV04 study where we could assess biomarker performance after first-line platinum treatment, post-first-line anthracycline treatment and post-first-line taxane treatment (Extended Data Figs. 2–4, Supplementary Figs. 2–4 and Supplementary Table 3).

For each patient, we sequenced tumor material collected at diagnosis using sWGS, derived copy number profiles and applied our three classifiers. Then, if eligible, patients were included in the emulation of a phase 2 biomarker trial, with a single-arm study design where all patients received the chemotherapy of interest (Extended Data Fig. 5a). Patients were considered eligible if they were 18 years or older, confirmed HGSOC, had progression free survival (PFS) intervals measurable by CA125 using the Gynecologic Cancer InterGroup criteria[30,31] and recorded tumor stage.

The primary objective of the trial emulation was to test the biomarker's ability to predict resistance with an endpoint of CA125-based PFS.

For each of the three emulations, we considered patients who received treatment lines containing first-line platinum-based chemotherapies, post-first-line taxanes and post-first-line anthracyclines. Treatment lines were accepted that were either monotherapy or given in combination with other treatments. Predicted resistant and sensitive arms were compared using Cox proportional-hazards models, correcting for tumor stage, age at diagnosis, treatment line (for taxane), general aneuploidy (via the weighted genome instability index (wGII)) and whether the patient received maintenance therapy (during first-line treatment or before treatment with taxane or anthracycline).

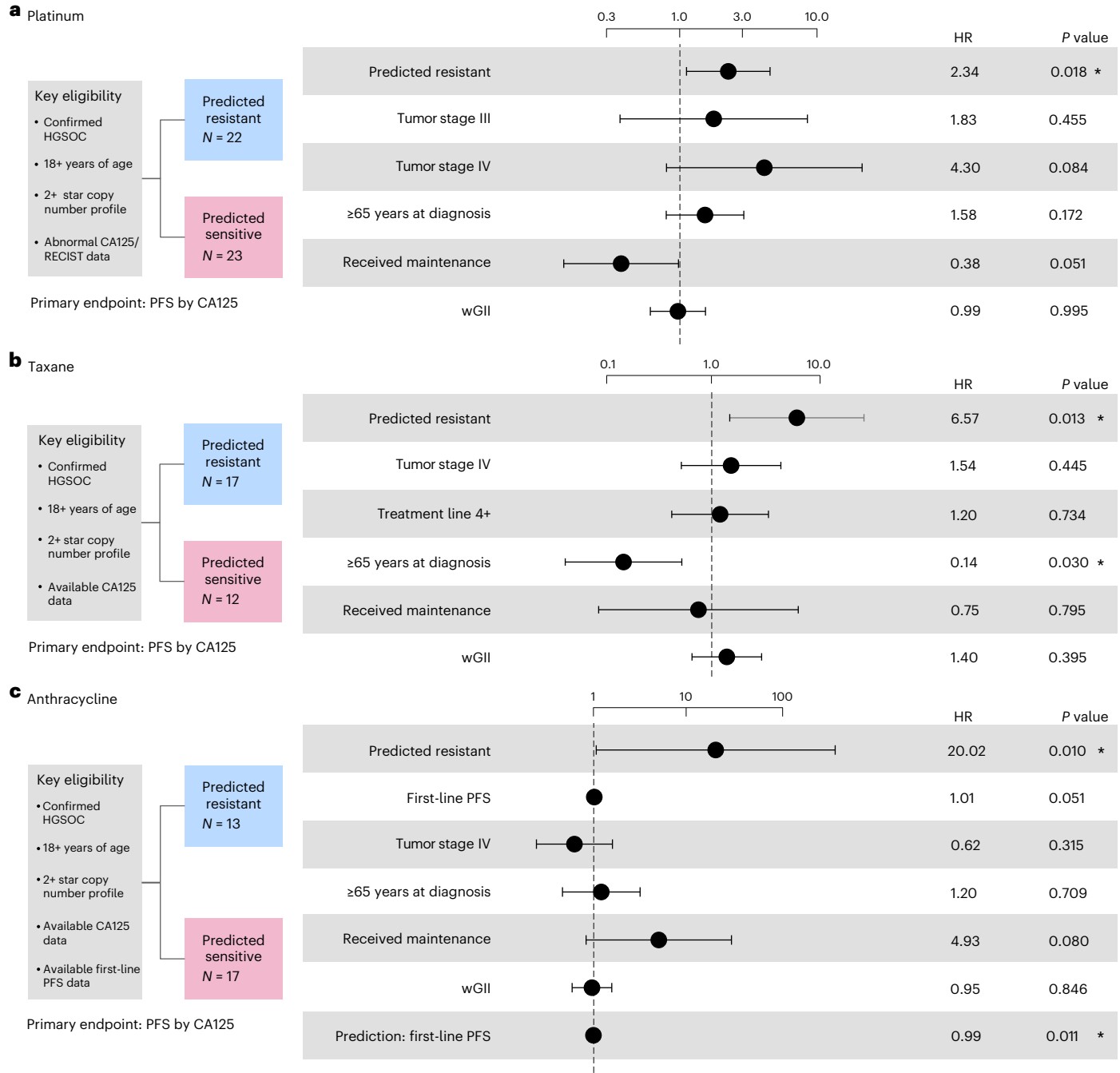

**Fig. 2 | Performance assessment of platinum, taxane and anthracycline resistance prediction in the clinical OV04 study. a**, Cox proportional-hazards model results for predicting resistance to platinum-based chemotherapy. **b**, Cox proportional-hazards model results for predicting resistance to taxane. **c**, Cox proportional-hazards model results for predicting resistance to anthracycline.

PFS was used as the endpoint. All analyses were evaluated at a significance level of 0.05. An asterisk denotes significant results. Dots and error bars represent the HR and its 95% CI, respectively. Multiple testing correction was not applied as each combination of biomarker and cohort is unique.

We defined maintenance therapies as either niraparib, olaparib, bevacizumab or letrozole. For taxane and anthracycline treatments, eligibility criteria and survival analysis design was given careful consideration to control for the effects of first-line platinum-free interval on treatment response at second line[32] (see Supplementary Note 2 for details).

The 22 (out of 45) patients predicted as platinum resistant showed significantly increased risk of progression after treatment with platinum (hazard ratio (HR) of 2.340, 95% confidence interval (CI) 1.155–4.742, $P$ = 0.0183; Fig. 2a). The 17 (out of 29) patients predicted as taxane resistant also showed significantly increased risk of progression after

treatment post-first-line with taxane (HR of 6.567, 95% CI 1.489–28.957, $P$ = 0.013; Fig. 2b). Finally, the 13 (out of 30) patients predicted as anthracycline resistant showed increased risk of progression after treatment with anthracycline post-first-line (HR of 20.020, 95% CI 1.059–378.635, $P$ = 0.010; Fig. 2c). For all analyses, the significance level for detecting an effect was 0.05.

**Emulating biomarker trials to assess performance**

Since the pilot study confirmed that retrospective real-world data can be used to test performance, we sought orthogonal validation using

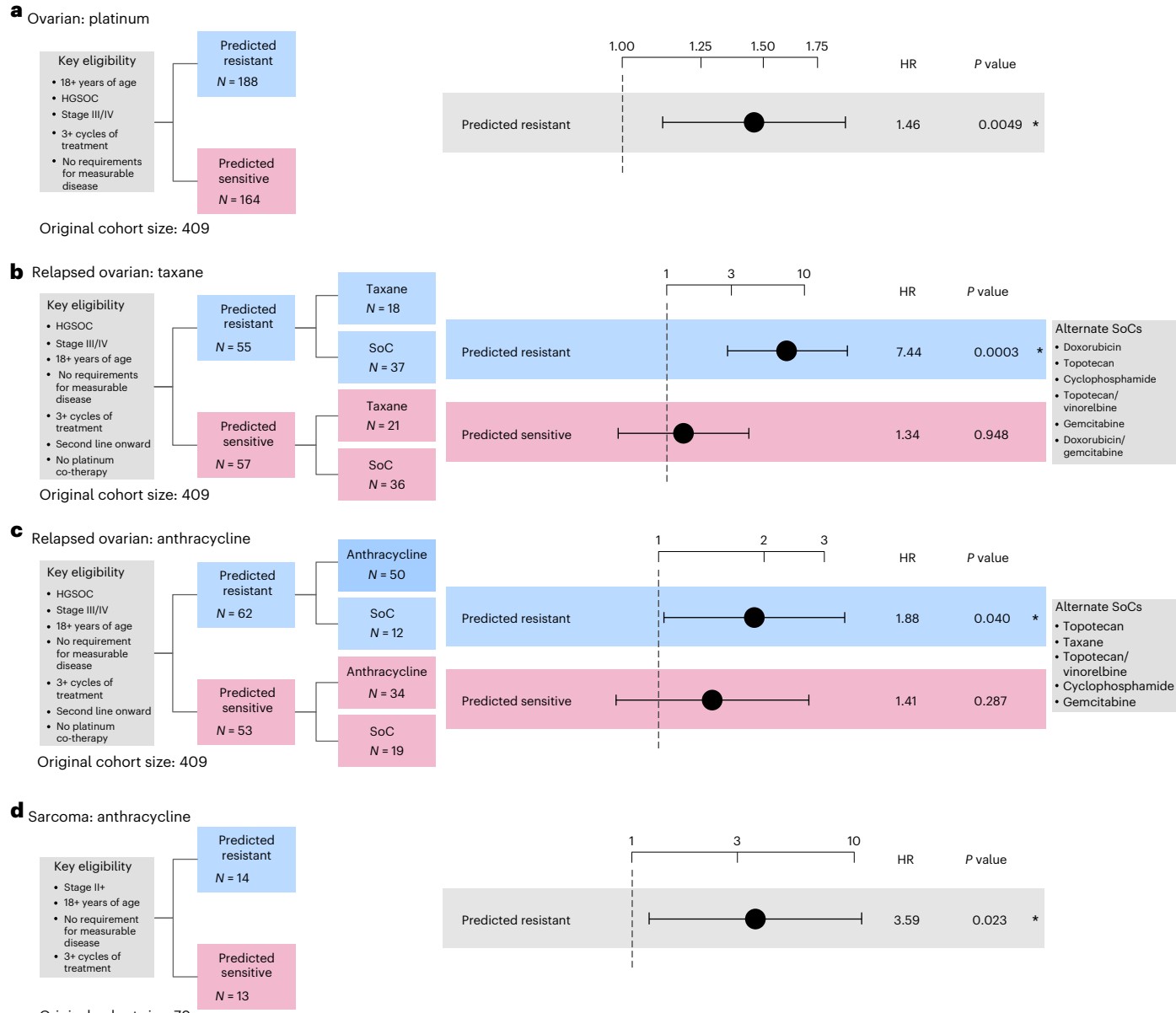

**Fig. 3 | Performance assessment of platinum, taxane and anthracycline resistance prediction across real-world cohorts of primary tumors. a**, Cox proportional-hazards regression models showing TTF in patients with primary ovarian cancer predicted as resistant to first-line platinum-based treatment stratified by age at diagnosis (<60, 60–69 and ≥70 years old) and controlling for tumor stage. **b**, Cox proportional-hazards regression models showing TTF in patients with relapsed ovarian cancer predicted as resistant to taxane stratified by age at diagnosis (<65 and ≥65 years) and controlling for an interaction term between the treatment arm and first-line TTF. The reported HR is a point estimate at 6 months after first-line treatment (further details in Supplementary Note 2). **c**, Cox proportional-hazards regression models showing TTF in patients with

relapsed ovarian cancer predicted as resistant to anthracycline stratified by age at diagnosis (<65 and ≥65 years) and controlling for platinum sensitivity (≤6 and >6 months first-line TTF). **d**, Cox proportional-hazards regression models showing TTF in patients with primary sarcoma predicted as resistant to anthracycline controlled for ifosfamide as a co-therapy. The limited sample size precluded us from correcting the model by other clinical covariates. All analyses were evaluated at a significance level of 0.05. An asterisk denotes significant results. The dots and error bars represent the HR and its 95% CI, respectively. Multiple testing correction was not applied as each combination of biomarker and cohort is unique. Inverse probability weighting was applied in all analyses to account for treatment selection biases across patients due to year of treatment/biopsy.

additional real-world data to emulate either phase 3 RCTs (Extended Data Fig. 5b), phase 3 RCT enrichment trials (Extended Data Fig. 5c) or phase 2 single-arm trials (Extended Data Fig. 5a).

To achieve this, we assembled a series of pan-cancer real-world retrospective cohorts[33,34] consisting of patients with primary or metastatic disease who were treated with one of the chemotherapies of interest or with an alternate standard of care (SoC), had clinical response data enabling time to treatment failure (TTF) calculation and had sufficient quality genomic data to compute CIN signatures. To identify which

cohorts were powered for phase 3 or phase 2 analysis, we used the HRs determined from the pilot study, alongside censoring and prediction ratio data from the cohorts, to carry out calculations for a one-tailed power analysis with a power of 0.8 and a significance level of 0.05 (Supplementary Table 4). Cohorts that had sufficient sample sizes to meet the power requirements were then taken forward for analysis (summarized in Extended Data Tables 1 and 2). We were sufficiently powered to perform phase 3 emulation assessments for patients with relapsed ovarian, metastatic breast and metastatic prostate cancer

treated with taxanes, and patients with relapsed ovarian and metastatic breast treated with anthracyclines. We were sufficiently powered to perform phase 2 emulation assessments (but not phase 3) for patients with primary ovarian cancer treated with platinum and sarcoma treated with anthracyclines.

In the phase 3 RCT emulations, patients were classified as resistant or sensitive to the chemotherapy of interest based on our biomarkers. Within these sensitive or resistant groups, patients were then retrospectively assigned to the experimental arm (treated with the chemotherapy of interest) or to the control arm (treated with an appropriate SoC therapy). Inverse probability weighting was applied to account for potential treatment selection biases across patients due to year of treatment/biopsy. The experimental and control arms were then compared using Cox proportional-hazards models for both the predicted resistant and predicted sensitive populations, with TTF as the primary endpoint.

In the phase 2 single-arm emulations, patients that were treated with the chemotherapy of interest were classified as resistant or sensitive based on our biomarkers, then these groups were compared using a Cox proportional-hazards model with TTF as the primary endpoint.

**Platinum resistance in primary ovarian cancer (single arm).** Prediction of resistance to platinum-based treatment in primary HGSOC was assessed using a cohort of 352 patients (Supplementary Table 5). Patients were considered eligible if they were 18 years of age or over, had grade 3 and stage III/IV cancer, with no requirement for measurable disease (according to the Response Evaluation Criteria In Solid Tumors (RECIST) 1.0). Statistical analysis of the cohort was performed using a stratified Cox proportional-hazards model by age at diagnosis (<65 or ≥65 years) and controlling for tumor stage (III or IV). Patients predicted as platinum resistant showed significantly increased risk of treatment failure compared with those patients classified as sensitive (HR of 1.459, 95% CI 1.121–1.899, $P = 0.0049$; Fig. 3a and Supplementary Fig. 5).

**Taxane resistance in relapsed ovarian cancer (RCT).** Prediction of resistance to taxane-based treatment in relapsed HGSOC was assessed across a cohort of 112 patients (Supplementary Table 5). Eligibility criteria, trial design and survival analysis were given careful consideration to control for the effects of first-line platinum-free interval on treatment response at second line[32] (Supplementary Note 2). Patients were considered eligible if they were 18 years of age or over, had grade 3 and stage III/IV cancer, with no requirement for measurable disease. Patients were assigned to the experimental arm if they received at least three cycles of taxane treatment post-first line. Patients were assigned to the control arm if they received treatment containing any of the five most common non-taxane- and non-platinum-based therapies, being doxorubicin, gemcitabine, cyclophosphamide, topotecan and vinorelbine. Statistical analysis of the cohort was performed using stratified Cox proportional-hazards models by age at diagnosis (<65 or ≥65 years), including an interaction between first-line TTF and treatment group (see Supplementary Note 2 for details).

In patients predicted to be resistant, the use of taxane-based treatment resulted in significantly higher risk of treatment failure compared with the SoC treatment (HR of 7.435, 95% CI 3.967–20.458, $P = 0.0003$; Fig. 3b, Extended Data Fig. 6a and Supplementary Fig. 6), suggesting predicted resistant patients do not derive benefit from taxane treatment. By contrast, the predicted sensitive patients showed no significant difference in TTF between the experimental and control arms (Fig. 3b, Extended Data Fig. 6b and Supplementary Fig. 7). This is in agreement with previous studies reporting comparable response among different chemotherapies after first-line treatment[35]. We found similar results when restricting the analysis to patients receiving single-agent taxane treatments, indicating that the differences in TTF between the experimental and control arms were not caused by the co-therapy administered (Supplementary Figs. 8 and 9).

**Anthracycline resistance in relapsed ovarian cancer (RCT).** The prediction of resistance to anthracycline-based treatment in HGSOC was conducted in a cohort of 115 patients (Supplementary Table 5). Patients were considered eligible if they were 18 years of age or over and had grade 3 and stage III/IV cancer. There was no requirement for measurable disease. Patients were assigned to the experimental arm if they received at least three cycles of anthracycline and patients were assigned to the control arm if they received treatment containing any of the five most common non-anthracycline- and non-platinum-based therapies, being gemcitabine, vinorelbine, cyclophosphamide, taxane and topotecan. Statistical analysis of the cohort was performed using stratified Cox proportional-hazards models by age at diagnosis (<65 or ≥65 years) and controlling for platinum sensitivity (≤6 months or >6 months first-line TTF).

In patients predicted to be resistant, the use of anthracycline-based treatment resulted in significantly increased risk of treatment failure compared with the SoC treatment (HR of 1.881, 95% CI 1.029–3.439, $P = 0.040$; Fig. 3c, Extended Data Fig. 7a and Supplementary Fig. 10). By contrast the predicted sensitive analyses showed no significant difference in TTF between the experimental and control arms (Fig. 3c, Extended Data Fig. 7b and Supplementary Fig. 11). Similar results were found restricting to single-agent anthracycline treatment (Extended Data Fig. 7c and Supplementary Figs. 12 and 13).

**Anthracycline resistance in sarcoma (single arm).** Prediction of resistance to anthracycline-based treatment in sarcoma was assessed across 27 patients (Supplementary Table 5). Patients were considered eligible if they were 18 years or over with no requirement for measurable disease. Statistical analysis of the cohort was performed using a Cox proportional-hazards model controlling for the presence of isophosphamide as a co-therapy[36]. Tumor stage annotation was not available for this cohort and so was not included in the analysis. Patients predicted to be resistant demonstrated a significantly increased risk of treatment failure when they received anthracycline-based treatment compared with those predicted as sensitive (HR of 3.591, 95% CI 1.193–10.811, $P = 0.023$; Fig. 3d and Supplementary Fig. 14).

**Taxane resistance in metastatic prostate cancer (RCT).** Prediction of resistance to taxane-based treatment was conducted in a cohort of 238 patients with metastatic prostate cancer (Supplementary Table 5). Patients were considered eligible if they were 18 years or over and had presented with metastatic tumors. Treatment lines were only considered if they occurred immediately after a biopsy event. Patients were assigned to the experimental arm if they received at least three cycles of taxane or to the control arm if they had received any of the five most common non-taxane therapies given within the cohort: abiraterone, enzalutamide, olaparib, pembrolizumab or radium-223. Statistical analysis of the cohort was performed using Cox proportional-hazards models controlling for age at diagnosis (46–85 years). Tumor stage annotation was not available for this cohort.

Patients predicted to be resistant demonstrated a significantly increased risk of treatment failure compared with SoC (HR of 5.462, 95% CI 2.189–13.628, $P = 0.0003$; Fig. 4a, Extended Data Fig. 6c and Supplementary Fig. 15), while no significant differences were detected between treatment arms for patients predicted to be sensitive (Fig. 4a, Extended Data Fig. 6d and Supplementary Fig. 16). Similar results were observed when limiting the analyses to single-agent taxane treatment (Supplementary Figs. 17 and 18).

**Taxane resistance in metastatic breast cancer (RCT).** Prediction of resistance to taxane-based treatment was conducted across 134 patients with metastatic breast cancer (Supplementary Table 5). Patients were considered eligible if they were 18 years or over and had presented with metastatic tumors. Patients were assigned to the experimental arm if they received at least three cycles of taxane and to

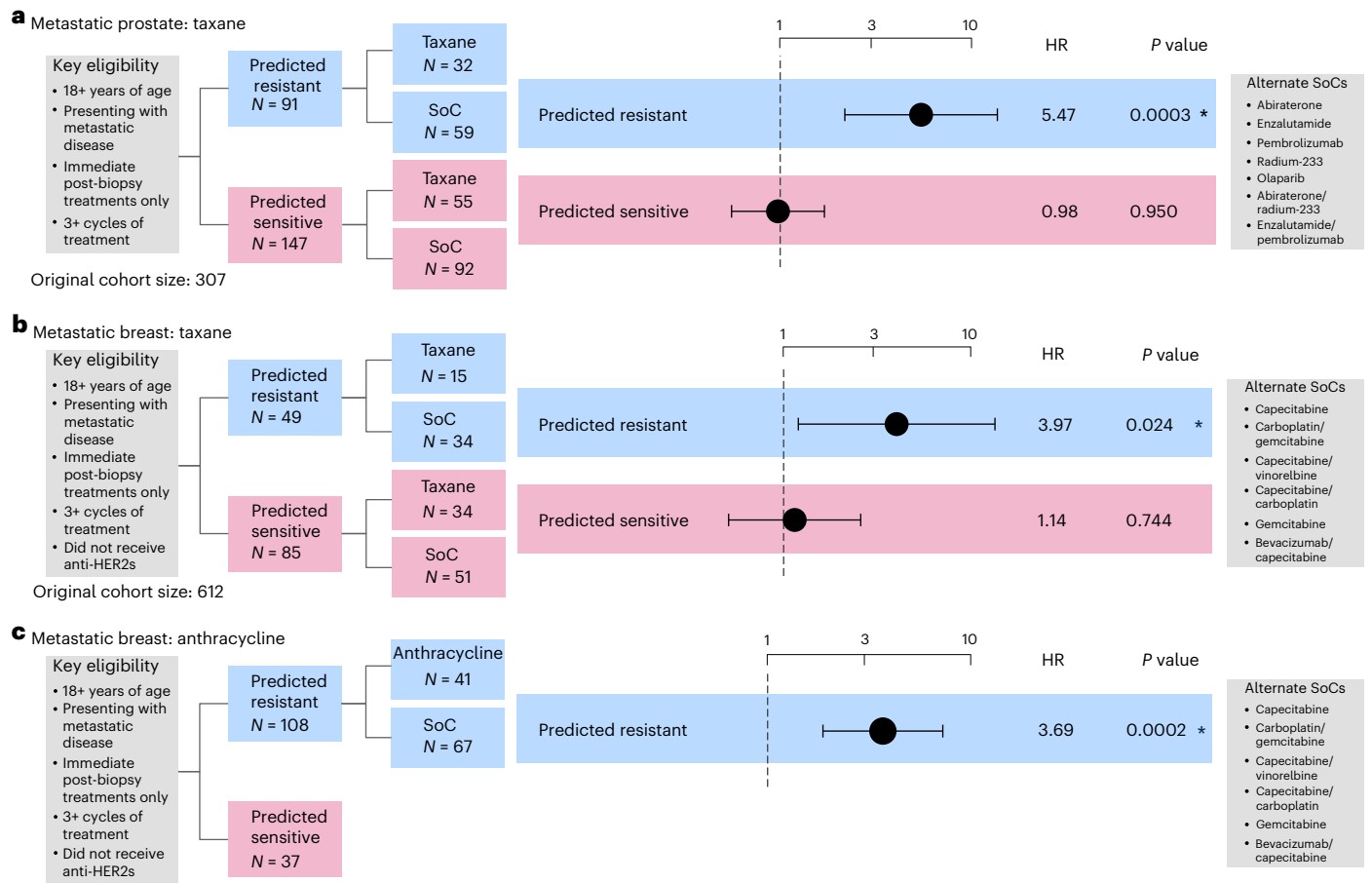

**Fig. 4 | Performance assessment of taxane and anthracycline resistance prediction across real-world cohorts of metastatic tumors. a**, Cox proportional-hazards regression models showing TTF in patients with metastatic prostate cancer predicted as resistant to taxane controlled for age at diagnosis. Gleason grade was not available for correction. **b,c**, Cox proportional-hazards regression models showing TTF in metastatic breast patients predicted as resistant to taxane (**b**) and anthracycline (**c**). Regression models were controlled for age at diagnosis. The limited sample size in the anthracycline-sensitive arm precluded us from performing a survival analysis. All analyses were evaluated at a significance level of 0.05. An asterisk denotes significant results. The dots and error bars represent the HR and its 95% CI, respectively. Multiple testing correction was not applied as each combination of biomarker and cohort is unique. Inverse probability weighting was applied in all analyses to account for treatment selection biases across patients due to year of treatment/biopsy.

the control arm if they received gemcitabine or capecitabine, in some cases in combination with bevacizumab, vinorelbine, gemcitabine or carboplatin. Patients were not included if they received hormone therapies. Statistical analysis of the cohort was performed using a Cox proportional-hazards model controlling for age at diagnosis (33–80 years).

Patients predicted to be resistant demonstrated a significantly increased risk of treatment failure compared with SoC (HR of 3.976, 95% CI 1.196–13.219, $P$ = 0.024; Fig. 4b, Extended Data Fig. 6e and Supplementary Fig. 19), while no significant differences were detected in survival time between the treatment arms for patients predicted to be sensitive (Fig. 4b, Extended Data Fig. 6f and Supplementary Fig. 20).

**Anthracycline resistance in metastatic breast cancer (enrich).** The prediction of resistance to anthracycline-based treatment in breast cancer was conducted in a cohort of 108 patients (Supplementary Table 5). Patients were considered eligible if they were 18 years of age or over and had stage III/IV cancer with no requirement for measurable disease. Patients were assigned to the experimental arm if they received at least three cycles of anthracycline-based treatment and assigned to the control arm if they received treatment containing any of the five most common non-anthracycline therapies, being bevacizumab, vinorelbine, gemcitabine or carboplatin. Statistical analysis

of the cohort was performed using Cox proportional-hazards models controlling for age at diagnosis.

Patients predicted to be resistant showed significantly increased risk of treatment failure compared with SoC (HR of 3.685, 95% CI 1.866–7.277, $P$ = 0.0002; Fig. 4c and Supplementary Fig. 21). We were not powered to determine performance in the predicted sensitive group, thus this emulation is considered to have an enrichment design (Extended Data Fig. 5c).

**Assessing routes to clinical implementation**
In the analyses presented here, our CIN signature biomarkers were determined using copy number profiles derived from either deep whole-genome sequencing (WGS) or sWGS, or Affymetrix SNP6 arrays. However, these assays are not yet routine in clinical practise. Comprehensive capture gene panels currently dominate the diagnostic DNA testing landscape in oncology. Therefore, we wanted to assess the feasibility of using the Illumina TruSight Oncology 500 (TSO500) assay to apply our resistance classifications. In addition, we also assessed the feasibility of using sWGS from liquid biopsies collected at the time of diagnosis. For eight tumors in the Cambridge Translational Cancer Research Ovarian Study 04 (OV04) pilot study with greater than 40% tumor purity, we performed profiling of the same DNA aliquot as the sWGS using the TSO500 assay; and for

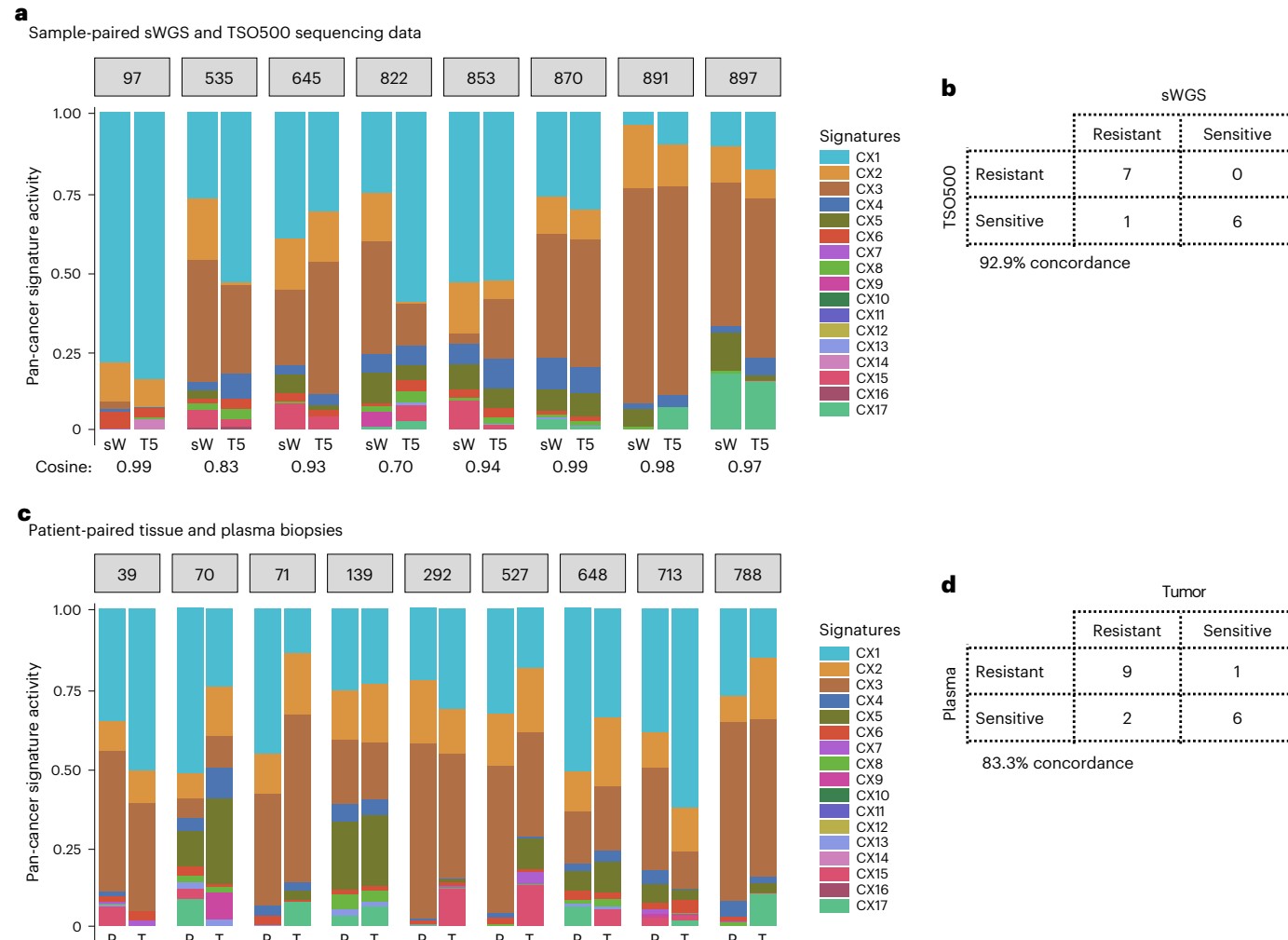

**Fig. 5 | Comparison of response prediction between paired samples. a**, A bar plot showing the activities of the 17 CIN signatures (CX) in tumor biopsies sequenced by using both sWGS (sW) and TSO500 (T5). Only tissue biopsies from patients with high-quality sWGS-derived copy number profiles that met the inclusion criteria were also sequenced with TSO500. A cosine similarity close to 1 indicates similar activities between sample-matched data. **b**, A contingency table showing the number of patients predicted as sensitive or resistant using our signature-based clinical classifiers in tumor samples sequenced by sWGS and TSO500. **c**, A bar plot showing signature activities in matched tumor tissue (T) and plasma (P) samples from the same patient. Only high-quality plasma samples were used for deriving copy number profiles and signature quantification. A cosine similarity close to 1 indicates similar activities between patient-matched biopsies. **d**, A contingency table showing the number of patients predicted as sensitive or resistant using our signature-based clinical classifiers in tumor tissue and plasma biopsies. The numbers in gray boxes represent the patient IDs.

29 patients, plasma samples were used to extract cell-free DNA, which underwent sWGS (Extended Data Table 3).

For samples profiled with the TSO500 assay, the copy number profiles derived showed a median percentage of genome-wide copy number difference of 16% (Supplementary Fig. 22). CIN signatures computed across these samples showed a median cosine similarity with the matched sWGS of 0.92 (Fig. 5a). Applying our classifiers showed 93% concordance, with only one sample given a prediction that did not match the sWGS-based prediction (Fig. 5b).

For cell-free DNA samples, DNA copy number profiles were generated and samples were categorized based on their circulating tumor DNA fraction as either low or high. Of the 29 patients who had plasma samples available, 9 samples (31%) were considered to have high circulating tumor DNA fraction and were subjected to CIN signature analysis. The remaining plasma samples had insufficient overall tumor DNA to assess CIN using the currently available CIN signature methods. Between plasma and tumor tissue pairs, the median percentage of genome-wide copy number difference was 20.8% (Supplementary

Fig. 23), and activity levels of CIN signatures had a median cosine similarity of 0.90 (Fig. 5c). Resistance classification for relevant therapies across the patients showed 83.3% concordance (Fig. 5d), with two patients showing alternate predictions for platinum and one patient for taxane. These results suggest that for approximately 31% of patients with ovarian cancer, our predictors may be applied using a simple blood test, without the need for a tumor biopsy or a surgical specimen, although testing over larger cohorts is necessary.

## Discussion

In this study, we demonstrate the potential for CIN signature biomarkers to predict resistance to multiple chemotherapies. Previous approaches have largely focused on platinum-based chemotherapies and rely on cell culture or gene expression assays[37,38]. However, these tests have failed to reach widespread adoption in the clinic. The OncotypeDX recurrence score is the only test to have been widely adopted; however, it is not a direct test for chemotherapy response, rather it determines whether a patient will be adequately managed

with hormone therapies alone (compared with a combination with chemotherapy)[39]. Homologous recombination deficiency tests such as Myriad myChoice or HRDetect may also have the potential to predict response to platinum-based treatment[40]. However, these do not appear to perform as well as our CIN signature biomarkers (Supplementary Note 3). The ability to predict resistance to multiple chemotherapies from a single genomic assay is thus a unique offering.

Our analysis encompassed three common chemotherapies, platinum-based, taxanes and anthracyclines. When testing the performance of our biomarkers we leveraged real-world data to retrospectively emulate phase 3 RCTs. However, we were unable to emulate a phase 3 biomarker trials for our platinum biomarker in ovarian and anthracycline biomarker in sarcoma. This was due to an absence of alternative SoC treatments at first line. Instead, we pursued phase 2 single-arm studies. Further follow-up work is required to assess the true predictive capacity of the resistance biomarkers in this context.

Further work will also be necessary to understand the regulatory pathway of a clinical decision support test for already approved chemotherapies. One of the main challenges to overcome will be the heterogeneity of genomic testing in a clinical environment. Currently, different hospital systems employ different genomic assays including gene panel sequencing and WGS. Thus, the use of CIN signature biomarkers for chemotherapy response prediction will need to be enabled across a variety of technologies. In this regard, we showed a proof-of-concept that our resistance classifications may be applied using the regulatory approved TSO500 assay, alongside shallow and deep WGS, and single-nucleotide polymorphism (SNP) arrays. Furthermore, in a subset of patients, it may be possible to perform the predictions using liquid biopsies. However, further studies will be necessary to determine the optimal clinical implementation strategy and will need to assess the trade off between taking a biopsy or blood draw as well as assessing the risk of biopsy, the cost of applying the test to blood draws or tissue, the fraction of patients with adequate biopsy material and the stage at which the test is administered, to name a few.

Importantly, our study introduces biomarkers for patient stratification for multiple medicines that were not originally developed as targeted therapies. CIN signature analysis can be applied widely across cancer types[13] and thus our results have broad future implications for patient stratification and precision medicine in cancer.

## Online content

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

¹Spanish National Cancer Research Centre (CNIO), Madrid, Spain. ²Tailor Bio Ltd, Cambridge, UK. ³Cancer Research UK Cambridge Institute, University of Cambridge, Cambridge, UK. ⁴Department of Oncology, University of Cambridge, Cambridge, UK. ⁵Cambridge University Hospitals NHS Foundation Trust, Cambridge, UK. ⁶Cancer Research UK Major Centre Cambridge, University of Cambridge, Cambridge, UK. ⁷H12O–CNIO Lung Cancer Clinical Research Unit, Health Research Institute Hospital 12 de Octubre, Madrid, Spain. ⁸Institute for Women's Health, University College London, University College London Hospitals Biomedical Research Centre, London, UK. ⁹Early Phase Clinical Trials Team, Department of Oncology, University of Cambridge, Cambridge, UK. ¹⁰These authors contributed equally: Joe Sneath Thompson, Laura Madrid, Barbara Hernando. ¹¹These authors jointly supervised this work: Anna M. Piskorz, Geoff Macintyre. ✉e-mail: anna.piskorz@cruk.cam.ac.uk; gmacintyre@cnio.es

## Methods

### Statistics and reproducibility

Details on statistical analyses are outlined in each relevant section of Methods and the Supplementary Information. All code is provided to completely reproduce all analyses performed in this study and all experimental details are outlined in relevant sections of Methods and the Supplementary Methods. For biomarker discovery and optimization, no statistical method was used to predetermine sample size. For biomarker performance assessment, power analyses were conducted to ensure the clinical cohorts were well powered (detailed in the Supplementary Methods). In vitro experiments and single-arm trials were not randomized. For retrospectively emulated phase 3 trials, patients were pseudo-randomized to the control and experimental arm. Patients were filtered for biomarker clinical assessment based on the exclusion criteria detailed in the 'Cohort curation' sections of the Supplementary Methods. Flowcharts detailing inclusion and exclusion criteria for all cohorts used in this study are available via Figshare at ref. [41].

### Sample cohorts

**The OV04 study.** Clinical data and samples for patients with HGSOC were collected as part of the prospective Cambridge Translational Cancer Research Ovarian Study 04 (CTCROV04) approved by the Institutional Ethics Committee (REC07/Q0106/63). OV04 is an ongoing observational study that records patient clinical data and collects patient material for the purpose of biomarker and scientific discovery. Patients provided written, informed consent for participation in this study and for the use of their donated tissue for the laboratory studies carried out in this work. The samples included primary ascites spheroids, tumor tissue, plasma samples and tumor tissue-derived organoids and cell lines. DNA was extracted and sequenced, copy number profiles derived, CIN signatures computed and biomarkers called (see details below). Treatment histories for each patient are shown in Supplementary Figs. 24–73. Clinical data (where applicable) were curated and response determined (further details in the Supplementary Methods).

**TCGA collection.** In this study, we used data from the TCGA consisting of high-quality copy number profiles from 7,880 patients and matching clinical records for 7,105 patients, representing 33 cancer types. These data were curated to provide clinical response data and biopsy-level activities of CIN signatures for biomarker performance analysis (further details in the Supplementary Methods).

**The HMF dataset.** In this study we use data from The Hartwig Medical Foundation (HMF), which manages a multicenter database of genomic and clinical data collected from 2,979 patients with metastatic cancer in the Netherlands. This dataset, representing 35 cancer types, was curated to provide clinical response data and biopsy-level activities of CIN signatures for biomarker performance analysis (further details in the Supplementary Methods).

### OV04 sample processing and DNA isolation

**Tissue samples.** Formalin-fixed, paraffin-embedded tissue blocks were cut as 8 μm sections and tumor-enriched regions were recovered by macrodissection based on regions marked on an adjacent hematoxylin and eosin-stained section by the study pathologist. DNA was extracted from 3–10 sections at 8 μm thickness using the QIAmp DNA Micro kit (Qiagen, 56204) with the following modification to the original protocol: an additional incubation step with buffer ATL at 95 °C for 15 min was introduced before adding proteinase K. The paraffin was removed using a xylene/ethanol method. DNA extraction from fresh frozen tumor tissues and spheroids fraction was performed using the Allprep DNA/RNA tissue kit (Qiagen, 80204) following the manufacturer's instructions.

**Plasma samples.** We focused on selected plasma time points collected before the primary line of chemotherapy treatment and before anthracycline treatment (usually before the second or third line of therapy). DNA was extracted from 2 or 4 ml of plasma using the QIAamp circulating nucleic acid kit (Qiagen, 55114) or QIAsymphony (Qiagen, 937556), according to the manufacturer's instructions.

**Organoids.** Organoids were derived as previously described[42]. Samples were obtained from patients via surgical resection, ward drains or surgical washings. Solid tumors were assessed by a pathologist and only tumor samples with ≥50% tumor cellularity were selected for organoid model derivation. Organoid culture medium was refreshed every 2 days. To passage the organoids, the domes were scraped and collected in a Falcon tube, TrypLE (Invitrogen, 12604013) was added and they were incubated at 37 °C for approximately 10 min. The suspension was centrifuged at 800$g$ for 2 min and the cell pellet was resuspended in 7.5 mg ml$^{-1}$ BME-2 supplemented with complete media and plated as 20 μl droplets in a six-well plate. After allowing the BME-2 to polymerize, complete medium was added and cells incubated at 37 °C. DNA was extracted from cell pellets using the Qiagen Allprep DNA/RNA (Qiagen, 80204) extraction kit according to the manufacturer's instructions.

**Spheroids.** Ascitic fluid was collected from patients, with between 100 ml and 2 l volume. The fluid was initially gently centrifuged at 800$g$ for 5 min and the majority of the supernatant was removed. The sample was filtered using autoclaved muslin cloth and the flow through was then filtered again using a 40 μm cell strainer. Spheroids from the strainer were then recovered by a 10 ml wash with PBS and centrifuged at 1,500$g$ for 5 min. The spheroid fraction was divided in two: a cell pellet for DNA extraction and resuspension of cells in filtered acellular ascitic supernatant and 8% dimethylsulfoxide for the drug screen. Spheroids were thawed and put in media overnight to fully recover before dispensing for the drug screen. DNA extraction was performed using the Qiagen Allprep DNA/RNA (Qiagen, 80204) extraction kit according to the manufacturer's instructions.

**Cell lines.** All cell lines were maintained in Dulbecco's modified Eagle medium/F12 or RPMI1640 plus 10% of fetal calf serum. Cell line identities were confirmed by short tandem repeat profiling. Cells were regularly screened for mycoplasma using a MycoAlert Mycoplasma Detection kit (Lonza, LT07-118). The sulforhodamine B colorimetric assay was used for quantifying cell numbers and cell proliferation in culture. DNA extraction from cell pellets of approximately $2 \times 10^6$ million cells per sample was performed using the Qiagen Allprep DNA/RNA (Qiagen, 80204) extraction kit following the manufacturer's recommendations.

### OV04 DNA sequencing

**sWGS.** *Tissue samples, organoids, spheroids and cell lines.* WGS libraries were prepared from 50 ng DNA using Illumina DNA prep (S) Tagmentation (Illumina, 20025523) and SMARTer Thruplex DNA-Seq (Takara, R400676) reagents, following the manufacturer's protocol. Library quality and quantity were assessed with D5000 on a 4200 Tapestation, a Fragment Analyzer next-generation sequencing (NGS) kit (Agilent Technologies, 5067-5582) and the Qubit BR dsDNA assay, according to the supplier's recommendations. Libraries were then diluted to 10 nmol l$^{-1}$ and pooled together in equal ratios and sequenced using paired-end (PE) 50 mode on a NovaSeq6000 S2 flow cell 100 cycles kit (Illumina, 20028316), aiming for 80 million reads per sample.

*Plasma samples.* A total of 10 μl of extracted circulating nucleic acids was taken as input for whole-genome library preparation using the SMARTer ThruPLEX DNA-Seq (Takara, R400676) library prep kit with the following modifications: no DNA shearing was performed, 14 PCR cycles were applied, library purification using Ampure beads (Beckman Coulter, 10136224) was performed separately for each sample and elution was performed using 20 μl of Tris EDTA buffer. Generated

libraries were quantified using the Fragment Analyzer NGS kit (Agilent Technologies, cat no. DNF-467-0500) diluted to 10 nmol l$^{-1}$ and pooled in the same proportions. All libraries were sequenced using the NovaSeq6000 S2 flow cell 300 cycle kit (Illumina, 20028314) using PE 150 bp mode to achieve at least 80 million reads per sample.

**Illumina TSO500 assay.** Hybridization-based NGS libraries were prepared from 40 ng DNA using the TruSight Oncology 500 Library Preparation kit (Illumina, 20076480), following the manufacturer's protocol. Library quality and quantity were assessed with the High Sensitivity/D5000 Screentape assay (Agilent Technologies, 5067-5592) on a 4150/4200 Tapestation Quant-IT/Qubit dsDNA HS (Qiagen, Q32851) assay system according to the supplier's recommendations. Libraries were then pooled together in equal ratios and sequenced using PE 150 bp mode on a NovaSeq S1 flow cell 300 cycles kit (Illumina, 20028317) or S4 flow cell 300 cycles kit (Illumina, 20028312) aiming for 100 million reads per sample.

### OV04 read alignment
Reads were aligned as single-end against the human genome assembly GRCh37 using BWA-MEM v0.7.17 (ref. [43]), following which duplicate reads were identified and marked using the MarkDuplicates tool in the GATK v4.1.8.1 (ref. [44]) tool suite.

### Copy number profiling
**OV04 sWGS.** After alignment, relative copy number was computed using QDNAseqmod[45] with a bin size of 50 kb. Bins mapped to centromeres and regions of undefined sequence in the reference genome hg19 were excluded. Read counts were corrected for the relationship between sequence mappability and GC content.

Absolute tumor copy number (the number of chromosome copies of each DNA segment in the tumor cells in a sample) was computed for every bin across each sample. Each segmented relative copy number bin estimate $j$ was transformed from relative copy number (rCN) to absolute copy number (aCN) by

$$aCN_j = \frac{1}{Purity} \times \left( \frac{rCN_j}{d} - 2 \times (1 - Purity) \right) - 2,$$

where purity is the fraction of tumor cells in the sample and $d$ is a constant proportional to the read depth, which is computed from the mean relative copy number of the sample, $r$, and the average absolute copy number of the tumor cells in the sample, ploidy, by

$$d = \frac{r}{(Ploidy \times Purity + 2 \times (1 - Purity))}.$$

Both purity and ploidy were unobserved in the data and were estimated using a grid search of purities ranging from 0.05 to 1 in 0.01 increments and ploidies ranging from 1.8 to 8 in increments of 0.1, minimizing the following mean squared error:

$$e_{purity,ploidy} = \frac{1}{J} \times \sum_{j=1}^{J} (aCN - round(aCN))^2.$$

Cell lines and organoid samples were assumed 100% pure, so purity was fixed to 1 and a search was only performed across ploidy states. Purity/ploidy values were excluded from consideration if they resulted in a fit which showed greater than 10 megabases of the genome with homozygous loss. For tissue samples, an additional filter was used, removing fits which did not show at least one genomic segment at every integer copy number state from 1 to ploidy.

**OV04 TSO500.** After alignment, PE raw reads were split into equally-sized bins of 50 kb in size. Bins were annotated with GC content, mappability and replication timing, after a bin-level filtering based on a panel of 40 normal samples sequenced by deep whole-genome sequencing as part of the 1000 Genomes Project[46]. The remaining annotated bins were then interrogated for overlaps with target regions (bed file). Only bins with less than 25% overlap with the target regions and a maximum of 10 total overlaps per bin were kept. For the remaining set of bins, we removed on-target reads and counted the number of reads per bin. To correct for artificially high off-target read counts in parts of the genome with high sequence similarity to the target regions, we generated a score per bin that quantified the magnitude of this bias and used it for a single linear model fit and correction. For bins with any overlap with the bed file, we filled the gap caused by the removal of the on-target reads with a pseudocount estimation using the per-bin mean off-target read count. GC and replication timing corrections were performed using locally estimated scatter plot smoothing fitting and correction. We segmented these data using the same segmentation procedure used in QDNAseqmod[45]. Absolute copy numbers were inferred as described above in the section 'OV04 sWGS'. For purity/ploidy selection, we fixed a range of ±0.2 with respect to the ploidy and ±0.05 with respect to the ploidy using the paired sWGS curated data.

**Hartwig WGS.** We downloaded copy number profiles derived using PURPLE from a total of 5,200 samples released by the HMF[34,47]. As PURPLE[47] determines the allele-specific copy number of every base of the genome, the genome binning resolution is substantially higher compared with the resolution of the copy number profiles used to derive feature components of the CIN signature encoding[13]. To avoid incorrect mapping of signatures due to differences in segmentation resolution, copy number profiles were binned into 30 kb bins and then resegmented. In our previous work[13], we estimated this bin size as appropriate to have a segmentation agreement between copy number calls derived from SNP6 arrays and WGS/whole-exome sequencing. The copy number value of each 30 kb bin was computed by averaging the copy number value of the segments spanning a given bin. For bins spanning multiple copy number segments, this resegmentation may generate artificial segments with a 30 kb length. Segments with a 30 kb length were therefore removed to avoid this artificial oversegmentation, and we then applied a smoothing procedure for merging continuous segments with a ±0.1 difference in copy number.

**TCGA SNP6.** ASCAT-derived TCGA copy number profiles were downloaded from ref. [48].

### Quantification of copy number signatures
We quantified the activities of 17 CIN signatures[13]. We extracted five copy number features (segment size, breakpoint count per 10 MB, changepoint of copy number, breakpoint count per chromosome arm and length of segments with oscillating copy number) from the absolute copy number profiles to then compute a sum-of-posterior probability vector per sample. The sum-of posterior probability vectors were finally used to compute signature activities using the LCD function found in the YAPSA[49] package in R, and the signature definition matrix. To ensure robustness of the signature activities and to enable trust in small signature activities, we applied the signature-specific thresholds derived from the TCGA pan-cancer cohort in our previous work[13].

### Biomarker trial emulations
**Identifying experimental and control arm treatments for phase 3 designs.** We emulated phase 3 biomarker trials to compare the TTFs of patients in the experimental arm (where patients received the chemotherapy of interest) with the TTFs of patients in the control arm (where the patients received an appropriate alternative SoC treatment) (Supplementary Fig. 74).

To construct the experimental arm, we selected the closest treatment line to diagnosis containing the chemotherapy of interest (platin/taxane/anthracycline), either administered in combination with another therapy or as a single agent. For HMF patients with multiple metastatic biopsies, each treatment line was linked to the most recent biopsy, better reflecting the patients' tumor state at the time of treatment. To account for metastasis progression, only treatments given immediately after a biopsy were retained. In ovarian cancer cohorts, we excluded platinum treatment to avoid the effect of the response to this chemotherapy on TTF intervals (see further details in Supplementary Note 2). To validate that TTF differences between predicted resistant and sensitive patients were not influenced by the co-therapy, we also performed this analysis by limiting to single-agent administration where possible.

To construct the control arm, we first excluded all treatment lines that did not contain the chemotherapy of interest. As the filtering at this stage was performed at the level of the treatment line instead of at a patient level, it was possible for patients to have treatment lines in both experimental and control arms. Hence, patients appearing in both arms were removed from the control. Then, the control arm was further filtered to retain only the most common SoCs, which were identified based on frequency. The top five monotherapies formed the control arm. Platinum was excluded from SoCs in relapsed ovarian cancer.

Both arms were further filtered to remove treatment lines with an insufficient therapy exposure: at least three cycles or a treatment length of at least 28 days. Finally, only one treatment line per patient was used in the survival analysis, prioritizing the first viable treatment. Patients without clinical follow-up beyond 1 year were excluded if their last treatment line had a TTF exceeding 730 days.

**Selecting a suitable cohort for phase 2 single-arm designs.** We emulated a phase 2 trial for assessing biomarker performance to predict resistance to first-line platinum in ovarian cancer and to first-line anthracycline in sarcoma. Patients with stage I tumors, non-calculable TTF, fewer than three cycles of treatment or fewer than 28 days were excluded. We included all ovarian and sarcoma TCGA patients who received treatment, alone or in combination, at first line.

Patient counts for single-agent and combination treatments are shown in Supplementary Figs. 75–78.

### Survival analysis
The performance assessment of our signature-based biomarkers to identify patients resistant to platinum-based chemotherapy, taxanes and anthracyclines was carried out using Cox proportional-hazards modeling (function coxph from the survival package in R[50]). We used the function cox.zph to test the proportional-hazards assumption for all Cox proportional-hazards model fits. Further details on survival time period calculations and inverse probability weighting can be found in the Supplementary Methods. Detailed results of all the survival analyses performed in this study can be seen in Supplementary Figs. 5–21. Kaplan–Meier survival curves were generated using the survfit function from the survival package in R[50], and used to represent differences in treatment effectiveness across treatment arms in a univariate mode (Extended Data Figs. 6 and 7).

### Clinical implementation feasibility study
**Concordance between sWGS and TSO500 pairs.** We compared absolute copy number profiles (by using the getDifference function from our CNpare tool in R[51]), signature activities (by computing cosine similarity) and drug resistance predictions (by observing classification concordance) obtained from tissues sequenced using sWGS and the regulatory approved Illumina TSO500 assay. Supplementary Fig. 22 illustrates differences between matched sWGS- and TSO500-derived copy number profiles of all patients, while Fig. 5a shows signature composition concordance between pairs.

**Concordance between tissue and plasma pairs.** We also assessed concordance between tissue and plasma samples collected from the same patient following the same approach. Supplementary Fig. 23 illustrates differences between matched tissue- and plasma-derived copy number profiles of all patients, while Fig. 5c shows signature composition concordance between pairs.

### Reporting summary
Further information on research design is available in the Nature Portfolio Reporting Summary linked to this article.

### Data availability
Raw data from the OV04 study are available via the European Genome Phenome Archive (EGAS50000000992). Access to raw, processed and clinical data from the Hartwig Medical Foundation can be requested at https://www.hartwigmedicalfoundation.nl/en/data/data-access-request/. XML files used to construct the TCGA clinical data can be accessed through the Genomic Data Commons portal, and the procedure for requesting access to controlled genomic data is outlined at https://gdc.cancer.gov/access-data/obtaining-access-controlled-data. All data required for reproducing these analyses are available via Figshare at https://doi.org/10.6084/m9.figshare.27210297 (ref. 41). Information on the data sources used can be found in Supplementary Table 7.

### Code availability
The full analysis code is available via GitHub at https://github.com/macintyrelab/Thompson2025_ChemoResistancePrediction/ and Zenodo at https://doi.org/10.5281/ZENODO.15583246 (ref. 52). Input data needed for running the code are available via Figshare at https://doi.org/10.6084/m9.figshare.27210297 (ref. 41). Information on the R packages used in our analysis[45,49,51,53–98] can be found in Supplementary Table 8.

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

## Acknowledgements

We acknowledge the support of Tailor Bio, Illumina Accelerator, Illumina Cambridge, The University of Cambridge, Cambridge University Hospitals NHS Foundation Trust, Cancer Research UK and the Spanish National Cancer Research Centre (CNIO). We thank all patients who participated in and donated tissue samples to this study. The Addenbrooke's Human Research Tissue Bank is supported by the NIHR Cambridge Biomedical Research Centre. We also thank the OV04 study team for their help with clinical tissue samples. We thank the Cancer Research UK Cambridge Institute Bioinformatics, Compliance and Biobanking, Genomics, Histopathology, Microscopy, Research Instrumentation and Cell Services and Scientific Computing core facilities for their support with various aspects of this study. We thank N. Porta and H. Tovey for valuable discussions regarding trial design. We thank the wider Illumina team including E. Kiernan, P. Pal, V. Liebenberg and B. Kaura for their support in our research collaborations. J.S.T., B.H., M.E.-R., M.T., A.F.-S. and G.M. are hosted by the Centro Nacional de Investigaciones Oncológicas (CNIO), which is supported by the Instituto de Salud Carlos III and recognized as a 'Severo Ochoa' Centre of Excellence (ref. CEX2019-000891-S) by the Spanish Ministry of Science and Innovation (MCIN/AEI/10.13039/501100011033). We acknowledge funding and support from Cancer Research UK, and the Cancer Research UK Cambridge Centre (grant nos. 22905 and 100005 to C.M.S., M.V., M.S., J.P., D.D.S., D.S., A.M.P. and J.D.B.; A25177 to M.A.V.R.; A25117 to K.H.) and the CRUK Innovation Prize PO 1121956 to G.M., A.M.P. and J.D.B. J.S.T., M.E.-R., B.H., M.T., A.F.-S. and G.M. were supported by Spanish Ministry of Science and Innovation grants PID2019-111356RA-I00 and PID2022-137042OB-I00 (MCIN/AEI/10.13039/501100011033) and co-funded by the European Regional Development Fund (ERDF-EU). F.C.M. was funded by the Experimental Medicine Initiative from the University of Cambridge and the Academy of Medical Sciences (grant no. SGL016_1084). I.-G.F. was funded by The Mark Foundation for Cancer Research and the Cancer Research UK Cambridge Centre (grant no. C9685/A25177). This research was also supported by the NIHR Cambridge Biomedical Research Centre (no. BRC-1215-20014). Work in the Cancer Molecular Diagnostics Laboratory/Blood Processing Laboratory was supported by the NIHR Cambridge Biomedical Research Centre, Cancer Research UK Cambridge Centre and the Mark Foundation Institute for Integrated Cancer Medicine. M.Q.-F. is a recipient of grants PMP22/0032 and PI22/00317, awarded by the Instituto de Salud Carlos III. Parts of this work were funded by CRUK core grant C14303/A17197, A19274 (F.M. lab), the UK Research and Innovation's (UKRI) Innovate UK Data to Early Diagnosis challenge and Biomedical catalyst 2021: early and late stage awards and the UKRI Innovate UK Application of whole genome sequencing approaches to cancer award. A.F.-S. and J.S.T. received the support of a fellowship from La Caixa Foundation (ID 100010434; LCF/BQ/DR21/11880009 and LCF/BQ/DI22/11940038, respectively). B.H. was supported by philanthropists via the 'Amigos/as del CNIO' Programme, and also by La Caixa Foundation (ID 100010434; LCF/BQ/PR23/11980033). M.E.-R. received the support of a fellowship from the Spanish Ministry of Science and Innovation (grant no. PRE2020-092155). The views expressed are those of the authors and not necessarily those of Cancer Research UK, the NIHR or the Department of Health and Social Care. The funders had no role in study design, data collection and analysis, decision to publish, or preparation of the manuscript. This publication and the underlying study have been made possible partly based on data that Hartwig Medical Foundation has made available to the study through the Hartwig Medical Database.

## Author contributions

J.S.T., L.M. and B.H. contributed equally to this work. G.M. conceived and designed the study. J.S.T., L.M. and B.H. developed the methodology and software of the study. C.M.S., M.V. and J.D.B. designed and performed in vitro experiments. I.-G.F., J.P., A.M.P. and J.D.B. designed clinical cohorts from the OV04 study. J.S.T., L.M., B.H., C.M.S, M.V., M.E.-R., W.-K.L., D.G.-L., J.H., M.S., K.H., M.J.-L., M.A.V.R., A.R., O.A., J.P., H.D., A.E.C., D.D.S., D.G-S, M.T., A.F.-S., D.S., F.C.M., I.-G.F., G.C.-P, M.Q.-F., F.M., J.Y., J.D.B., A.M.P. and G.M. provided access to data and/or contributed to gathering, processing and curating data. J.S.T., L.M., B.H., A.M.P. and G.M. wrote the paper. J.S.T., L.M., B.H. and G.M. produced and contributed to the visualizations of the study. A.M.P. and G.M supervised the project. All authors had access to all of the data in the study. All authors contributed to the review and the editing of the paper. All authors approved the paper before submission.

## Competing interests

G.M., A.M.P., J.D.B., J.Y. and F.M. are co-founders, directors and shareholders of Tailor Bio Ltd. J.S.T., L.M., D.G.-L., A.R., O.A. and A.E.C. are current or recent employees and shareholders of Tailor Bio Ltd. M.E.-R., W.-K.L., H.D. and D.D.S. are current or previous employees of Tailor Bio Ltd. G.M., F.M., A.M.P. and J.D.B. are inventors on a patent on using copy number signatures to predict response to doxorubicin treatment in ovarian cancer (patent no. PCT/EP2021/065058). G.M., B.H. and F.M. are inventors on a patent on a method for identifying pan-cancer copy number signatures (patent no. PCT/EP2022/077473). G.M., D.G.-L. and D.G.-S. are inventors on a patent on a method for capture bias correction for copy number calling in targeted sequencing data (patent no. PCT/EP2024/070580). The other authors declare no competing interests.

## Additional information

**Extended data** is available for this paper at https://doi.org/10.1038/s41588-025-02233-y.

**Correspondence and requests for materials** should be addressed to Anna M. Piskorz or Geoff Macintyre.

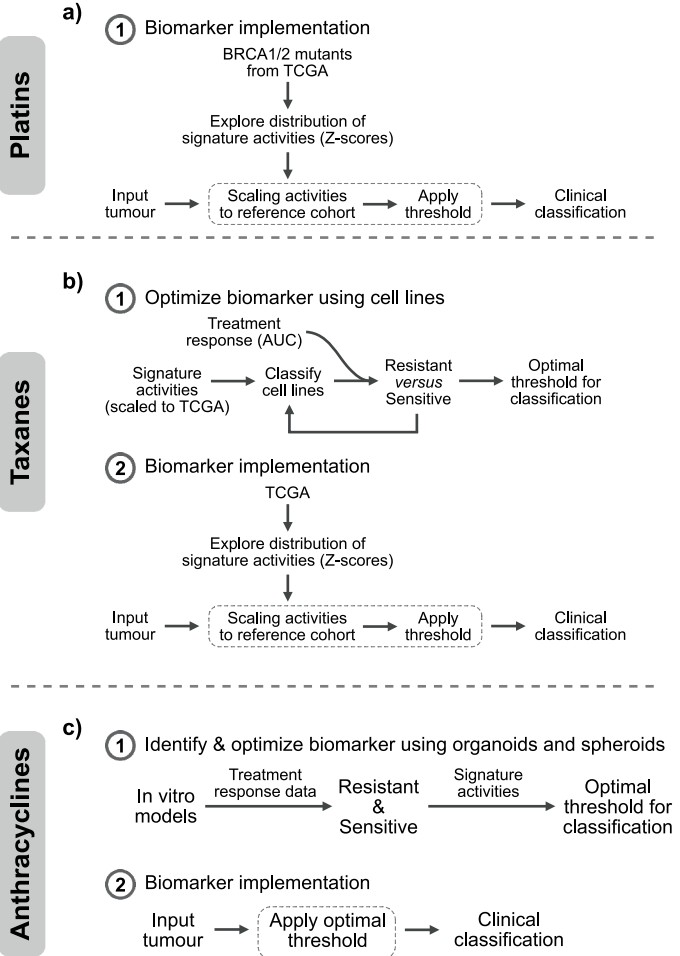

**Biomarker development**

**Extended Data Fig. 1 | Workflow for the development and optimisation of three biomarkers for predicting resistance to platinum, taxanes and anthracyclines. a**) Implementation of a robust signature scaling procedure for pan-cancer application of the platinum biomarker[13]. All *BRCA1/2* mutants in the TCGA dataset (n=375) are used as reference for scaling signature activities of new samples (see Methods). **b**) Optimisation and pan-cancer implementation of a clinical biomarker for predicting taxane resistance. Biomarker optimisation

was initially performed using a collection of 285 cancer cell lines treated with taxanes. To achieve a pan-cancer implementation, all TCGA samples were used as reference for scaling signature activities of new samples (see Methods). **c**) Identification, optimisation and implementation of a biomarker for predicting resistance to anthracyclines. *In vitro* models derived from 23 ovarian cancer patients were treated with doxorubicin and then used to identify and optimise the biomarker (see Methods).

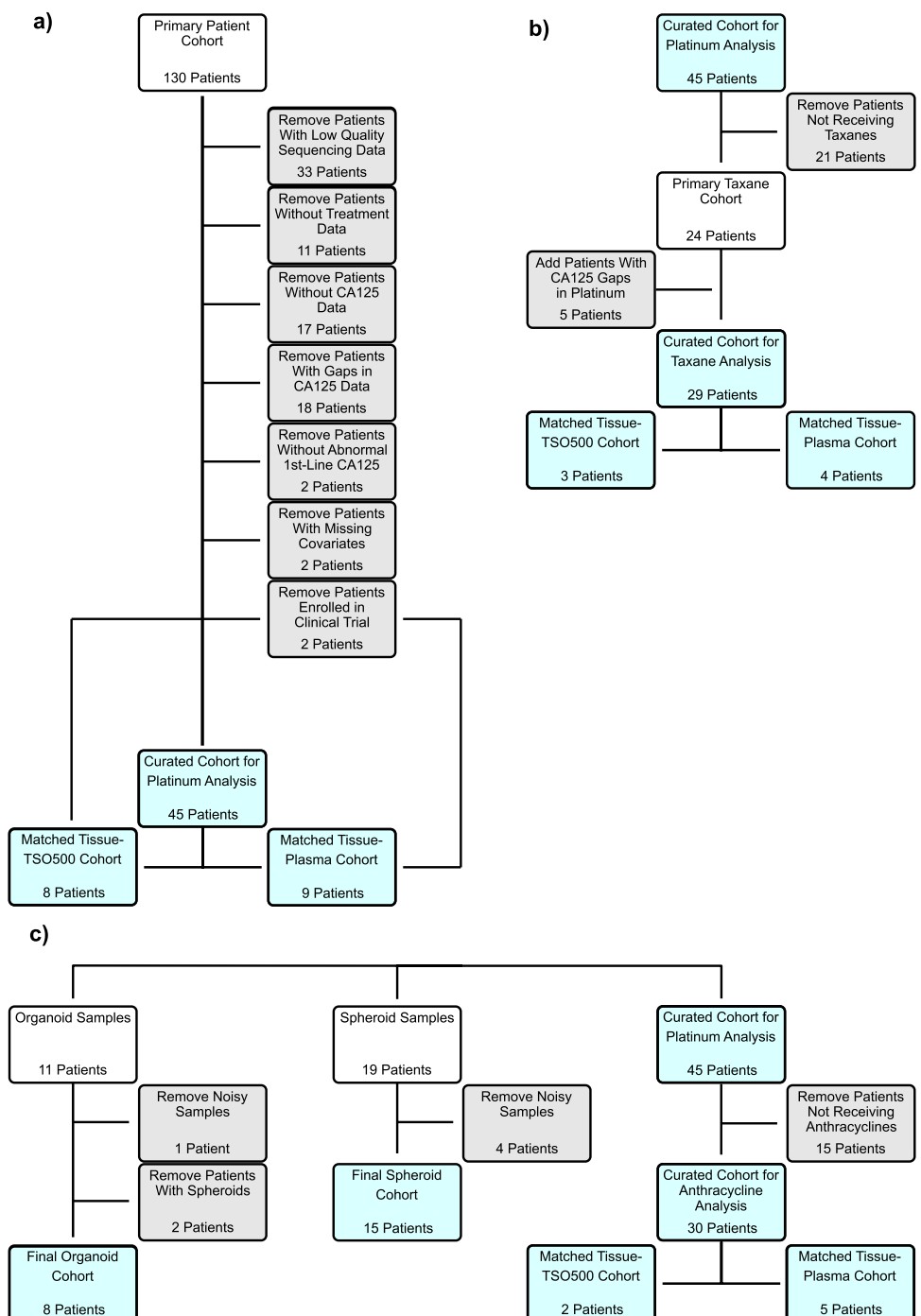

**Extended Data Fig. 2 | Pilot study (OV04) REMARK diagram. a)** REMARK diagram summarising the quality control filtering of samples and patients from the OV04 study obtain a curated cohort for assessing for platinum resistance prediction. **b)** REMARK diagram summarising the quality control filtering of samples and patients from the OV04 study to obtain a curated cohort for assessing taxane resistance prediction. **c)** REMARK diagram detailing the filtering procedure to obtain the organoids and spheroids for anthracycline biomarker discovery, as well as summarising the quality control filtering of samples and patients from the OV04 study to obtain a curated cohort for assessing anthracycline resistance prediction.

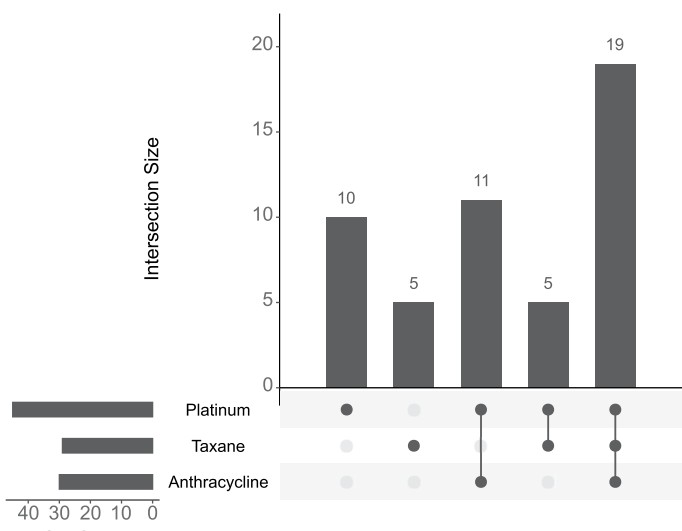

**Extended Data Fig. 3 | OV04 patients included in the single-arm study testing biomarker performance for platinum, taxane and anthracycline.** UpSet plot illustrating the number of ovarian cancer patients included across three distinct single-arm trial designs emulated in the OV04 cohort. The bar chart in the lower left displays the total number of patients tested with each of the three chemotherapy treatments. The main bar chart shows the size of each intersection between sets, as indicated by the matrix of overlapping sets below.

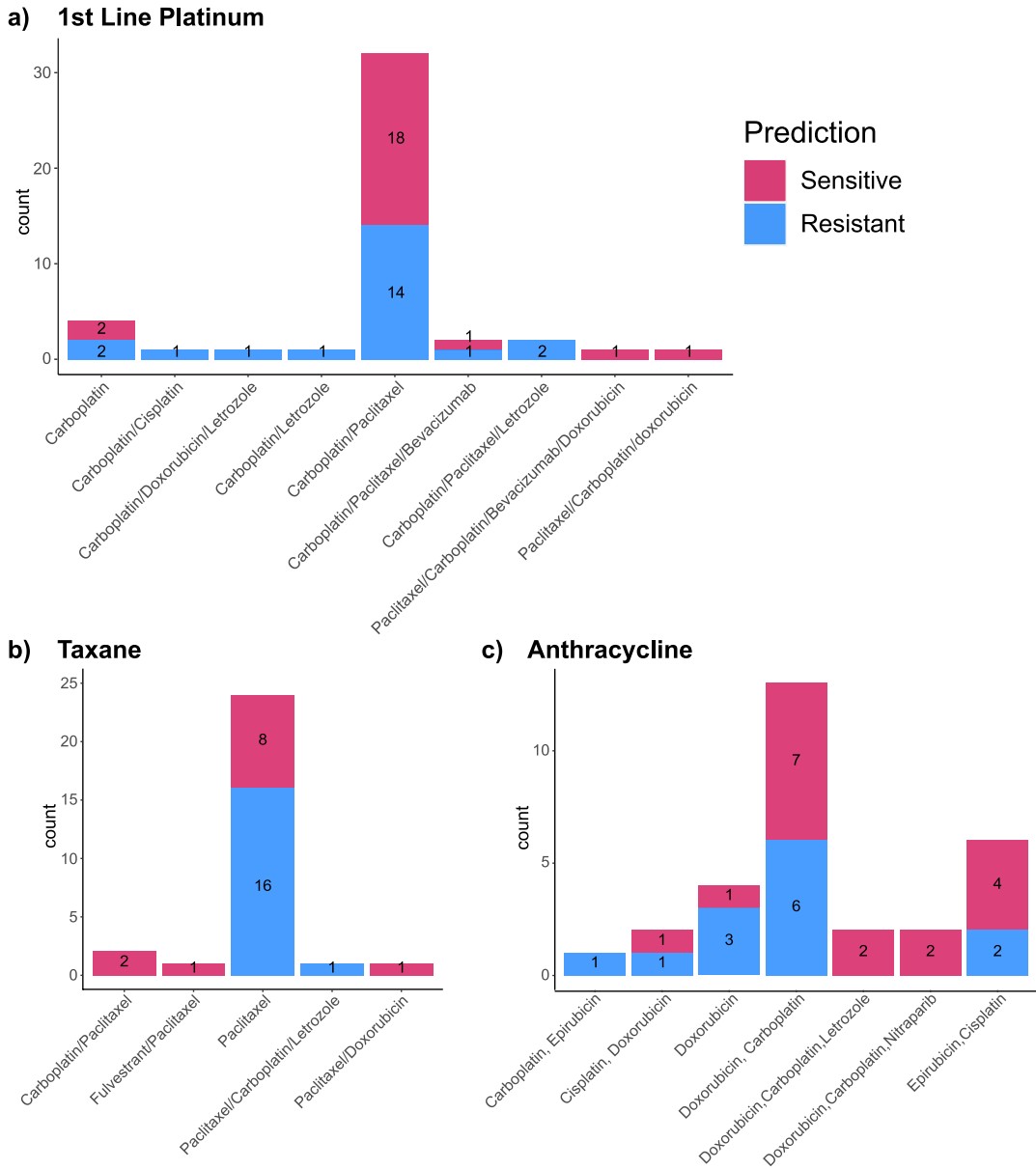

**Extended Data Fig. 4 | Distribution of different treatment combinations with platinum, taxanes and anthracyclines in the OV04 cohort. a)** Number of patients in the OV04 cohort for testing biomarker performance of platinum-based resistance. **b)** Number of patients in the OV04 cohort for testing biomarker performance of taxane resistance. **c)** Number of patients in the OV04 cohort for testing biomarker performance of anthracycline resistance.

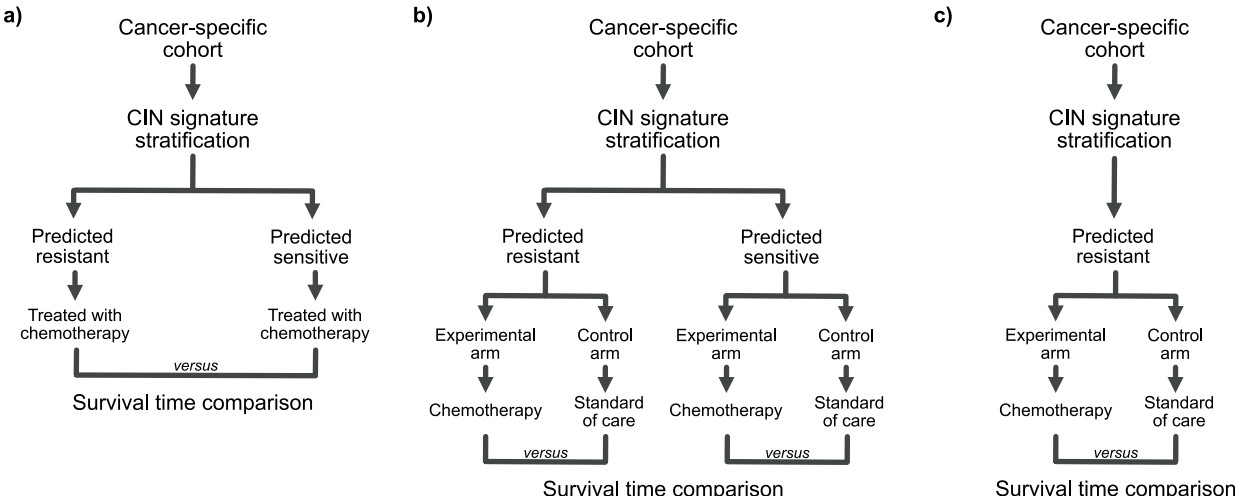

**Extended Data Fig. 5 | Summary of the biomarker trial designs that were used for trial emulation. a**) Schematic illustrating the design of a Phase II single-arm biomarker trial. **b**) Schematic illustrating the design of a Phase III randomised-control biomarker trial. **c**) Schematic illustrating the design of a Phase III randomised-control enrichment biomarker trial.

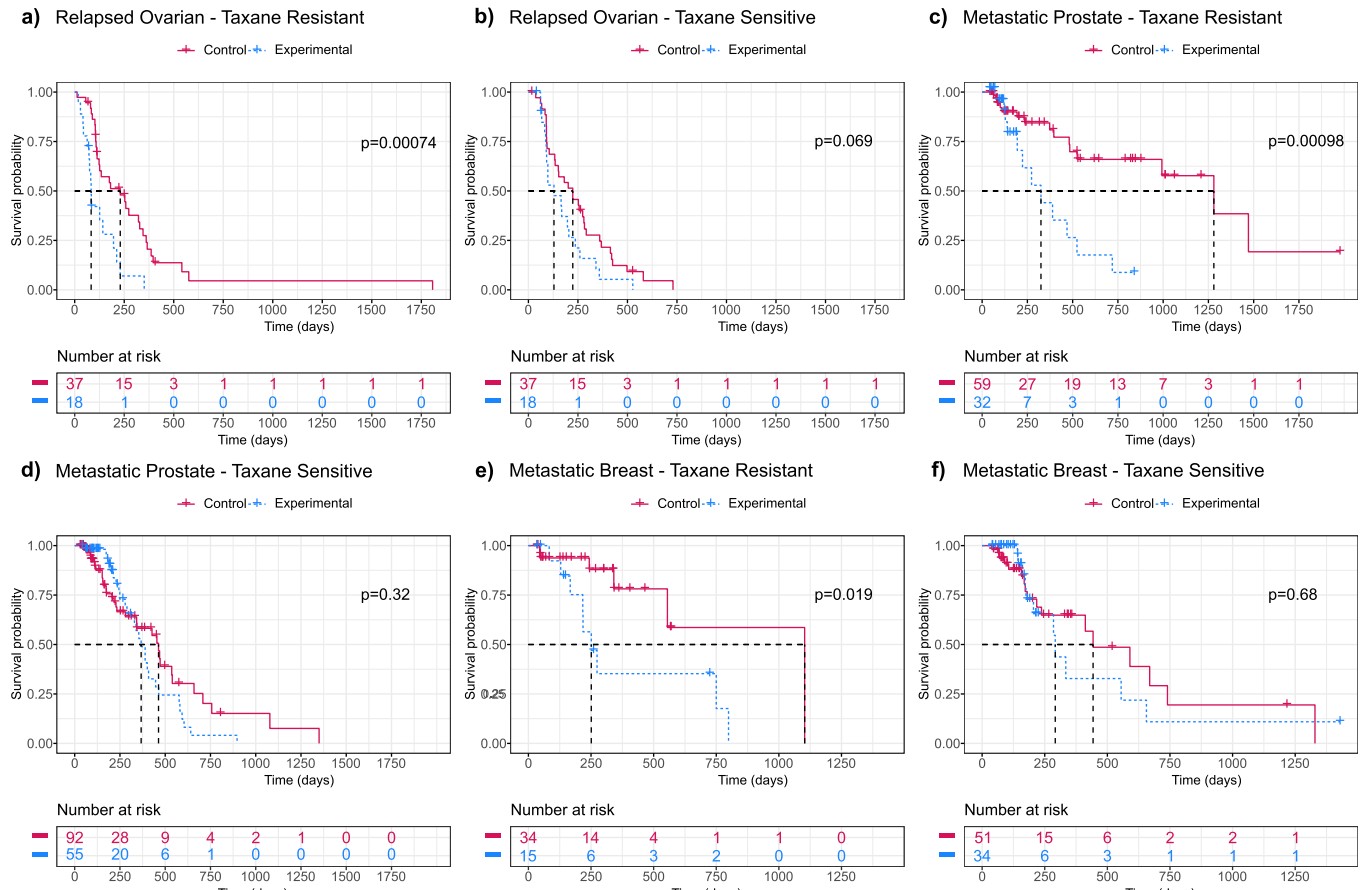

**Extended Data Fig. 6 | Kaplan-Meier survival curves for the taxane biomarker.**
Kaplan-Meier curves comparing time to treatment failure between patients receiving taxane therapy (Experimental arm) and those treated with other standard-of-care therapies (Control arm) across predicted resistant and sensitive subgroups in the emulated randomized controlled trials. Univariate p-values (p) were calculated using the log-rank test. **a**) Predicted resistant patients with relapsed ovarian cancer from the TCGA cohort. **b**) Predicted sensitive patients with relapsed ovarian cancer from the TCGA cohort. **c**) Predicted resistant patients with metastatic prostate cancer from the HMF cohort. **d**) Predicted sensitive patients with metastatic prostate cancer from the HMF cohort. **e**) Predicted sensitive patients with metastatic breast cancer from the HMF cohort. **f**) Predicted sensitive patients with metastatic breast cancer from the HMF cohort.

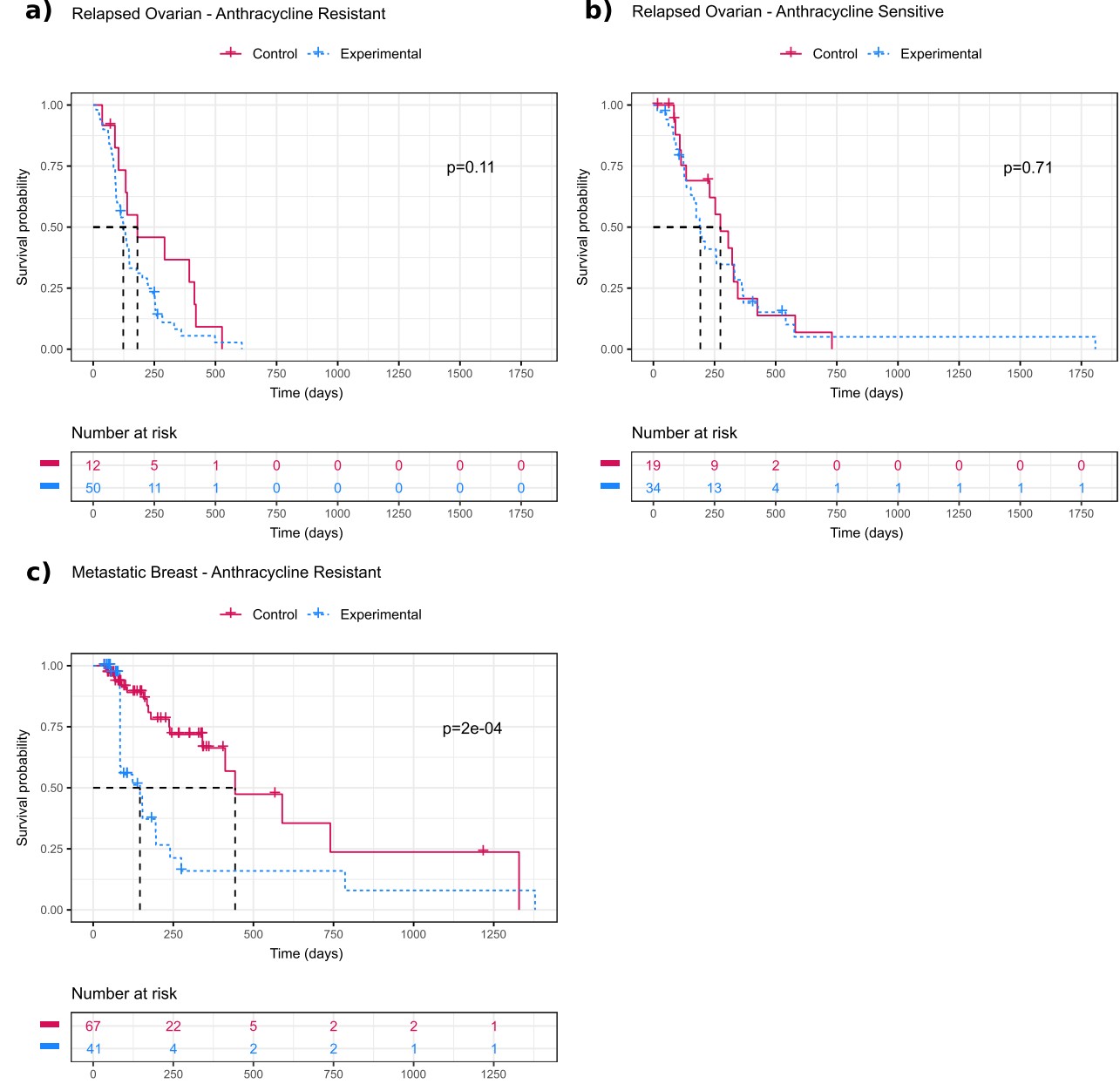

**Extended Data Fig. 7 | Kaplan-Meier survival curves for the anthracycline biomarker.** Kaplan-Meier curves comparing time to treatment failure between patients receiving taxane therapy (Experimental arm) and those treated with other standard-of-care therapies (Control arm) across predicted resistant and sensitive subgroups in the emulated randomized controlled trials. Univariate p-values (p) were calculated using the log-rank test. **a**) Predicted resistant patients with relapsed ovarian cancer from the TCGA cohort. **b**) Predicted sensitive patients with relapsed ovarian cancer from the TCGA cohort. **c**) Predicted resistant patients with metastatic breast cancer from the HMF cohort.

**Extended Data Table 1 | Power calculation results for Phase III survival analysis cohorts**

| Dataset | Cohort | Chemo | Required Resistant Sample Size | Powered Resistant HR | Actual Resistant Sample Size | Powered | Required Sensitive Sample Size | Powered Sensitive HR | Actual Sensitive Sample Size | Powered |
|---------|--------|-------|------|------|------|------|------|------|------|------|
| HMF | Breast | Taxane | 35 | 4.104 | 34 | **TRUE*** | 29 | 0.376 | 51 | **TRUE** |
| HMF | Breast | Anthracycline | 26 | 2.273 | 67 | **TRUE** | 26 | 0.243 | 18 | FALSE |
| HMF | Prostate | Taxane | 34 | 2.796 | 59 | **TRUE** | 24 | 0.506 | 92 | **TRUE** |
| TCGA | OV | Taxane | 14 | 1.990 | 25 | **TRUE** | 12 | 0.540 | 34 | **TRUE** |
| TCGA | OV | Anthracycline | 11 | 1.832 | 35 | **TRUE** | 14 | 0.503 | 50 | **TRUE** |

Summary of cohorts with sufficient power for emulated phase III biomarker trials. Only cohorts with sufficient statistical power to emulate Phase III biomarker trials are shown. Cohorts with available data for power analysis are listed in Supplementary Table 5. Bold text indicates analyses that met power requirements, while asterisks denote cohorts that were near the threshold and included in the analysis. Power was calculated using a one-tailed test (see Power Analysis section in Methods).

**Extended Data Table 2 | Power calculation results for Phase II survival analysis cohorts**

| Dataset | Cohort | Chemotherapy | Required Sample Size | Actual Sample Size | Powered HR | Powered | Used in Phase III |
|---------|--------|--------------|----------------------|--------------------|------------|---------|-------------------|
| HMF | Breast | Taxane | 28 | 52 | 3.442 | **TRUE** | **TRUE** |
| HMF | Breast | Anthracycline | 21 | 60 | 2.678 | **TRUE** | **TRUE** |
| TCGA | OV | Platinum | 32 | 346 | 1.391 | **TRUE** | FALSE |
| TCGA | OV | Taxane | 11 | 108 | 1.718 | **TRUE** | **TRUE** |
| TCGA | OV | Anthracycline | 11 | 81 | 1.858 | **TRUE** | **TRUE** |
| TCGA | SARC | Anthracycline | 15 | 27 | 3.449 | **TRUE** | FALSE |

Summary of cohorts with sufficient power for emulated phase II biomarker trials. Only cohorts with sufficient statistical power to emulate Phase II biomarker trials are shown. Only powered cohorts not already included in Phase III emulations were considered. Cohorts with available data for power analysis are listed in Supplementary Table 5. Bold text indicates cohorts that were powered and also used in Phase III analysis. Power was calculated using a one-tailed test (see Power Analysis section in Methods).

**Extended Data Table 3 | Concordance of signature-based clinical classification across sample types and sequencing technologies**

| | | Platin | | Taxane | | Anthracycline | |
|---|---|---|---|---|---|---|---|
| | | Tissue sWGS | | Tissue sWGS | | Tissue sWGS | |
| | | Resistant | Sensitive | Resistant | Sensitive | Resistant | Sensitive |
| **Plasma sWGS** | **Resistant** | 3 | 0 | 3 | 0 | 1 | 1 |
| | **Sensitive** | 2 | 4 | 0 | 1 | 0 | 3 |
| **Tissue TSO500** | **Resistant** | 3 | 0 | 2 | 0 | 2 | 0 |
| | **Sensitive** | 0 | 5 | 1 | 0 | 0 | 1 |

Contingency tables showing the number of patients predicted as sensitive or resistant by the three signature-based clinical classifiers using sequencing data from tumor and plasma samples obtained with sWGS and/or TSO500.

# Reporting Summary

## Statistics

For all statistical analyses, confirm that the following items are present in the figure legend, table legend, main text, or Methods section.

| n/a | Confirmed | |
|---|---|---|
| ☐ | ☒ | The exact sample size (*n*) for each experimental group/condition, given as a discrete number and unit of measurement |
| ☐ | ☒ | A statement on whether measurements were taken from distinct samples or whether the same sample was measured repeatedly |
| ☐ | ☒ | The statistical test(s) used AND whether they are one- or two-sided<br>*Only common tests should be described solely by name; describe more complex techniques in the Methods section.* |
| ☐ | ☒ | A description of all covariates tested |
| ☐ | ☒ | A description of any assumptions or corrections, such as tests of normality and adjustment for multiple comparisons |
| ☐ | ☒ | A full description of the statistical parameters including central tendency (e.g. means) or other basic estimates (e.g. regression coefficient) AND variation (e.g. standard deviation) or associated estimates of uncertainty (e.g. confidence intervals) |
| ☐ | ☒ | For null hypothesis testing, the test statistic (e.g. *F*, *t*, *r*) with confidence intervals, effect sizes, degrees of freedom and *P* value noted<br>*Give P values as exact values whenever suitable.* |
| ☒ | ☐ | For Bayesian analysis, information on the choice of priors and Markov chain Monte Carlo settings |
| ☒ | ☐ | For hierarchical and complex designs, identification of the appropriate level for tests and full reporting of outcomes |
| ☒ | ☐ | Estimates of effect sizes (e.g. Cohen's *d*, Pearson's *r*), indicating how they were calculated |

*Our web collection on statistics for biologists contains articles on many of the points above.*

## Software and code

Policy information about availability of computer code

| Data collection | All software is currently open for review via the github repository:<br>For most analyses R (v4.2.2) free statistical software was used. The following packages were used:<br>AnnotationDbi 1.56.2<br>Biobase 2.54.0<br>BiocGenerics 0.40.0<br>broom 1.0.3<br>car 3.1.1<br>CNpare 0.99.0<br>cobalt 4.5.5<br>data.table 1.14.8<br>DESeq2 1.34.0<br>DiagrammeR 1.0.11<br>DiagrammeRsvg 0.1<br>doMC 1.3.8<br>doParallel 1.0.17<br>dplyr 1.1.0<br>drc 3.0-1<br>fgsea 1.20.0<br>foreach 1.5.2<br>freshr 1.0.2<br>GenomicDataCommons 1.18.0 |
|---|---|

GenomicRanges 1.46.1
GenomeInfoDb 1.30.1
ggplot2 3.5.1
ggpubr 0.6.0
ggthemes 4.2.4
grid 4.1.2
iterators 1.0.14
IRanges 2.28.0
interactionRCS 0.1.1
knitr 1.42
MatrixGenerics 1.6.0
matrixStats 0.63.0
mclust 6.1.1
org.Hs.eg.db 3.14.0
QDNAseq 1.30.0
QDNAseqmod 1.21.0
readr 2.1.4
readxl 1.4.2
reshape 0.8.9
reshape2 1.4.4
rlang 1.1.3
rstudioapi 0.14
rsvg 2.6.0
S4Vectors 0.32.4
stats4 4.1.2
stringr 1.5.0
SummarizedExperiment 1.24.0
survival 3.4.0
survminer 0.4.9
swimplot 1.2.0
TCGAbiolinks 2.29.6
this.path 1.2.0
UpSetR 1.4.0
WeightIt 1.3.2
xml2 1.3.3
YAPSA 1.32.0

Data analysis

## Generating copy-number profiles
Reads were aligned using bwa-mem (v0.7.17)
Duplicate reads were identified and marked using MarkDuplicates from the GATK toolsuite (v4.1.8.1).
A modified version of the QDNAseq R package (v1.21.0, available here: https://github.com/gmaci/QDNAseqmod) was used to fit absolute copy-number profiles from mapped reads.

## Computing copy number signatures
Pancancer copy number signatures were computed using the YAPSA package from R (v.1.32.0)

## Chemotherapy response predictions
Code available in Github repository

## Gene enrichment analysis
Differential expression analysis was performed using the DESeq2 R package (v1.34.0)

## Survival analysis:
Kaplan Meier analysis: survfit function from survival R package (v3.5-5)
Cox proportional hazard models: coxph function from survival R package (v3.5-5)
Interpretation of interactions between covariates: intEST function from interactionRCS package (v0.1.1)
Inverse Probability Weighting: weightit function from the WeightIt package (v1.3.3)

## Plasma-Tissue concordance
Absolute copy number profiles were compared using the getDifference function from the CNpare R package (https://github.com/macintyrelab/CNpare, accessed 2023-07-20)

## Statistical analyses
Cosine similarity
Linear regression: lm function from base R (v4.2.2)
Wilcoxon rank test: wilcox.test function from stats4 (v4.1.2)

For manuscripts utilizing custom algorithms or software that are central to the research but not yet described in published literature, software must be made available to editors and reviewers. We strongly encourage code deposition in a community repository (e.g. GitHub). See the Nature Portfolio guidelines for submitting code & software for further information.

## Data

Policy information about availability of data

All manuscripts must include a data availability statement. This statement should provide the following information, where applicable:
- Accession codes, unique identifiers, or web links for publicly available datasets
- A description of any restrictions on data availability
- For clinical datasets or third party data, please ensure that the statement adheres to our policy

All data used in this study is described in detail in Supplementary Table 8.
All the required input data to reproduce the code supporting this publication is available at the Github repository accompanying this submission. All the outputs from the code supporting this publication are available without restriction either at the Github repository or the accompanying figshare. Raw data generated in this study is available through the European Genome-phenome Archive (EGA) at the following link: .
OV04 clinical and raw data: Provided by the Brenton lab
OV04 copy number profiles: Generated in this project. Available via the accompanying Github repository.
Cell line, organoid, and spheroid doxorubicin dose-response data: Generated in this project. Available via the Github repository
TCGA copy number profiles: Drews et al 2022. Files hosted on http://github.com/VanLoo-lab/ascat
TCGA clinical data: Genomic Data Commons Data Portal: https://portal.gdc.cancer.gov/
Transcriptomics data for the TCGA ovarian cancer samples: Genomic Data Commons Data Portal: https://portal.gdc.cancer.gov/
TTF data for TCGA ovarian cancer samples: Villalobos, Wang, and Sikic 2018 JCO Clinical Cancer Informatics doi:10.1200/CCI.17.00096
Triple negative status for TCGA breast cancer samples: Lehmann et al 2021, accessed via https://github.com/TransBioInfoLab/TNBC_analysis
Triple negative status for TCGA breast cancer samples: Kalecky et al 2020 BMC Cancer doi:10.1186/s12885-020-6600-6.
Triple negative status for TCGA breast cancer samples: Thennavan et al 2021 Cell Genomics doi:10.1016/j.xgen.2021.100067.
ER/PR/HER2 status for TCGA breast cancer samples: Lehmann et al 2016 PloS One doi:10.1371/journal.pone.0157368
ER/PR/HER2 status for TCGA breast cancer samples: Thennavan et al 2021 Cell Genomics doi:10.1016/j.xgen.2021.100067.
Hallmark pathways for differential expression analysis: Human MSigDB Collections: https://www.gsea-msigdb.org/gsea/msigdb/index.jsp
HMF clinical and raw data: Provided by HMF as part of data access request DR-343
PRISM drug repurposing screen: Corsello et al. 2020. Files accessed via https://github.com/macintyrelab/CINSignatureBiomarkerAnalysis
Copy number profiles of cell lines from the CCLE project: Drews et al 2022. Files hosted on https://github.com/macintyrelab/CINSignatureBiomarkerAnalysis

## Research involving human participants, their data, or biological material

Policy information about studies with human participants or human data. See also policy information about sex, gender (identity/presentation), and sexual orientation and race, ethnicity and racism.

| | |
|---|---|
| Reporting on sex and gender | This study did not do any sex or gender-based analyses. This was because we expect our findings to equally apply to individuals of both sexes/genders. |
| Reporting on race, ethnicity, or other socially relevant groupings | This study does not report on race, ethnic groups or other socially relevant subgroupings. This was because we did not expect these factors to be significant covariates in any analyses. |
| Population characteristics | The only covariate-relevant population characteristics in this study were age at diagnosis, tumour stage and subtypes. These covariates were included as part of multivariate Cox proportional hazard models. Population characteristics for OV04 can be found in Supplementary Table 1 and characteristics for the TCGA and HMF cohorts can be found in Supplementary Table 5. |
| Recruitment | No patients were recruited as part of this project. Patient data used in this project had previously been recruited as part of the ongoing CTCROV04 observational study, and for that purpose patients provided written, informed consent for participation in this study and for the use of their donated tissue for the laboratory studies carried out in this work. All the remaining data was collected from published sources, of which we did not have control over the recruitment process |
| Ethics oversight | Ethics approval for the CTCROV04 study was given by the Institutional Ethics Committee (REC07/Q0106/63, REC08/H0306/61 and REC07/Q0106/63). |

Note that full information on the approval of the study protocol must also be provided in the manuscript.

# Field-specific reporting

Please select the one below that is the best fit for your research. If you are not sure, read the appropriate sections before making your selection.

☒ Life sciences　　　☐ Behavioural & social sciences　　　☐ Ecological, evolutionary & environmental sciences

For a reference copy of the document with all sections, see nature.com/documents/nr-reporting-summary-flat.pdf

# Life sciences study design

All studies must disclose on these points even when the disclosure is negative.

| | |
|---|---|
| Sample size | (Numbers represented below are after filtering data)<br>Ovarian cancer cell lines: 4 cell lines |

OV04 organoids: 8 organoids
OV04 spheroids: 15 spheroids
OV04 tissue: 50 patients
OV04 plasma: 9 patients
TCGA: 7,105 patients
TCGA-OV Platinum Phase II Single-arm study: 352 patients
TCGA-OV Taxane Phase III Randomised Control Study: 112 patients
TCGA-OV Anhtracycline Phase III Randomised Control Study: 115 patients
TCGA-SARC Anhtracycline Phase II Single-arm study: 27 patients
HMF: 7,437 patients
HMF-Prostate Taxane Phase III Randomised Control Study: 238 patients
HMF-Breast Taxane Phase III Randomised Control Study: 134 patients
HMF-Breast Anthracycline Enriched Arm Trial Study: 108 patients

Sample sizes for TCGA and HMF cohorts were checked against power calculations following the approach in 'Methods - Power Analysis'. OV04 and the cancer cell line analyses were considered initial pilot studies and as such were not subject to sample size requirements. However a literature search of published phase II biomarker trials showed that the sample sizes used in the OV04 analyses were within a similar range.

| | |
|---|---|
| Data exclusions | This information is also provided as part of the Methods section of the manuscript. |

Cell lines that showed 100% confluence after 48 hours of being exposed to low doses of doxorubicin were discarded from further analyses due to signal saturation.
Organoids and spheroids were removed from analysis if they showed more then 20% standard deviation across more than 3 dose concentrations when treated with doxorubicin.
The copy number profiles of tissue samples from OV04 were manually curated according to a 3-star rating system, and 1-star samples were removed from downstream analysis.
The copy number profiles of plasma samples from OV04 were manually curated to identify proportions of ctDNA for each sample. Those with low levels of ctDNA were removed from downstream analysis.
For OV04 paclitaxel analysis, only patients that had been treated with post-1st line single-agent paclitaxel were kept. Additionally patients were removed if the single-agent paclitaxel treatment involved <3 or >25 cycles of paclitaxel.
For OV04 doxorubicin analysis, patients who received a combination of doxorubicin and platinum at 1st line were removed. Additionally patients were removed if treatment involved <3 cycles of doxorubicin.
TCGA and HMF patients were also removed if their clinical drug treatment data was too sparse for TTF calculation, or clinical characteristics were not available. In metastatic cases, only treatment lines that occurred immediately following a biopsy were used for the survival analyses.

| | |
|---|---|
| Replication | The platinum response predictor was initially developed using data from the TCGA and the Pan-Cancer Analysis of Whole Genomes (PCAWG) Dataset in Drews et al 2022. In this study, this classifier is reformulated to be applied across multiple tumour types. Then, we tested the pan-cancer applicability in the TCGA-ESCA cohort, and validated in the OV04 and TCGA-OV cohorts via Phase II single-arm trial emulation. The taxane response predictor is optimised using cell lines from the DepMap project, tested in the OV04 cohort using a Phase II single-arm trial emulation, and validated in the TCGA-OV, HMF-Prostate and HMF-Breast cohorts via Phase III randomised controlled study emulation. The doxorubicin response predictor is developed in a combination of cell lines, patient-derived organoids and spheroids, tested in the OV04 cohort using a Phase II single-arm trial emulation, and validated in the TCGA-OV and HMF-Breast cohorts via Phase III randomised controlled study emulation and in the TCGA-SARC cohort via Phase II single-arm trial emulation. |
| Randomization | No randomization was performed - this was a descriptive study, not an experimental study. |
| Blinding | No blinding was undertaken - this was a descriptive study, not an experimental study. |

# Behavioural & social sciences study design

All studies must disclose on these points even when the disclosure is negative.

| | |
|---|---|
| Study description | Briefly describe the study type including whether data are quantitative, qualitative, or mixed-methods (e.g. qualitative cross-sectional, quantitative experimental, mixed-methods case study). |
| Research sample | State the research sample (e.g. Harvard university undergraduates, villagers in rural India) and provide relevant demographic information (e.g. age, sex) and indicate whether the sample is representative. Provide a rationale for the study sample chosen. For studies involving existing datasets, please describe the dataset and source. |
| Sampling strategy | Describe the sampling procedure (e.g. random, snowball, stratified, convenience). Describe the statistical methods that were used to predetermine sample size OR if no sample-size calculation was performed, describe how sample sizes were chosen and provide a rationale for why these sample sizes are sufficient. For qualitative data, please indicate whether data saturation was considered, and what criteria were used to decide that no further sampling was needed. |
| Data collection | Provide details about the data collection procedure, including the instruments or devices used to record the data (e.g. pen and paper, computer, eye tracker, video or audio equipment) whether anyone was present besides the participant(s) and the researcher, and whether the researcher was blind to experimental condition and/or the study hypothesis during data collection. |
| Timing | Indicate the start and stop dates of data collection. If there is a gap between collection periods, state the dates for each sample cohort. |
| Data exclusions | If no data were excluded from the analyses, state so OR if data were excluded, provide the exact number of exclusions and the rationale behind them, indicating whether exclusion criteria were pre-established. |

| Non-participation | State how many participants dropped out/declined participation and the reason(s) given OR provide response rate OR state that no participants dropped out/declined participation. |
| Randomization | If participants were not allocated into experimental groups, state so OR describe how participants were allocated to groups, and if allocation was not random, describe how covariates were controlled. |

# Ecological, evolutionary & environmental sciences study design

All studies must disclose on these points even when the disclosure is negative.

| Study description | Briefly describe the study. For quantitative data include treatment factors and interactions, design structure (e.g. factorial, nested, hierarchical), nature and number of experimental units and replicates. |
| Research sample | Describe the research sample (e.g. a group of tagged Passer domesticus, all Stenocereus thurberi within Organ Pipe Cactus National Monument), and provide a rationale for the sample choice. When relevant, describe the organism taxa, source, sex, age range and any manipulations. State what population the sample is meant to represent when applicable. For studies involving existing datasets, describe the data and its source. |
| Sampling strategy | Note the sampling procedure. Describe the statistical methods that were used to predetermine sample size OR if no sample-size calculation was performed, describe how sample sizes were chosen and provide a rationale for why these sample sizes are sufficient. |
| Data collection | Describe the data collection procedure, including who recorded the data and how. |
| Timing and spatial scale | Indicate the start and stop dates of data collection, noting the frequency and periodicity of sampling and providing a rationale for these choices. If there is a gap between collection periods, state the dates for each sample cohort. Specify the spatial scale from which the data are taken |
| Data exclusions | If no data were excluded from the analyses, state so OR if data were excluded, describe the exclusions and the rationale behind them, indicating whether exclusion criteria were pre-established. |
| Reproducibility | Describe the measures taken to verify the reproducibility of experimental findings. For each experiment, note whether any attempts to repeat the experiment failed OR state that all attempts to repeat the experiment were successful. |
| Randomization | Describe how samples/organisms/participants were allocated into groups. If allocation was not random, describe how covariates were controlled. If this is not relevant to your study, explain why. |
| Blinding | Describe the extent of blinding used during data acquisition and analysis. If blinding was not possible, describe why OR explain why blinding was not relevant to your study. |

Did the study involve field work?  ☐ Yes  ☐ No

## Field work, collection and transport

| Field conditions | Describe the study conditions for field work, providing relevant parameters (e.g. temperature, rainfall). |
| Location | State the location of the sampling or experiment, providing relevant parameters (e.g. latitude and longitude, elevation, water depth). |
| Access & import/export | Describe the efforts you have made to access habitats and to collect and import/export your samples in a responsible manner and in compliance with local, national and international laws, noting any permits that were obtained (give the name of the issuing authority, the date of issue, and any identifying information). |
| Disturbance | Describe any disturbance caused by the study and how it was minimized. |

# Reporting for specific materials, systems and methods

We require information from authors about some types of materials, experimental systems and methods used in many studies. Here, indicate whether each material, system or method listed is relevant to your study. If you are not sure if a list item applies to your research, read the appropriate section before selecting a response.

## Materials & experimental systems

| n/a | Involved in the study |
|---|---|
| ☐ | ☒ Antibodies |
| ☐ | ☒ Eukaryotic cell lines |
| ☒ | ☐ Palaeontology and archaeology |
| ☒ | ☐ Animals and other organisms |
| ☒ | ☐ Clinical data |
| ☒ | ☐ Dual use research of concern |
| ☒ | ☐ Plants |

## Methods

| n/a | Involved in the study |
|---|---|
| ☒ | ☐ ChIP-seq |
| ☒ | ☐ Flow cytometry |
| ☒ | ☐ MRI-based neuroimaging |

## Antibodies

| | |
|---|---|
| Antibodies used | phospho-histo H3 (pHH3; 06-570 Merck) and secondary antibody (Alexa Fluor 555, 1µg/ml; A-21429 Invitrogen) were used as part of the micronuclei counting described in the Methods. |
| Validation | Antibody validation information is available from the manufacturers (antibodies are pre-validated by manufacturer). |

## Eukaryotic cell lines

Policy information about cell lines and Sex and Gender in Research

| | |
|---|---|
| Cell line source(s) | Two of the 4 ovarian cancer cell lines (CIOV2 and CIOV4) were derived in-house from OV04 patients, OVCAR3 was obtained from the American Type Culture Collection, and PEO23 was a kind gift from Langdon lab. |
| Authentication | Cell line identities were confirmed by STR profiling. |
| Mycoplasma contamination | Cells were regularly screened for mycoplasma using a MycoAlert Mycoplasma Detection Kit (Lonza). We confirm that cells tested always negative in the Mycoplasma assays. |
| Commonly misidentified lines (See ICLAC register) | No misidentified lines were used as part of this study |

## Palaeontology and Archaeology

| | |
|---|---|
| Specimen provenance | *Provide provenance information for specimens and describe permits that were obtained for the work (including the name of the issuing authority, the date of issue, and any identifying information). Permits should encompass collection and, where applicable, export.* |
| Specimen deposition | *Indicate where the specimens have been deposited to permit free access by other researchers.* |
| Dating methods | *If new dates are provided, describe how they were obtained (e.g. collection, storage, sample pretreatment and measurement), where they were obtained (i.e. lab name), the calibration program and the protocol for quality assurance OR state that no new dates are provided.* |

☐ Tick this box to confirm that the raw and calibrated dates are available in the paper or in Supplementary Information.

| | |
|---|---|
| Ethics oversight | *Identify the organization(s) that approved or provided guidance on the study protocol, OR state that no ethical approval or guidance was required and explain why not.* |

Note that full information on the approval of the study protocol must also be provided in the manuscript.

## Animals and other research organisms

Policy information about studies involving animals; ARRIVE guidelines recommended for reporting animal research, and Sex and Gender in Research

| | |
|---|---|
| Laboratory animals | *For laboratory animals, report species, strain and age OR state that the study did not involve laboratory animals.* |
| Wild animals | *Provide details on animals observed in or captured in the field; report species and age where possible. Describe how animals were caught and transported and what happened to captive animals after the study (if killed, explain why and describe method; if released, say where and when) OR state that the study did not involve wild animals.* |
| Reporting on sex | *Indicate if findings apply to only one sex; describe whether sex was considered in study design, methods used for assigning sex. Provide data disaggregated for sex where this information has been collected in the source data as appropriate; provide overall* |

*numbers in this Reporting Summary. Please state if this information has not been collected. Report sex-based analyses where performed, justify reasons for lack of sex-based analysis.*

Field-collected samples | *For laboratory work with field-collected samples, describe all relevant parameters such as housing, maintenance, temperature, photoperiod and end-of-experiment protocol OR state that the study did not involve samples collected from the field.*

Ethics oversight | *Identify the organization(s) that approved or provided guidance on the study protocol, OR state that no ethical approval or guidance was required and explain why not.*

Note that full information on the approval of the study protocol must also be provided in the manuscript.

## Clinical data

Policy information about clinical studies

All manuscripts should comply with the ICMJE guidelines for publication of clinical research and a completed CONSORT checklist must be included with all submissions.

Clinical trial registration | *Provide the trial registration number from ClinicalTrials.gov or an equivalent agency.*

Study protocol | *Note where the full trial protocol can be accessed OR if not available, explain why.*

Data collection | *Describe the settings and locales of data collection, noting the time periods of recruitment and data collection.*

Outcomes | *Describe how you pre-defined primary and secondary outcome measures and how you assessed these measures.*

## Dual use research of concern

Policy information about dual use research of concern

### Hazards

Could the accidental, deliberate or reckless misuse of agents or technologies generated in the work, or the application of information presented in the manuscript, pose a threat to:

No | Yes
- ☐ ☐ Public health
- ☐ ☐ National security
- ☐ ☐ Crops and/or livestock
- ☐ ☐ Ecosystems
- ☐ ☐ Any other significant area

### Experiments of concern

Does the work involve any of these experiments of concern:

No | Yes
- ☐ ☐ Demonstrate how to render a vaccine ineffective
- ☐ ☐ Confer resistance to therapeutically useful antibiotics or antiviral agents
- ☐ ☐ Enhance the virulence of a pathogen or render a nonpathogen virulent
- ☐ ☐ Increase transmissibility of a pathogen
- ☐ ☐ Alter the host range of a pathogen
- ☐ ☐ Enable evasion of diagnostic/detection modalities
- ☐ ☐ Enable the weaponization of a biological agent or toxin
- ☐ ☐ Any other potentially harmful combination of experiments and agents

## Plants

| | |
|---|---|
| Seed stocks | *Report on the source of all seed stocks or other plant material used. If applicable, state the seed stock centre and catalogue number. If plant specimens were collected from the field, describe the collection location, date and sampling procedures.* |
| Novel plant genotypes | *Describe the methods by which all novel plant genotypes were produced. This includes those generated by transgenic approaches, gene editing, chemical/radiation-based mutagenesis and hybridization. For transgenic lines, describe the transformation method, the number of independent lines analyzed and the generation upon which experiments were performed. For gene-edited lines, describe the editor used, the endogenous sequence targeted for editing, the targeting guide RNA sequence (if applicable) and how the editor was applied.* |
| Authentication | *Describe any authentication procedures for each seed stock used or novel genotype generated. Describe any experiments used to assess the effect of a mutation and, where applicable, how potential secondary effects (e.g. second site T-DNA insertions, mosiacism, off-target gene editing) were examined.* |

## ChIP-seq

### Data deposition

☐ Confirm that both raw and final processed data have been deposited in a public database such as GEO.

☐ Confirm that you have deposited or provided access to graph files (e.g. BED files) for the called peaks.

| | |
|---|---|
| Data access links<br>*May remain private before publication.* | *For "Initial submission" or "Revised version" documents, provide reviewer access links. For your "Final submission" document, provide a link to the deposited data.* |
| Files in database submission | *Provide a list of all files available in the database submission.* |
| Genome browser session<br>(e.g. UCSC) | *Provide a link to an anonymized genome browser session for "Initial submission" and "Revised version" documents only, to enable peer review. Write "no longer applicable" for "Final submission" documents.* |

### Methodology

| | |
|---|---|
| Replicates | *Describe the experimental replicates, specifying number, type and replicate agreement.* |
| Sequencing depth | *Describe the sequencing depth for each experiment, providing the total number of reads, uniquely mapped reads, length of reads and whether they were paired- or single-end.* |
| Antibodies | *Describe the antibodies used for the ChIP-seq experiments; as applicable, provide supplier name, catalog number, clone name, and lot number.* |
| Peak calling parameters | *Specify the command line program and parameters used for read mapping and peak calling, including the ChIP, control and index files used.* |
| Data quality | *Describe the methods used to ensure data quality in full detail, including how many peaks are at FDR 5% and above 5-fold enrichment.* |
| Software | *Describe the software used to collect and analyze the ChIP-seq data. For custom code that has been deposited into a community repository, provide accession details.* |

## Flow Cytometry

### Plots

Confirm that:

☐ The axis labels state the marker and fluorochrome used (e.g. CD4-FITC).

☐ The axis scales are clearly visible. Include numbers along axes only for bottom left plot of group (a 'group' is an analysis of identical markers).

☐ All plots are contour plots with outliers or pseudocolor plots.

☐ A numerical value for number of cells or percentage (with statistics) is provided.

### Methodology

| | |
|---|---|
| Sample preparation | *Describe the sample preparation, detailing the biological source of the cells and any tissue processing steps used.* |
| Instrument | *Identify the instrument used for data collection, specifying make and model number.* |
| Software | *Describe the software used to collect and analyze the flow cytometry data. For custom code that has been deposited into a community repository, provide accession details.* |

| Cell population abundance | *Describe the abundance of the relevant cell populations within post-sort fractions, providing details on the purity of the samples and how it was determined.* |
|---|---|
| Gating strategy | *Describe the gating strategy used for all relevant experiments, specifying the preliminary FSC/SSC gates of the starting cell population, indicating where boundaries between "positive" and "negative" staining cell populations are defined.* |

☐ Tick this box to confirm that a figure exemplifying the gating strategy is provided in the Supplementary Information.

# Magnetic resonance imaging

## Experimental design

| Design type | *Indicate task or resting state; event-related or block design.* |
|---|---|
| Design specifications | *Specify the number of blocks, trials or experimental units per session and/or subject, and specify the length of each trial or block (if trials are blocked) and interval between trials.* |
| Behavioral performance measures | *State number and/or type of variables recorded (e.g. correct button press, response time) and what statistics were used to establish that the subjects were performing the task as expected (e.g. mean, range, and/or standard deviation across subjects).* |

## Acquisition

| Imaging type(s) | *Specify: functional, structural, diffusion, perfusion.* |
|---|---|
| Field strength | *Specify in Tesla* |
| Sequence & imaging parameters | *Specify the pulse sequence type (gradient echo, spin echo, etc.), imaging type (EPI, spiral, etc.), field of view, matrix size, slice thickness, orientation and TE/TR/flip angle.* |
| Area of acquisition | *State whether a whole brain scan was used OR define the area of acquisition, describing how the region was determined.* |

Diffusion MRI    ☐ Used    ☐ Not used

## Preprocessing

| Preprocessing software | *Provide detail on software version and revision number and on specific parameters (model/functions, brain extraction, segmentation, smoothing kernel size, etc.).* |
|---|---|
| Normalization | *If data were normalized/standardized, describe the approach(es): specify linear or non-linear and define image types used for transformation OR indicate that data were not normalized and explain rationale for lack of normalization.* |
| Normalization template | *Describe the template used for normalization/transformation, specifying subject space or group standardized space (e.g. original Talairach, MNI305, ICBM152) OR indicate that the data were not normalized.* |
| Noise and artifact removal | *Describe your procedure(s) for artifact and structured noise removal, specifying motion parameters, tissue signals and physiological signals (heart rate, respiration).* |
| Volume censoring | *Define your software and/or method and criteria for volume censoring, and state the extent of such censoring.* |

## Statistical modeling & inference

| Model type and settings | *Specify type (mass univariate, multivariate, RSA, predictive, etc.) and describe essential details of the model at the first and second levels (e.g. fixed, random or mixed effects; drift or auto-correlation).* |
|---|---|
| Effect(s) tested | *Define precise effect in terms of the task or stimulus conditions instead of psychological concepts and indicate whether ANOVA or factorial designs were used.* |

Specify type of analysis:    ☐ Whole brain    ☐ ROI-based    ☐ Both

| Statistic type for inference
(See Eklund et al. 2016) | *Specify voxel-wise or cluster-wise and report all relevant parameters for cluster-wise methods.* |
|---|---|
| Correction | *Describe the type of correction and how it is obtained for multiple comparisons (e.g. FWE, FDR, permutation or Monte Carlo).* |

## Models & analysis

| n/a | Involved in the study |
|---|---|
| ☐ | ☐ Functional and/or effective connectivity |
| ☐ | ☐ Graph analysis |
| ☐ | ☐ Multivariate modeling or predictive analysis |

**Functional and/or effective connectivity**

*Report the measures of dependence used and the model details (e.g. Pearson correlation, partial correlation, mutual information).*

**Graph analysis**

*Report the dependent variable and connectivity measure, specifying weighted graph or binarized graph, subject- or group-level, and the global and/or node summaries used (e.g. clustering coefficient, efficiency, etc.).*

**Multivariate modeling and predictive analysis**

*Specify independent variables, features extraction and dimension reduction, model, training and evaluation metrics.*

