## [Peer Review File · Nature Genetics]

Predicting resistance to chemotherapy using chromosomal instability signatures

Corresponding Author: Dr Geoff Macintyre

A version of this paper was originally rejected for publication by Nature Genetics, however that decision was reconsidered after appeal by the authors.

Version 0:

Decision Letter:

14th Sep 2023

Dear Dr Macintyre,

How are you? I hope you're well.

Your Article entitled "Predicting response to cytotoxic chemotherapy" has now been seen by 3 referees, whose comments are attached. In the light of their advice we have decided that we cannot offer to publish your manuscript in Nature Genetics.

While the referees find your work of some interest, they raise concerns about the strength of the novel conclusions that can be drawn at this stage. We feel that these reservations are sufficiently important as to preclude publication of this study in Nature Genetics.

Although we regret that we cannot offer to publish your paper in Nature Genetics given these reviews, I have discussed your manuscript and the reviewers' comments with our colleagues at Nature Communications. They would send the appropriately revised version out for further review if you transfer the revised manuscript to Nature Communications. Should you wish to have your revised paper considered by Nature Communications, please use the link to the Springer Nature manuscript transfer service in the footnote once the revision is ready, and include a point-by-point response to the reviewers' concerns.

Please note that Nature Communications will require all technical and statistical concerns raised by the referees to be addressed in full before sending the revised paper back to the original referees. They would also expect your revision to address the concerns from Reviewer #2 regarding the clinical utility of your proposed biomarkers.

Your handling editor at Nature Communications would be Dr Ilse Valtierra (ilseariadna.valtierragutierrez@nature.com). If there is anything you would like to discuss before transferring the paper and its reviews, please don't hesitate to contact her by e-mail.

To improve the transparency of reporting and the reproducibility of published results, and to verify compliance with all required editorial policies, in order to proceed with peer review at Nature Communications authors must complete the following checklists. Please note that these forms are dynamic 'smart pdfs' and must therefore be downloaded and completed in Adobe Reader, instead of opening them in a web browser. Please provide them with your transferred manuscript.

* Reporting summary: <https://www.nature.com/documents/nr-reporting-summary.pdf>

* Editorial policy checklist: <https://www.nature.com/documents/nr-editorial-policy-checklist.pdf>

* Code and software submission checklist: <https://www.nature.com/documents/nr-software-policy.pdf>

Please note that Nature Communications is a fully open access journal. For information about article processing charges, open access funding, and advice and support from Springer Nature, please consult the Nature Communications Open Access page (www.nature.com/ncomms/open_access/index.html).

I am sorry that we cannot be more positive on this occasion but hope that you will find our referees' comments helpful when preparing your paper for submission elsewhere.

Sincerely,

Safia Danovi
Editor
Nature Genetics

Referee expertise:

Referee #1: cancer genomics

Referee #2: cancer genomics, drug sensitivity

Referee #3: cancer genomics, mutational signatures

Reviewers' Comments:

Reviewer #1:

Remarks to the Author:

This is a very interesting paper developing precision biomarkers based on chromosomal instability (CIN) to predict response to platins, taxanes, and anthracyclines. This work is related to previous work by the team (Nature) where they have reported CIN signatures in taxane treated cancer cell lines. The authors report a new CIN signature related to anthracyclines caused by the formation of micronuclei in the cytoplasm. Anthracycline resistant tumors are tolerant of these micronuclei, while sensitive ones are not. The authors previously developed a copy number based biomarker, that can be determined from shallow WGS, focusing on patients with ovarian cancer. The authors generate new data in cell lines, organoids, and spheroids showing that this biomarker is relatively predictive of response to anthracyclines (Fig 1). Next, the authors show that the marker is predictive of response to platinum, paclitaxel and doxorubicin in patients using a retrospective study from ovarian cancer patients (41) in a clinical trial (OV04) (Fig2). The authors follow this up with a pseudo-RCT with TCGA data. The data from these two figures provides the most convincing evidence that the biomarker is predictive of drug sensitivity (Fig3). Finally the authors end by examining the feasibility of using ctDNA from a liquid biopsy for their metric (Fig4).

Main Method: CIN signature estimation

Estimate genome wide CN features from shallow WGS, for each CN; For each feature, decompose into a mixture of gaussians; for each CN event in each sample compute probability of each component for each feature. Sum the probability across all CN events per sample. Concat all CN sum-of-probs together to get patient x sum of probabilities. These were further featurized via matrix factorization

Major Issues

Overall issues:

Their main method relies on using their hand crafted signatures. However these signatures are an aggregate of ML derived features. It would be interesting to see how the rest of their hand crafted signatures performed, as well as the full set of ML derived features, specifically for this prediction task. The authors also present an investigation as to why the biomarker is potentially prognostic, beyond that it is a marker for micronuclei formation. The paper is plagued by small sample sizes, and I encourage the authors to look for orthogonal validation.

Fig 1. The authors claim "However, 1 of the resistant spheroids also showed 0 signature 6 activity, suggesting 90% specificity for identifying responsive spheroids ($p=0.004$, permutation test). We also used our pan-cancer CIN signatures to identify responsive cases in both organoids and spheroids (CX8, CX9 or CX13 > 0.01 activity) and found an overall sensitivity of 100% and specificity of 79%". However, the sample sizes are quite small, and often skewed to a specific treatment. Because of this, it is difficult to believe the calculations of sensitivity and specificity the authors present. Larger sample sizes would also allow for stronger statistical tests than a permutation test. Finally, the use of RNA-seq from TCGA as a validation of their genomic based marker is handy wavy at best. At the very least it, the author should perform their own RNA-seq and show that the activation/ repression of these pathways is the same as TCGA.

Fig 2. Data in this section is encouraging. It is unclear who was in first line of treatment or second line and if first line could influence results of second line in OV04 data. What is the objective of the trial? What are the criteria for different treatments? What is the end point? It seems to be 41 patients but highly overlapping. Again, small sample sizes and overlapping data.

Fig 3. Sufficient patients? Did they do power analysis? Multiple hypotheses corrections? Results seem barely significant.

Fig 4. While the results are interesting, again the sample sizes are far too small. Having only 9/29 patients have enough ctDNA in plasma for liquid biopsy seems like it's too low to be actually put into practice. They claim "These results suggest that for approximately 17-31% of patients, our cytotoxic chemotherapy response predictor may be applied using a simple blood test", but would need to see some data in a larger sample size to believe this.

Reviewer #2:

Remarks to the Author:

The paper by Thompson et al is fundamentally a biomarker study. They purport to have devised an integrated biomarker that can predict response to platinum chemotherapy, taxanes and anthracyclines. The paper initially focuses on ovarian cancer but then seeks to generalize the findings using TCGA data.

The research objective is clearly a worthy area of study and if successful, this work could have implications for patient care. Although most novel therapies are now targeted or immune based therapies, cytotoxic chemotherapies remain a mainstay of the systemic treatment of many cancer types and the ability to predict which agents would be most effective using pretreatment tumor or blood samples would be an important advance. The work could also have utility for predicting response to antibody drug conjugates. The specific approach used here is the identification of CIN signatures inferred from whole genome sequencing of tumor and cfDNA.

My major concern with the paper is that one cannot infer the clinical utility of their biomarker given the data presented. The authors use the term "response" but do not in fact measure response based on widely accepted criteria such as RECIST. The data are also presented as a Cox proportional hazard models which precludes a patient level view of how well the predictor distinguishes "responders" versus "non-responders". This is critical as similar such biomarkers have not been adopted clinically because they provide insufficient distinction between responding and non-responding populations as to be of clinical utility for medical decision making. In sum, at minimum, the authors need a dataset to validate their approach in which RECIST response data is available and they should present the data using waterfall plots with annotation of biomarker positive and negative patients clearly distinguishable.

Reviewer #3:

Remarks to the Author:

In their study, Thompson, Madrid et al. describe using signatures of chromosomal instability in tumor samples to predict the response to different chemotherapeutic agents in ovarian cancer. While the performance in ovarian cancer was validated for taxanes and doxorubicin in TCGA data, this was only extended to breast cancer. The assay is extended to a limited number of liquid biopsies from the same ovarian cancer donors, highlighting the potential this assay could be used clinically.

Overall, I think this study currently overpromises and underdelivers. The biggest caveat of the described assay and classification is that, in essence, it only evaluates two biological phenotypes that may confer resistance to chemotherapeutic agents: micronuclei-tolerance as a resistance mechanism to doxorubicin and homologous recombination impairment as a resistance mechanism to platins and taxanes. There are many other pathways to therapy resistance, to which a CIN signature-based approach would be blind. Combined with the inability to replicate the signature-based prediction broadly across cancer types, I'm inclined to believe the applicability of this assay is much more limited than promised. Specific comments are detailed below.

Comments

- The prediction of chemotherapy response is very limited to a few cancer types. From the TCGA cohort, only ovarian and breast cancers had a sufficient number to be considered for a randomized controlled study analysis, and validation failed in cervical, head and neck, and uterine cancer cohorts. The classifier works well for ovarian cancers and breast cancers but seems by no means well-tested or successful in a "pan-cancer" setting.
- Based on Extended Data Figures 3 and 4, only TCGA patients with >20 CN events were included in the prediction model. While it makes sense to only classify cases where there is something to classify, it hides the requirement that a tumor needs to exhibit many CNVs for this approach to work. How many tumors with fewer than 20 CNs were excluded and of these, how many were resistant to chemotherapy?
- The study goes back and forth between using the signatures from Macintyre et al. (2018) and Drews et al. (2022), using ovarian CIN signature 6 to predict chemotherapy response in ovarian cancer and pan-cancer signatures (CXs) to predict response in breast cancer. Rather than using two different signature reference sets, I suggest using signatures from one reference set to improve the clarity and coherence of the study, especially since the authors already use the equivalence between ovarian CIN signature 6 and CX8, CX9 and CX13.
- Whenever a copy number signature is introduced, I suggest adding a few words introducing the features and presumed etiology of the signature (as already done for CX8/CX9/CX13). This will help guide readers less familiar with these sets of signatures. A big omission in the text is the link between CX3 and CX5 and homologous recombination impairment, which is only mentioned in the abstract.
- Essentially, estimating the activity of CX2, CX3 and CX5 seems to be a proxy for detecting homologous recombination deficiency, which is commonly used to predict sensitivity to PARP-inhibitors. Do the authors think such methods would work well to predict resistance to the chemotherapeutics included in this study? What is the advantage of the approach described here?

Typos

- In Fig 1d, the corresponding dot in the right plot is missing for the top sample (119120).
- The text mentions classifying 41 patients into platinum-sensitive and platinum-resistant, but Fig 2a shows 39 patients.

****Although we cannot offer to publish your manuscript, I have consulted with my colleague at Nature Communications, and they have agreed to continue the review of your manuscript. To transfer your manuscript please use our manuscript transfer portal. You will not have to re-supply manuscript metadata and files, unless you wish to make modifications. For more information, please see our [manuscript transfer FAQ](http://www.nature.com/authors/author_resources/transfer_manuscripts.html?WT.mc_id=EML_NPG_1511_AUTHORTRANSF&WT.ec_id=AUTHOR) page.**

Version 1:

Decision Letter:

IMPORTANT: Please note the reference number: NG-A63066R-Z Macintyre. This number must be quoted whenever you communicate with us regarding this paper.

15th Nov 2023

Dear Dr Macintyre,

Thank you for asking us to reconsider our decision on your manuscript "Predicting resistance to cytotoxic chemotherapy". I have now discussed your proposed revision plan with my colleagues, and we think that you have some valid points. We therefore invite you to revise your manuscript along the lines that you propose.

When preparing a revision, please ensure that it fully complies with our editorial requirements for format and style; details can be found in the Guide to Authors on our website (<http://www.nature.com/ng/>).

Please be sure that your manuscript is accompanied by a separate letter detailing the changes you have made and your response to the points raised. At this stage we will need you to upload:

1) a copy of the manuscript in MS Word .docx format.

2) The Editorial Policy Checklist:

<https://www.nature.com/documents/nr-editorial-policy-checklist.pdf>

3) The Reporting Summary:

(Here you can read about the role of the Reporting Summary in reproducible science:

<https://www.nature.com/news/announcement-towards-greater-reproducibility-for-life-sciences-research-in-nature-1.22062>)

Please use the link below to be taken directly to the site and view and revise your manuscript:

Link Redacted

With kind wishes,

Safia Danovi
Editor
Nature Genetics

Version 2:

Decision Letter:

*)\$)'

!

!- 2

"

/ 10

1 2 0

' " 9' 22

4 ! 2 2 4

"3 2

2 2 4

"

6 ! 4 !

2! 2 2

8 2 "3

;

"

3 2 1 " 2 4

2 22 " (3 " 1

2 2 " 2

3 22 7 " 4 2 2 "

3 2

?&=- E N 2 ! 7 7 ! 22 "-
4 2 ! 2 " 2 "6 4

?&=- 2 2 2

Q 10 @ " " 2 @ @ @ C @ 1" 2 d Q@L"

?&=- 8 (22 @ " " 2 @ 2 @7 7
22 " : ! ! = 2 4

4 7 "

2 Q 10 @ " " 2 @ 7 @ 7 @ 7 d
"Q@L"

4 2 2

% 4

6 R % 4 2 2 7 ! 2 4
2 ! " "-

3 2 4 "- 2 !
4 "

2 8 4 "

8 22 2 > 5 6 4 (2 : 6(± 4 B !
- :B - = " / "B -
22 2 4 " > 2 (5 2 2 4

B - 2 Q 10 @ " " 2 @ d " 2 @ Q@L"

3 4 2 4 4"

(!

Safia Danovi, PhD
Senior Editor, Nature Genetics
ORCID: 0009-0007-7822-5479

Reviewers' Comments:

Reviewer #1 (Remarks to the Author):

The authors have addressed most concerns.

Reviewer #2 (Remarks to the Author):

The conclusion of the revised manuscript by Thompson et al is that CIN signatures can be used to predict resistance to various chemotherapies across multiple different types of cancer. The study was a retrospective analysis that sought to identify the potential clinical utility of CIN signatures as a predictive biomarker through pseudo-randomization to a single chemotherapy treatment arm or alternative standard-of-care arm.

Overall, the authors were responsive to many of the prior concerns raised by the reviewers during the initial review and the paper is now improved. My primary concern remains unresolved however. Specifically, the authors have not been able to provide evidence that the biomarker has clinical utility (and thus will be of interest to the clinical oncology community) through a valid patient level analysis. They state that RECIST response data was unavailable for the patients analyzed and therefore they generated waterfall plots where the y axis is baseline change compared to the average control arm (I do not see how a reader can know what this really means). In my opinion, if they are going to use a time to event metric to study time to drug resistance, they need to show this patient level data as a swimmers plot, Kaplan-Meier curves or both. If a swimmers plot is used, it should include both time on therapy and time to next therapy on the plot as chemotherapy is often held not for progression but for toxicity. Based on the data shown, such plots would likely show that many of the patients with "resistant signatures" had some of the longer times on therapy. If so, I am doubtful that such a signature would be used by clinicians for treatment decisions lacking prospective randomized data. I therefore would suggest a repeat review following inclusions of such plots and not simply the hazard ratios provided in the main figures.

Reviewer #3 (Remarks to the Author):

In their revised manuscript, Thompson, Madrid, Hernando et al. present their work on predicting resistance to cytotoxic chemotherapy based on signatures of chromosomal instability. I commend the authors on the scope of the revisions and think the inclusion of the Hartwig data, the more careful phrasing and scope of the manuscript, and the enhanced focus on clinical translation has substantially improved the study. The authors have adequately addressed my previous concerns.

Small typo on l.24, it should be "many patients can..."

I congratulate the authors on the revised study, and have no further comments.
Tim Coorens

Reviewer #3 (Remarks on code availability):

Code looks to be all present and is organized well, so it's easy to follow for the reader/reviewer.

Reviewer #4 (Remarks to the Author):

This study introduces chromosomal instability (CIN) biomarkers as a potential tool to predict resistance to key chemotherapy regimens, including platinum-, taxane-, and anthracycline-based treatments. Using retrospective analyses of real-world cohorts and emulated clinical trials, the authors demonstrate that CIN signatures effectively stratify patients likely to experience treatment failure across multiple cancer types. They also highlight the feasibility of integrating these biomarkers into clinical practice using genomic assays from tissue or liquid biopsies. The work offers a promising step toward personalized chemotherapy. However, I have some statistical concerns about the RCT emulations in the paper.

Detailed comments:

For the RCT emulations, could the authors provide more methodological details? The paper currently states: "Within these sensitive or resistant groups, patients were then retrospectively assigned to the experimental arm (treated with the chemotherapy of interest) or to the control arm (treated with an appropriate standard-of-care therapy)." However, in

observational data, treatment selection bias is a significant concern. To emulate randomization effectively, it is essential to address this bias and recover the randomization process. Methods such as inverse probability weighting or propensity score matching are commonly employed to balance treatment arms and mimic randomization. Patient characteristics and clinical factors that may influence treatment assignment in real-world data should be incorporated in the propensity scores to fully control for selection bias. Could the authors conduct such analyses to validate the findings presented?

At the top of Page 9, it is stated: "Statistical analysis of the cohort was performed using Cox proportional-hazards models stratifying patients by age at diagnosis (<65 or ≥65 years) and tumour stage (III, IV)." Typically, stratified analyses involve fitting separate Cox models for different age groups and tumor stages. However, I did not see multiple hazard ratios (HRs) reported for different age groups or tumor stages, and in Figure 2, the HR is also reported for the tumor stage. Could the authors clarify whether the Cox model was adjusted for age and tumor stage or if the baseline hazard function was allowed to differ by age and tumor stage? Similar phrasing appears in several parts of the paper. Please clarify and provide the rationale for the model selection.

Reviewer #4 (Remarks on code availability):

The code and data were made available online, accompanied by a README file. While I successfully executed several R scripts, this required installing additional packages that were not listed in the code. Notably, packages such as ggpubr, DiagrammeR, and data.table were missing. I recommend that the authors conduct a thorough review of the code dependencies and explicitly include all required packages to ensure seamless execution.

Version 3:

Decision Letter:

Our ref: NG-A63066R2

26th Feb 2025

Dear Dr Macintyre,

Thank you for submitting your revised manuscript "Predicting resistance to cytotoxic chemotherapy" (NG-A63066R2). It has now been seen by the original referees and their comments are below. The reviewers find that the paper has improved in revision, and therefore we'll be happy in principle to publish it in Nature Genetics, pending minor revisions to satisfy the referees' final requests and to comply with our editorial and formatting guidelines.

Sincerely,

Safia Danovi, PhD
Senior Editor, Nature Genetics
ORCID: 0009-0007-7822-5479

Reviewer #2 (Remarks to the Author):

Overall, I continue to find this manuscript to be interesting but remain concerned about the clinical utility of the CIN signature as a predictive biomarker of drug response. The authors were responsive to the prior review by including swimmers plots and KM curves where possible given the data available. However, the swimmers' plots included do not show any difference between the biomarker sensitive and resistant patients. Notably, in the Figure legend for Supplemental Figure 54 they state, "No statistical difference in treatment length was observed between resistant and sensitive patients". This would suggest to me that the data does not support the biomarker as clinically useful as the differences are unlikely to prompt clinicians to alter their treatment decisions. I agree with the plan by the editorial team to pursue further statistical review and would defer to such statistical reviewers in regards to analyzing the methodology employed and the statistical significance of the findings. In regards to their clinical significance, I suspect the impact will be low.

Reviewer #4 (Remarks to the Author):

The authors have addressed all my comments. I have no further comments.

Reviewer 1

Remarks to the Author:

This is a very interesting paper developing precision biomarkers based on chromosomal instability (CIN) to predict response to platins, taxanes, and anthracyclines. This work is related to previous work by the team (Nature) where they have reported CIN signatures in taxane treated cancer cell lines. The authors report a new CIN signature related to anthracyclines caused by the formation of micronuclei in the cytoplasm. Anthracycline resistant tumors are tolerant of these micronuclei, while sensitive ones are not. The authors previously developed a copy number based biomarker, that can be determined from shallow WGS, focusing on patients with ovarian cancer. The authors generate new data in cell lines, organoids, and spheroids showing that this biomarker is relatively predictive of response to anthracyclines (Fig 1). Next, the authors show that the marker is predictive of response to platinum, paclitaxel and doxorubicin in patients using a retrospective study from ovarian cancer patients (41) in a clinical trial (OV04) (Fig2). The authors follow this up with a pseudo-RCT with TCGA data. The data from these two figures provides the most convincing evidence that the biomarker is predictive of drug sensitivity (Fig3). Finally the authors end by examining the feasibility of using ctDNA from a liquid biopsy for their metric (Fig4).

Main Method: CIN signature estimation

Estimate genome wide CN features from shallow WGS, for each CN; For each feature, decompose into a mixture of gaussians; for each CN event in each sample compute probability of each component for each feature. Sum the probability across all CN events per sample. Concat all CN sum-of-probs together to get patient x sum of probabilities. These were further featurized via matrix factorization

Major Issues

Overall issues:

Their main method relies on using their hand crafted signatures. However these signatures are an aggregate of ML derived features. It would be interesting to see how the rest of their hand crafted signatures performed, as well as the full set of ML derived features, specifically for this prediction task.

This is a good suggestion by the reviewer. In the revised manuscript we tested all CIN signatures individually, and the copy number feature components prior to signature decomposition, for their capacity to predict resistance to platinum/taxane/anthracycline treatment using data from our ovarian cancer pilot study (see **Supplementary Table 6, Methods lines 904-916, reproduced below for convenience**). In all cases the remaining CIN signatures and components did not show predictive capacity. For the signatures, we hypothesised this is likely due to two key factors: the defective biology read out by the signature does not confer resistance/sensitivity to the chemotherapies studied; and/or, the signature requires optimisation before a predictive signal can be seen (either through scaling

or use in combination with another signature). For the components, this suggests general measures of CIN are not strong predictors of resistance.

This insight reinforces the approach we have adopted in the manuscript, whereby we prioritise CIN signature biomarkers based on a clear link between the putative underlying aetiology of the signature and the mechanism of action of the drug; and we optimise each biomarker such that its predictive signal can be maximised and generalised. Taking this into account, we decided to reformulate the presentation of our biomarkers in the manuscript and perform additional optimisation (see revised section “**CIN signatures as biomarkers of chemotherapy resistance**” **main manuscript line 101**). We therefore thank the reviewer as we believe this has substantially improved the robustness and presentation of our approach.

Description of analysis reproduced from **Methods line 904-916**:

&RS\ QXP EHUMJQDMUH DFVWVWV

We tested the individual performance capacity of each of the 17 CIN signatures⁹ by applying a univariate cox proportional hazards model to predict survival periods based on signature activities. No CIN signature activity showed significant association with PFS after multiple testing correction in the OV04 cohort for any of the three chemotherapies tested (**Supplementary Table 6**).

&RS\ QXP EHUIHDMUH FRP SRQHQW

We also tested the performance capacity of the different copy number feature components defining the encoding space of our compendium signatures⁹ by applying a univariate cox proportional hazards model to predict survival periods based on the sum-of-posteriors per feature component. No copy number feature components showed significant association with PFS after multiple testing correction in the OV04 cohort for any of the three chemotherapies tested (**Supplementary Table 6**).

The authors also present an investigation as to why the biomarker is potentially prognostic, beyond that it is a marker for micronuclei formation. The paper is plagued by small sample sizes, and I encourage the authors to look for orthogonal validation.

We thank the reviewer for encouraging us to extend our sample numbers and seek orthogonal validation. In the revised manuscript we have included new datasets and extended our patient cohorts:

- Our total patient cohort has increased from n=442 to **n=740**.
- We sequenced a further 83 patient samples and curated their clinical data, which after quality control filtering, increased our cohort by 9 patients to a total of **n=50**.
- We include a new biomarker optimisation procedure which uses cell line drug response data for **n=297** cells

We have consolidated our organoid and spheroid data for anthracycline biomarker optimisation, totalling **n=23**

- We extended our phase III randomised-control trial emulation to include the Hartwig Medical Foundation metastasis cohort, increasing the retrospective patient cohort by **n=282**.
- We profiled **n=8** patients from our ovarian study with the Illumina TSO500 panel to assess feasibility of implementation of our predictions on a targeted panel.

We have also altered the analysis structure of the manuscript such that preclinical data is used for optimisation, the ovarian cohort is used as a pilot study to assess feasibility of using real-world data to emulate biomarker trials, and the larger TCGA and Hartwig cohorts are used as orthogonal validation via emulation of phase III RCTs.

Fig 1. The authors claim “However, 1 of the resistant spheroids also showed 0 signature 6 activity, suggesting 90% specificity for identifying responsive spheroids ($p=0.004$, permutation test). We also used our pan-cancer CIN signatures to identify responsive cases in both organoids and spheroids ($CX8, CX9$ or $CX13 > 0.01$ activity) and found an overall sensitivity of 100% and specificity of 79%”. However, the sample sizes are quite small, and often skewed to a specific treatment. Because of this, it is difficult to believe the calculations of sensitivity and specificity the authors present. Larger sample sizes would also allow for stronger statistical tests than a permutation test.

Due to the change in analysis structure, where we now use the organoid and spheroid data to choose the optimal pan-cancer threshold for the anthracycline predictor, we no longer report on sensitivity and specificity, or perform permutation testing using these data, rather, we validate performance in our independent patient cohorts via biomarker trial emulations.

Finally, the use of RNA-seq from TCGA as a validation of their genomic based marker is handy wavy at best. At the very least it, the author should perform their own RNA-seq and show that that the activation/ repression of these pathways is the same as TCGA.

Unfortunately it was not possible to generate our own RNAseq data on these samples due to technical limitations: for the organoids, we were not powered to detect changes between sensitive and resistant groups as there were only 2 sensitive organoids. For the spheroids, they only survived as a short term culture and it was not possible to retrospectively perform RNA profiling.

Despite not having these data, the RNAseq analysis performed on 317 ovarian cancer patient tumours (186 predicted resistant vs 131 sensitive) was sufficiently powered to explore expression program changes linked to non-canonical NF- κ B signalling. However, we realised that our description of this analysis was poor and we omitted any details of *how* the enriched gene lists explicitly relate to a switch from cGAS-STING to non-canonical NF- κ B signalling in the previous manuscript draft. We have provided a further explanation of this analysis in **Supplementary Note 1, reproduced here for convenience**. It is also important to note that the RNAseq data is only one piece of evidence that supports the role of micronuclei tolerance in anthracycline resistance. We also observed a suppressed rate of micronuclei formation in

resistant cell lines and a link between signatures of DNA amplification and doxorubicin resistance *in vitro* using patient derived organoids and spheroids.

Reproduced Supplementary Note 1

Supplementary Note 1 - Tolerance to micronuclei and anthracycline resistance

Anthracyclines can cause DNA damage resulting in extrachromosomal DNA (ecDNA) encapsulated in micronuclei¹, and thus tumours resistant to anthracyclines may tolerate the ongoing formation of micronuclei. To identify such resistant tumours, we rely on three specific CIN signatures associated with high-level copy number changes of small segments (CX8, CX9 and CX13).

Existing studies show that low doses of the anthracycline doxorubicin can induce micronuclei formation in cancer cell lines². Therefore, to test if the presence of these amplification-related signatures was associated with any modulation in micronuclei formation rates, we treated a panel of four ovarian cancer cell lines with low dose doxorubicin and observed micronuclei induction rates using fluorescent imaging (**Supplementary Note Figure 1a**). We performed shallow whole genome sequencing on the cell lines prior to treatment, computed CIN signatures and also estimated the expected number of induced micronuclei using a model of micronuclei induction and inheritance (see **Methods**). Cell lines with high activity of amplification-related signatures (CX8, CX9 and CX13) showed fewer than expected micronuclei, whereas the cell line with no activity of amplification-related CIN signatures showed the expected number of micronuclei (**Supplementary Note Figure 1b**). This suggests cells with amplification-related signatures have a reduction in DNA damage and potential genome stabilisation.

We then aimed to construct and optimise a signature-based biomarker for predicting resistance to anthracycline treatment *in vitro*. We treated a cohort of 23 ovarian cancer patient-derived models (8 organoids and 15 spheroids) with the anthracycline doxorubicin, measured response via IC50 to then classify models as resistant or sensitive based on the expected number of sensitive cases (see **Methods**), and explored activity of amplification-related signatures computed from shallow whole-genome sequencing prior to treatment. We then used a grid search to explore a range of activity values for the three amplification-related signatures and determine the optimal activity threshold for maximising specificity. Maximising specificity facilitates identifying patients resistant to anthracyclines without preventing those who are sensitive from receiving the therapy. In this cohort, all sensitive models showed an activity of CX8, CX9 and CX13 lower than 0.01 (100% specificity), and 3 of the 14 resistant models showed activity lower than 0.01 (82% sensitivity). Therefore, thresholds of $CX8 > 0.01$, $CX9 > 0.01$ and $CX13 > 0.01$ were selected as optimal for identifying resistant models (**Figure 1f**).

Gidd`Ya YbhUfm' BchY' :][ifY' %W'

b) Activity of amplification-related signatures in ovarian cell lines

c) TCGA-OV

Supplementary Note Figure 1. Predicting doxorubicin response using copy number signatures linked to extrachromosomal DNA. **a)** Overview of experimental design for exploring the presence of amplification-related CIN signatures (CX8, CX9 and CX13) and micronuclei induction. **b)** Evaluation of micronuclei induction under doxorubicin treatment. Boxplots show the observed frequency of cells with micronuclei (y-axis) in the presence of 0 and 0.025 μM of doxorubicin. Red line indicates the expected frequency of cells with micronuclei according to growth rates and micronuclei persistence across cell divisions. **c)** Gene set enrichment analysis results showing HALLMARK gene sets that are highly enriched in TCGA-OV tumours predicted as resistant compared with those predicted as sensitive to doxorubicin. Only significant results after FDR correction are shown (q-value < 0.05).

Fig 2. Data in this section is encouraging. It is unclear who was in first line of treatment or second line...

We agree that this was unclear in the previous version of the manuscript and thank the reviewer for pointing it out. In the revised manuscript we have clearly articulated the breakdown of which patients were included in each analysis, the eligibility criteria, the endpoint used for survival analyses, and which treatment line was used.

Patient level filtering for each analysis can be found in **Extended Data Figure 2**.

Eligibility criteria can be found in **Figure 2, Methods lines 69-127**.

The primary end-point of each trial emulation is now clearly articulated in the main text.

Details of chemotherapy administration as single-agent or in combination with other therapies can be found in **Extended Data Figure 4**

Treatment lines are summarised in **Supplementary Note Figure 2** and individual patient clinical history plots showing all lines of treatments, with relevant lines annotated, can be found in **Supplementary Figures 1-50 (an example is reproduced below for convenience)**.

Supplementary Figure 2. Clinical history plot for OV04 patient 14. Blood serum CA125 levels are shown over time, with the horizontal bars denoting cycles of treatment. The horizontal bar at 35 units/ml denotes the threshold between 'normal' and 'abnormal' CA125 readings. In cases where multiple treatments are given on the same day, the treatment date is shifted slightly to show all treatments. Red shapes indicate progression points annotated for the different chemotherapies of interest.

...and if first line could influence results of second line in OV04 data.

This is a very insightful comment from the reviewer and something that has substantially improved the way in which we perform our Cox model analysis for assessing taxane and anthracycline resistance at second-line and beyond. We have included a new

Supplementary Note 2 (reproduced below for convenience), which details how we have controlled for the effects of first-line platinum treatment interval on subsequent treatment PFS/TTF intervals.

Supplementary Note 2 - Accounting for platinum effects when predicting treatment resistance in relapsed ovarian cancer

Response to second-line treatment in relapsed ovarian cancer has been shown to be heavily influenced by the first-line platinum treatment-free interval⁶. As first-line treatment in our ovarian cohorts is platinum-based, the effect of this therapy can heavily influence PFS or TTF intervals for subsequent taxane or doxorubicin treatment, whether or not these were given in combination with platinum. At second-line, the effect of platinum-free treatment is likely strong enough to warrant including an interaction between first-line platinum PFS/TTF and the main predictor/experimental treatment. For subsequent lines, the patient is more likely to have become platinum resistant and thus the effect will be diminished and any effect of platinum sensitivity may not need to be accounted for using an interaction term. Avoiding the use of an interaction term is preferable to avoid any multiplicative effect of the resulting HR. When an interaction is to be included in the model, the HR can change over the different levels of first-line PFS/TTF. This can have a multiplicative effect on the final HR reported, potentially distorting interpretation. Therefore, to account for this and provide a more reasonable estimate of the HR, it is preferable to provide a point estimate for the HR, i.e. a HR estimated for a fixed first-line PFS or TTF. As we want to minimise the impact of platinum sensitivity on the HR we chose to report the HR computed at 6 months after first line treatment estimated using a restricted cubic splines approach⁷. This point represents the time at which a patient would be considered platinum-resistant and thus represents the point at which the effect of platinum is minimised.

Here, for each of the ovarian cohorts where we predicted taxane or anthracycline resistance, we report how we accounted for the effect of first-line platinum-based treatment.

Predicting resistance to taxane treatment in relapsed ovarian - OV04

For patients in the OV04 cohort treated with taxane-based therapy, the majority received their therapy at third-line or later (**Supplementary Note Figure 2a**), with 90% treated with taxane as a single agent (**Extended Data Figure 4**). Given the bias towards later lines and minimal effect from co-treatment with platinum, we did not use an interaction term in the Cox model. Rather, we elected to control for the treatment line under the assumption that lines closer to first-line might experience stronger effects compared to later lines. For completeness, we tested the model with an interaction, however, this did not have a significant impact.

Predicting resistance to anthracycline treatment in relapsed ovarian - OV04

For patients in the OV04 cohort treated with anthracycline-based therapy, most received anthracyclines at second-line (**Supplementary Note Figure 2b**) in combination with platinum (**Extended Data Figure 4**). The effect of combination treatment and proximity to first-line treatment means that PFS intervals in this context are likely to be heavily influenced by platinum. Thus in this case we chose to include an interaction between first-line PFS and the main predictor variable in our Cox model.

Predicting resistance to taxane treatment in relapsed ovarian - TCGA

Given the increased cohort size for TCGA, we could be more strict in our eligibility criteria to minimise the effect of platinum treatment. Therefore, we did not consider any patients that received taxane in combination with platinum. The majority of the remaining patients received taxane treatment second-line (**Supplementary Note Figure 2c**). This proximity to first-line treatment means that TTF intervals in this context are likely to be influenced by platinum. Thus in this case we chose to include an interaction between first-line TTF and the treatment arm variable in our Cox model.

Predicting resistance to anthracycline treatment in relapsed ovarian - TCGA

Similar to above, we did not consider patients who received anthracyclines in combination with platinum. Most of the remaining patients received anthracycline treatment distributed across lines 2, 3 or 4, thus in this instance, we did not include an interaction term in the Cox model but did control for the treatment line.

Supplementary Note Figure 2. Distribution of taxane-based and anthracycline-based treatment administration in ovarian cancer per treatment line. a) Taxane administration in the OV04 cohort. **b)** Anthracycline administration in the OV04 cohort. **c)** Administration of taxane in relapsed ovarian cancer from TCGA. **d)** Administration of anthracycline in relapsed ovarian cancer from TCGA.

What is the objective of the trial? What are the criteria for different treatments? What is the end point?

We thank the reviewer for encouraging us to provide more information on our trial emulation details and realise this was inadequate in the previous version of the manuscript. For all trial emulations we now include a clear objective, inclusion/exclusion criteria and an endpoint. These can be found in **Figures 2 and 3**, the accompanying text, **Methods lines 68-310**. In addition, **Supplementary Document 1** contains flowcharts detailing inclusion and exclusion

criteria specific for each patient cohort. For convenience we have reproduced the new Figure 3 below where many of these details are now presented.

New Figure 3

...
 • ...
 • ...
 • ...

...
 • ...
 • ...

...
 • ...
 • ...

...
 • ...
 • ...

...
 • ...
 • ...

...
 • ...
 • ...

UUM

Ovarian - Taxane

c) Relapsed Ovarian - Anthracycline

d) Sarcoma - Anthracycline

e) Metastatic Prostate - Taxane

f) Metastatic Breast - Taxane

g) Metastatic Breast - Anthracycline

Figure 3. Performance assessment of platinum, taxane and anthracycline resistance prediction across real-world cohorts. **a)** Cox proportional hazards regression models showing time to treatment failure in primary ovarian patients predicted as resistant to first-line platinum-based treatment stratified by age at diagnosis (<60, 60-69, and ≥70 years old), and controlling for wGII and tumour stage. **b)** Cox proportional hazards regression models showing time to treatment failure in relapsed ovarian cancer patients predicted as resistant to taxane stratified by age at diagnosis (<65 and ≥65 years) and tumour stage (III and IV), and including an interaction term between the treatment arm and first-line TTF. The reported hazard ratio is a point estimate at 6 months after first-line treatment (further details in **Supplementary Note 2**). **c)** Cox proportional hazards regression models showing time to treatment failure in relapsed ovarian cancer patients predicted as resistant to anthracycline stratified by age at diagnosis (<65 and ≥65 years) controlling for treatment line. **d)** Cox proportional hazards regression models showing time to treatment failure in primary sarcoma patients predicted as resistant to anthracycline controlled for ifosfamide as a co-therapy. The limited sample size precluded us from correcting the model by other clinical covariates. **e)** Cox proportional hazards regression models showing time to treatment failure in metastatic prostate patients predicted as resistant to taxane controlled for age at diagnosis. Gleason grade was not available for correction. **f-g)** Cox proportional hazards regression models showing time to treatment failure in metastatic breast patients predicted as resistant to d) taxane and e) anthracycline. Regression models were controlled for age at diagnosis and wGII. All analyses were evaluated at a significance level of 0.05. Asterisk denotes significant results. wGII, weighted genome instability index.

It seems to be 41 patients but highly overlapping. Again, small samples sizes and overlapping data.

We thank the reviewer for encouraging us to make the overlap in our ovarian cancer pilot study clear. In the revised manuscript, we have now extended this cohort to 503 patients. **Extended Data Figure 3 (reproduced below for convenience)** shows the degree of overlap of patients between the three trial emulations for OV04. Despite 19 of the patients appearing across all three phase II emulations, we believe this does not influence the conclusions drawn given the construction of the Cox models used. All patients for the platinum analysis received platinum first-line and thus were not affected by subsequent taxane or anthracycline treatment. While taxane, or anthracycline treatment may be influenced by first-line PFS, we have controlled for this effect (see reproduction of **Supplementary Note 2** above). Finally, any generally prognostic signals from the overlapping patients have been accounted for via the inclusion of additional covariates in the Cox model: tumour stage, age at diagnosis, maintenance treatment and weighted genome instability index.

The overlapping nature of these patients also reflects a real-world testing scenario as we are attempting to predict resistance to multiple lines of chemotherapy from a single test at diagnosis. These patients represent the standard clinical pathway, receiving different treatments at different lines, and thus our assessments of each therapy response prediction will reuse the same patients.

The cohorts used in our pilot study correspond to the typical magnitude of sample numbers seen in preclinical and clinical development of a companion diagnostic biomarker with an associated therapy. For example, previous Phase II trials of new medicines in ovarian cancer have used 41, 40 and 18 patients (<https://clinicaltrials.gov/study/NCT05751629>,

<https://clinicaltrials.gov/study/NCT00130520>, <https://clinicaltrials.gov/study/NCT03394885>). Had these been precision therapies that required a companion diagnostic test, biomarker efficacy would have been assessed across the same magnitude of samples as we present in our trial emulations using the OV04 cohort. The large scale TCGA and Hartwig cohorts were used to assess performance in a pseudo-randomised setting. The samples in these analyses are equivalent to those typically seen in Phase III trials.

Reproduced Extended Data Figure 3

Extended Data Figure 3. Pilot study (OV04) patients included in the single-arm study testing biomarker performance for platinum, taxane and anthracycline. UpSet plot showing the number of patients included in the three different single-arm trial designs emulated as part of our pilot study. The bar plot in the bottom left represents the total set size used for testing each of the three chemotherapies. The main bar plot represents the size of each set interaction, as represented in the intersection matrix below.

Fig 3. Sufficient patients? Did they do power analysis?

This is an excellent suggestion by the reviewer. In the revised manuscript we now use power analysis to determine which cohorts have sufficient patient numbers to be included in our trial emulations. We sought to power our trial emulations to match the hazard ratio observed in our pilot study (see **Figure 2** for pilot study hazard ratios). We first sought cohorts powered to perform a Phase III randomised control biomarker study (see **Extended Data Figure 5b** for design). If a cohort was not powered, we sought power in just the resistant arm to emulate an enrichment design (see **Extended Data Figure 5c** for design). If the cohort was still not powered, we sought power for a phase II single-arm biomarker trial design (see **Extended Data Figure 5a** for design).

Powered cohorts appear in **Extended Data Tables 1-2 (reproduced below for convenience)** and the full list of all cohorts tested appears in **Supplementary Table 5**.

Reproduced Extended Data Tables 1-2:

Dataset	Cohort	Chemo	Required Resistant Sample Size	Powered Resistant HR	Actual Resistant Sample Size	Powered	Required Sensitive Sample Size	Powered Sensitive HR	Actual Sensitive Sample Size	Powered
HMF	Breast	Taxane	37	4.390	34	TRUE*	28	0.383	51	TRUE
HMF	Breast	Anthracycline	25	2.278	67	TRUE	27	0.242	18	FALSE
HMF	Prostate	Taxane	34	2.858	58	TRUE	24	0.509	93	TRUE
TCGA	OV	Taxane	14	2.030	23	TRUE	12	0.546	36	TRUE
TCGA	OV	Anthracycline	11	1.816	38	TRUE	14	0.504	47	TRUE

Extended Data Table 1. Power calculation results for Phase III survival analysis cohorts. Only cohorts with statistical power to emulate phase III biomarker trials are represented here. Cohorts with available data for performing powered analysis are listed in Supplementary Table 5.

Dataset	Cohort	Chemotherapy	Required Sample Size	Actual Sample Size	Powered HR	Powered	Used in Phase III
HMF	Breast	Taxane	29	52	3.5222	TRUE	TRUE
HMF	Breast	Anthracycline	20	60	2.6430	TRUE	TRUE
TCGA	OV	Platinum	32	346	1.3905	TRUE	FALSE
TCGA	OV	Taxane	11	108	1.7174	TRUE	TRUE
TCGA	OV	Anthracycline	11	81	1.8468	TRUE	TRUE
TCGA	SARC	Anthracycline	15	27	3.4495	TRUE	FALSE

Extended Data Table 2. Power calculation results for Phase II survival analysis cohorts. Only cohorts with statistical power to emulate phase II biomarker trials are represented here. Only powered cohorts that were not tested via phase III were used for downstream analyses. Cohorts with available data for performing powered analysis are listed in Supplementary Table 5.

Multiple hypotheses corrections?

Given the revised structure of the performance assessment and independent nature of each of the trial emulations, we do not believe that multiple hypothesis correction is required.

Results seem barely significant.

The addition of more samples in our analyses means that all results presented in the manuscript are considered significant at the standard threshold of $\alpha < 0.05$. This is now clearly stated in the manuscript.

Fig 4. While the results are interesting, again the sample sizes are far too small. Having only 9/29 patients have enough ctDNA in plasma for liquid biopsy seems like it's too low to be actually put into practice. They claim "These results suggest that for approximately 17-31% of patients, our cytotoxic chemotherapy response predictor may be applied using a simple blood test", but would need to see some data in a larger sample size to believe this.

Our aim was to present proof-of-concept data on the potential to utilise liquid biopsies to obtain our predictions and have toned down the text to account for this. In future work, we hope to increase the number of liquid biopsy samples for analysis. While the range of samples with sufficient ctDNA is low (30%), this does not preclude that this could be clinically useful, especially when considering novel methods under development that can enrich the tumour DNA fraction³. The tradeoff between taking a biopsy or blood draw depends on many factors including the risk of biopsy, the cost of applying our test to an upfront blood draw, the fraction of patients with adequate biopsy material, the stage at which the test is administered, to name a few. In absence of these data it is not possible to decide whether this rate is clinically actionable. Gathering these data is part of a planned future project but is unfortunately outside the scope of this paper. To extend the feasibility analysis of clinical implementation, we have incorporated gene panel sequencing data (see new **Figure 4, reproduced here for convenience**) with an initial cohort of patients (n=8) serving as a proof of concept.

Reproduced Figure 4

a) Sample-paired sWGS and TSO500 sequencing data

b)

		sWGS	
		Resistant	Sensitive
TSO500	Resistant	7	0
	Sensitive	1	6

92.9% of concordance

c) Patient-paired tissue and plasma biopsies

d)

		Tumour	
		Resistant	Sensitive
Plasma	Resistant	9	1
	Sensitive	2	6

83.3% of concordance

Figure 4. Comparison of response prediction between paired samples. **a)** Barplot showing signature activities in tumour biopsies sequenced by using both shallow WGS (sW) and TSO500 (T5). Only tissue biopsies from patients with high-quality sWGS-derived copy number profiles that met the inclusion criteria were also sequenced with TSO500. A cosine similarity close to 1 indicates similar activities between sample-matched data. **b)** Contingency table showing the number of patients predicted as sensitive or resistant using our signature-based clinical classifiers in tumour samples sequenced by sWGS and TSO500. **c)** Barplot showing signature activities in matched tumour tissue (T) and plasma (P) samples from the same patient. Only high-quality plasma samples were used for deriving copy number profiles and signature quantification. A cosine similarity close to 1 indicates similar activities between patient-matched biopsies. **d)** Contingency table showing the number of patients predicted as sensitive or resistant using our signature-based clinical classifiers in tumour tissue and plasma biopsies.

Reviewer 2

The paper by Thompson et al is fundamentally a biomarker study. They purport to have devised an integrated biomarker that can predict response to platinum chemotherapy, taxanes and anthracyclines. The paper initially focuses on ovarian cancer but then seeks to generalize the findings using TCGA data.

The research objective is clearly a worthy area of study and if successful, this work could have implications for patient care. Although most novel therapies are now targeted or immune based therapies, cytotoxic chemotherapies remain a mainstay of the systemic treatment of many cancer types and the ability to predict which agents would be most effective using pretreatment tumor or blood samples would be an important advance. The work could also have utility for predicting response to antibody drug conjugates. The specific approach used here is the identification of CIN signatures inferred from whole genome sequencing of tumor and cfDNA.

My major concern with the paper is that one cannot infer the clinical utility of their biomarker given the data presented. The authors use the the term “response” but do not in fact measure response based on widely accepted criteria such as RECIST.

These are very insightful points from the reviewer which have prompted us to substantially change the way we present our results. In the original manuscript, we used the term “response” loosely in the title and the results section without respecting its clinical definition. We realise this mistake was misleading and made it challenging to assess the clinical utility of our predictors. We apologise for this. Our biomarkers are best used to predict **resistance** to platinum, taxanes and anthracyclines and our results best support this application. On reflection a more accurate title for the manuscript is “Predicting resistance to cytotoxic chemotherapy” and we have therefore updated both the title and text to reflect this.

The data are also presented as a Cox proportional hazard models which precludes a patient level view of how well the predictor distinguishes “responders” versus “non-responders”. This is critical as similar such biomarkers have not been adopted clinically because they provide insufficient distinction between responding and non-responding populations as to be of clinical utility for medical decision making. In sum, at minimum, the authors need a dataset to validate their approach in which RECIST response data is available and they should present the data using waterfall plots with annotation of biomarker positive and negative patients clearly distinguishable.

From our understanding of the RECIST criteria, it appears that a biomarker RCT trial that assesses the ability to predict *resistance* does not strictly require RECIST criteria. The RECIST 1.1 update criteria state that “Confirmation of response is required for trials with response primary endpoint but is no longer required in randomised studies since the control arm serves as appropriate means of interpretation of data”⁴. While it would obviously be better to include RECIST in our analyses, if we did, this would require that each patient have measurable disease at diagnosis thus limiting the patients which could assess in our trial emulations. This is why we opted to use TTF as an endpoint in our analysis. This logic, however, was not clearly articulated in our manuscript and we have substantially reworked the text to reflect this.

The RCT emulations, which have been substantially refined and extended into the metastatic setting show the utility of our biomarkers is in identifying patients resistant to chemotherapy treatments with respect to alternate standards of care (see the new **Figure 3, also reproduced above on p13**). The suggestion of waterfall plots demonstrating a patient level

view of this is excellent and we have added these to the manuscript (**Extended Data Figure 6, reproduced below for convenience**), albeit this is limited to reporting changes in TTF from an average baseline TTF, given the data.

Extended Data Figure 6. Waterfall plots showing the baseline change from the average time to treatment failure in the control arm (TTF) for the patients in the experimental arms of the Phase III analyses. a) Relapsed ovarian taxane. b) Relapsed ovarian anthracycline. c) Metastatic prostate taxane. d) Metastatic breast taxane. e) Metastatic breast anthracycline.

Reviewer 3

Remarks to the Author:

In their study, Thompson, Madrid et al. describe using signatures of chromosomal instability in tumor samples to predict the response to different chemotherapeutic agents in ovarian cancer. While the performance in ovarian cancer was validated for taxanes and doxorubicin in TCGA data, this was only extended to breast cancer. The assay is extended to a limited number of liquid biopsies from the same ovarian cancer donors, highlighting the potential this assay could be used clinically.

Overall, I think this study currently overpromises and underdelivers. The biggest caveat of the described assay and classification is that, in essence, it only evaluates two biological phenotypes that may confer resistance to chemotherapeutic agents: micronuclei-tolerance as a resistance mechanism to doxorubicin and homologous recombination impairment as a resistance mechanism to platins and taxanes. There are many other pathways to therapy resistance, to which a CIN signature-based approach would be blind.

The reviewer is correct that there are likely other factors causing therapy resistance beyond micronuclei-tolerance or (a lack of) different types of impaired homologous recombination. However, in this study, we aim to assess how much CIN signatures alone can encode a predictive signal of treatment resistance and whether this may warrant further consideration for use in the clinic. While it may be possible to measure the other factors mediating resistance, it will likely make clinical translation more complex, as additional assays will be needed. We reasoned that if we can make meaningful predictions using CIN signatures alone, it might create a clinical translation pathway which will only require a single genomic test. Reflecting on the reviewers comments, we have tried to substantially revise the text to highlight our focus on using CIN signatures alone and the benefits this might bring. We believe this has enhanced the presentation of our work and thank the reviewer for the critique.

Combined with the inability to replicate the signature-based prediction broadly across cancer types, I'm inclined to believe the applicability of this assay is much more limited than promised.

We thank the reviewer for challenging us to test our approach more broadly. In the revised manuscript we have included new patient data enabling us to assess performance in 740 patients. We now show our biomarkers have utility in ovarian cancer, breast cancer, sarcomas and prostate cancer, across both primary and metastatic treatment settings. See new **Figure 3** for an overview, **reproduced on page p13 above for convenience**).

Specific comments are detailed below.

Comments

- The prediction of chemotherapy response is very limited to a few cancer types. From the TCGA cohort, only ovarian and breast cancers had a sufficient number to be considered for a randomized controlled study analysis, and validation failed in cervical, head and neck, and uterine cancer cohorts. The classifier works well for ovarian cancers and breast cancers but seems by no means well-tested or successful in a “pan-cancer” setting.

We thank the reviewer for encouraging us to apply our biomarkers to more tumour types. In the revised manuscript we include new analyses specifically designed to optimise our biomarkers for implementation across multiple tumour types (**see section title “CIN signatures as biomarkers of chemotherapy resistance” main manuscript line 103**). Furthermore, we have extended our performance assessment to patients from the Hartwig Medical Foundation metastatic cohort. This has allowed us to demonstrate the utility of our approach in ovarian, breast, sarcoma and prostate cancer across a series of RCT emulations. The addition of upfront power calculations for these phase III RCT emulations has allowed us to determine which cohorts are powered to carry out a phase II or III trial emulation (**see response to reviewer point on page p15 for full explanation of power calculations**). Interestingly, for all cohorts for which we are powered, we demonstrate predictive potential for our biomarkers. The previously reported cohorts where the methodology did not seem to work were not in fact powered.

Despite the revised manuscript showing predictive potential across two more tumour types, we agree that additional testing will be required before we can truly claim this approach is pan-cancer. Thus we have attempted to tone down the text regarding claims around broad pan-cancer application.

- Based on Extended Data Figures 3 and 4, only TCGA patients with >20 CN events were included in the predication model. While it makes sense to only classify cases where there is something to classify, it hides the requirement that a tumor needs to exhibit many CNVs for this approach to work. How many tumors with fewer than 20 CNs were excluded and of these, how many were resistant to chemotherapy?

This is an insightful point from the reviewer and something we had actually overlooked in our previous analysis. Indeed, providing predictions for tumours with no detectable CIN is extremely important for clinical applicability of our test. In all cases, no CIN samples have a clear predicted outcome based on known biology. Tumours without CIN will not have impaired homologous recombination and are therefore likely to be resistant to taxanes platinum-based treatments. Whereas, tumours which have not yet acquired tolerance to micronuclei are likely to be sensitive to anthracyclines. We have therefore updated our classification rules such that our test can also provide predictions for no CIN samples (**see new Figure 1, reproduced here for convenience**). We thank the reviewer for this suggestion.

Reproduced Figure 1

Biomarker principle

Biomarker optimization

Taxanes

Anthracyclines

f) Organoids & spheroids derived from ovarian tumours

		Biomarker prediction	
		Resistant	Sensitive
In vitro response	Resistant	14	3
	Sensitive	0	6

87.0% concordance

Figure 1. Optimization of biomarkers to predict chemotherapy resistance pan-cancer. Workflow for using CIN signatures as biomarkers for predicting resistance to **a)** platinum-based chemotherapies, **c)** taxanes and **e)** anthracyclines. **b)** Cox proportional hazards regression models showing overall survival in TCGA-ESCA patients classified as predicted or sensitive to platinum-based chemotherapy after applying the classifier from a). The blue dot indicates results obtained after optimising the biomarkers for pan-cancer application, while the orange dot indicates results obtained using our original approach⁵. Cox proportional hazards models correcting for stage and age at diagnosis. ESCA, oesophageal adenocarcinoma. **d)** Dot plot showing the mean area under the dose response curve of cell lines predicted as resistant (y-axis) using a range of signature activities for thresholding (x-axis). A total of 285 cell lines having high-quality paclitaxel response data were included in the analysis. Red dot denotes the activity value selected as the optimal threshold. Dashed lines show the lower and upper CX5 activity thresholds that match the expected rate of cells as resistant (30 to 60%)⁶. **f)** Contingency table showing the agreement between the observed and the predicted response of patient-derived models to doxorubicin *in vitro*. Samples with at least one of the three amplification-related signatures (CX8/CX9/CX13) showing an activity higher than 0.01 were predicted as resistant.

- The study goes back and forth between using the signatures from Macintyre et al. (2018) and Drews et al. (2022), using ovarian CIN signature 6 to predict chemotherapy response in ovarian cancer and pan-cancer signatures (CXs) to predict response in breast cancer. Rather than using two different signature reference sets, I suggest using signatures from one reference set to improve the clarity and coherence of the study, especially since the authors already use the equivalence between ovarian CIN signature 6 and CX8, CX9 and CX13.

We thank the reviewer for the suggestion and have now adopted only the pan-cancer CIN signatures from Drews et al. 2022 across all analyses.

- Whenever a copy number signature is introduced, I suggest adding a few words introducing the features and presumed etiology of the signature (as already done for CX8/CX9/CX13). This will help guide readers less familiar with these sets of signatures. A big omission in the text is the link between CX3 and CX5 and homologous recombination impairment, which is only mentioned in the abstract.

We thank the reviewer for the suggestion and have now added descriptions in the text.

- Essentially, estimating the activity of CX2, CX3 and CX5 seems to a proxy for detecting homologous recombination deficiency, which is commonly used to predict sensitivity to PARP-inhibitors. Do the authors think such methods would work well to predict resistance to the chemotherapeutics included in thus study? What is the advantage of the approach described here?

This is another excellent suggestion by the reviewer. We have added an assessment of how two common predictors of HRD, HRDetect and Myriad myChoice score perform with respect to taxane and platinum response prediction (see **Supplementary Note 3, reproduced here for convenience**). In this instance, information for the predictors was only available to enable a phase II emulation. The only approach which potentially had predictive signal was HRDetect in predicting resistance to platinum in ovarian cancer (HR=3.87, p=0.01, 35/75 samples with available HRD scores; independent of tumour stage and age at diagnosis). The other results did not show predictive capacity, although some of these analyses were likely underpowered. The advantage of our IHR biomarkers is we are able to further break down HRD into 3 separate biomarkers of IHR which show differential capabilities to predict resistance to platinum and taxanes.

Reproduced Supplementary Note 3

Supplementary Note 3 - assessing performance of HRDetect and Myriad myChoice

Here we sought to evaluate the performance of two state-of-the-art HRD predictors, HRDetect and Myriad MyChoice in predicting resistance to taxane and platinum-based treatment. These HRD predictors incorporate additional data beyond copy number⁷⁻⁹.

We extracted predictions from application of these two methods to ovarian and breast cancer TCGA cohorts: HRDetect annotations were taken from previous publications^{7,8}, and a threshold of 0.7 was used to identify positive samples⁷; Myriad myChoice annotations were based on HRD scores⁹ which were taken from previous publications^{10,11}, and scores above or equal to 42 classified as positive.

We conducted a survival analysis comparing time to treatment failure (TTF) between patients predicted as HRD-positive or negative using these methods (thus sensitive or resistant to taxanes and platinum) (**Supplementary Note Figure 2**). Due to the limited number of samples with HRDetect and/or Myriad MyChoice information, we only were able to assess performance by a phase II single-arm trial emulation.

Out of the six emulations, only one showed a significant difference in TTF between predicted sensitive and resistant patients. HRDetect negative patients showed a significantly increased risk of treatment failure compared to sensitive platinum-based treatment in ovarian cancer (HR=3.87, p=0.01, 35/75 samples with available HRD scores; independent of tumour stage and age at diagnosis). The other results did not show predictive capacity (HRDetect in TCGA-OV taxane (n=6/42): HR=0.623 p=0.61; Myriad in TCGA-OV taxane (n=6/42): NA all are predicted positive; Myriad in TCGA-OV platin (n=35/75): HR=0.757 p=0.583; HRDetect in TCGA-BRCA taxane (n=8/75): HR=1.843, p=0.504; Myriad in TCGA-BRCA taxane (n=8/75): HR=1.843, p=0.504;). However, it is noted, in some instances these analyses were likely underpowered.

Supplementary Note Figure 3. Performance assessment of state-of-the-art HRD classifiers for predicting resistance to platinum and taxanes. Kaplan Meier curves for patients predicted to be HRD-positive (yellow) and HRD-negative (blue). HRD-positive patients are classified as sensitive, while HRD-negative patients are classified as resistant. TCGA-OV platinum: P-values from cox models corrected by tumour stage and age at diagnosis. TCGA-OV and TCGA-BRCA taxanes: P-values from univariate cox models.

Typos

- In Fig 1d, the corresponding dot in the right plot is missing for the top sample (119120).

The presentation of these data has changed and thus this comment is no longer applicable.

- The text mentions classifying 41 patients into platinum-sensitive and platinum-resistant, but Fig 2a shows 39 patients.

We thank the reviewer for pointing out this mistake. In the updated manuscript, we have confirmed that all numbers are adequately represented in the REMARK diagrams for our extended OV04 cohort (see **Extended Data Figure 2, reproduced here for convenience**).

Extended Data Figure 2. REMARK diagram. a) Flow diagram summarising the quality control filtering of samples and patients for the pilot study assessing the performance of platinum resistance prediction. **b)** Flow diagram detailing samples and patients filtered to get the curated cohort for the pilot study of taxane resistance prediction. **c)** Flow diagram detailing organoids and spheroids filtered for anthracycline biomarker discovery, as well as samples and patients filtered for the pilot study of anthracycline resistance prediction.

References for referee response

1. Blackledge, G., Lawton, F., Redman, C. & Kelly, K. Response of patients in phase II studies of chemotherapy in ovarian cancer: implications for patient treatment and the design of phase II trials. *Br. J. Cancer* **59**, 650–653 (1989).
2. Bellavia, A., Melloni, G. E. M., Park, J.-G., Discacciati, A. & Murphy, S. A. Estimating and presenting hazard ratios and absolute risks from a Cox model with complex nonlinear interactions. *Am. J. Epidemiol.* **193**, 1155–1160 (2024).
3. Markus, H., Chandrananda, D., Moore, E., Mouliere, F., Morris, J., Brenton, J. D., Smith, C. G. & Rosenfeld, N. Refined characterization of circulating tumor DNA through biological feature integration. *Sci. Rep.* **12**, 1928 (2022).
4. Eisenhauer, E. A., Therasse, P., Bogaerts, J., Schwartz, L. H., Sargent, D., Ford, R., Dancey, J., Arbuck, S., Gwyther, S., Mooney, M., Rubinstein, L., Shankar, L., Dodd, L., Kaplan, R., Lacombe, D. & Verweij, J. New response evaluation criteria in solid tumours: revised RECIST guideline (version 1.1). *Eur. J. Cancer* **45**, 228–247 (2009).
5. Drews, R. M., Hernando, B., Tarabichi, M., Haase, K., Lesluyes, T., Smith, P. S., Morrill Gavarró, L., Couturier, D.-L., Liu, L., Schneider, M., Brenton, J. D., Van Loo, P., Macintyre, G. & Markowitz, F. A pan-cancer compendium of chromosomal instability. *Nature* **606**, 976–983 (2022).
6. Baert, T., Ferrero, A., Sehouli, J., O'Donnell, D. M., González-Martín, A., Joly, F., van der Velden, J., Blecharz, P., Tan, D. S. P., Querleu, D., Colombo, N., du Bois, A. & Ledermann, J. A. The systemic treatment of recurrent ovarian cancer revisited. *Ann. Oncol.* **32**, 710–725 (2021).
7. Davies, H., Glodzik, D., Morganella, S., Yates, L. R., Staaf, J., Zou, X., Ramakrishna, M., Martin, S., Boyault, S., Sieuwerts, A. M., Simpson, P. T., King, T. A., Raine, K., Eyfjord, J. E., Kong, G., Borg, Å., Birney, E., Stunnenberg, H. G., van de Vijver, M. J., Børresen-Dale, A.-L., Martens, J. W. M., Span, P. N., Lakhani, S. R., Vincent-Salomon, A., Sotiriou, C., Tutt, A., Thompson, A. M., Van Laere, S., Richardson, A. L., Viari, A., Campbell, P. J., Stratton, M. R. & Nik-Zainal, S. HRDetect is a predictor of BRCA1 and BRCA2 deficiency based on mutational signatures. *Nat. Med.* **23**, 517–525 (2017).
8. Degasperi, A., Amarante, T. D., Czarnecki, J., Shooter, S., Zou, X., Glodzik, D., Morganella, S., Nanda, A. S., Badja, C., Koh, G., Momen, S. E., Georgakopoulos-Soares, I., Dias, J. M. L., Young, J., Memari, Y., Davies, H. & Nik-Zainal, S. A practical framework and online tool for mutational signature analyses show inter-tissue variation and driver dependencies. *Nat Cancer* **1**, 249–263 (2020).
9. Telli, M. L., Timms, K. M., Reid, J., Hennessy, B., Mills, G. B., Jensen, K. C., Szallasi, Z., Barry, W. T., Winer, E. P., Tung, N. M., Isakoff, S. J., Ryan, P. D., Greene-Colozzi, A., Gutin, A., Sangale, Z., Iliev, D., Neff, C., Abkevich, V., Jones, J. T., Lanchbury, J. S., Hartman, A.-R., Garber, J. E., Ford, J. M., Silver, D. P. & Richardson, A. L. Homologous Recombination Deficiency (HRD) Score Predicts Response to Platinum-Containing Neoadjuvant Chemotherapy in Patients with Triple-Negative Breast Cancer. *Clin. Cancer Res.* **22**, 3764–3773 (2016).

10. Marquard, A. M., Eklund, A. C., Joshi, T., Krzystanek, M., Favero, F., Wang, Z. C., Richardson, A. L., Silver, D. P., Szallasi, Z. & Birkbak, N. J. Pan-cancer analysis of genomic scar signatures associated with homologous recombination deficiency suggests novel indications for existing cancer drugs. *Biomark. Res.* **3**, 9 (2015).
11. Knijnenburg, T. A., Wang, L., Zimmermann, M. T., Chambwe, N., Gao, G. F., Cherniack, A. D., Fan, H., Shen, H., Way, G. P., Greene, C. S., Liu, Y., Akbani, R., Feng, B., Donehower, L. A., Miller, C., Shen, Y., Karimi, M., Chen, H., Kim, P., Jia, P., Shinbrot, E., Zhang, S., Liu, J., Hu, H., Bailey, M. H., Yau, C., Wolf, D., Zhao, Z., Weinstein, J. N., Li, L., Ding, L., Mills, G. B., Laird, P. W., Wheeler, D. A., Shmulevich, I., Cancer Genome Atlas Research Network, Monnat, R. J., Jr, Xiao, Y. & Wang, C. Genomic and molecular landscape of DNA damage repair deficiency across the cancer genome atlas. *Cell Rep.* **23**, 239–254.e6 (2018).

Predicting resistance to cytotoxic chemotherapy

Point-to-point response to referees for manuscript NG-A63066R1

We thank the reviewers for their constructive comments which have improved our work.

Main changes in this revision:

- Incorporation of inverse probability weighting into our RCT emulations to account for potential treatment selection bias
- Addition of swimmer plots and Kaplan Meier plots

This document provides a detailed point-to-point response to all reviewers' comments. We have listed the comments in full, followed by our responses in blue boxes. Text changes are highlighted in **blue** in the main manuscript, methods and supplementary files.

Reviewers' Comments

Reviewer #1 (Remarks to the Author)

The authors have addressed most concerns.

We thank the reviewer for the positive comment and previous suggestions which improved our manuscript.

Reviewer #2 (Remarks to the Author)

The conclusion of the revised manuscript by Thompson et al is that CIN signatures can be used to predict resistance to various chemotherapies across multiple different types of cancer. The study was a retrospective analysis that sought to identify the potential clinical utility of CIN signatures as a predictive biomarker through pseudo-randomization to a single chemotherapy treatment arm or alternative standard-of-care arm.

Overall, the authors were responsive to many of the prior concerns raised by the reviewers during the initial review and the paper is now improved. My primary concern remains unresolved however. Specifically, the authors have not been able to provide evidence that the biomarker has clinical utility (and thus will be of interest to the clinical oncology community) through a valid patient level analysis. They state that RECIST response data was unavailable for the patients analyzed and therefore they generated waterfall plots where the y axis is baseline change compared to the average control arm (I do not see how a reader can know what this really means). In my opinion, if they are going to use a time to event metric to study time to drug resistance, they need to show this patient level data as a swimmers plot, Kaplan-Meier curves or both. If a swimmers plot is used, it should include both time on therapy and time to next therapy on the plot as chemotherapy is often held not for progression but for toxicity. Based on the data shown, such plots would likely show that many of the patients with “resistant signatures” had some of the longer times on therapy. If so, I am doubtful that such a signature would be used by clinicians for treatment decisions lacking prospective randomized data. I therefore would suggest a repeat review following inclusions of such plots and not simply the hazard ratios provided in the main figures.

We thank the reviewer for their suggestion. We have now included swimmer plots for the OV04 cohort (**Supplementary Figures 54-56**, reproduced below for convenience). These plots do not reveal any bias in time on therapy between patients predicted to be resistant or sensitive and no statistical difference in times on therapy were observed across each of the treatments (platinum: p-value=0.4388; taxane: p-value=0.5736; doxorubicin: p-value=0.2078; Wilcoxon rank sum test performed after excluding censored patients). Therefore, reassuringly the reviewer’s concerns regarding toxicity mediated progression seem not to be present in the data.

Supplementary Figure 54. Swimmer plot showing treatment length, monitoring and outcomes in OV04 patients included in the platinum cohort. Bars indicate progression-free survival. Asterisks indicate the start of the next treatment line, triangles indicate those patients that are censored, and red lines indicate time on therapy. No statistical difference in treatment length was observed between resistant and sensitive patients (Wilcoxon rank sum test p-value = 0.44)

Supplementary Figure 55. Swimmer plot showing treatment length, monitoring and outcomes in OV04 patients included in the paclitaxel cohort. Bars indicate progression-free survival. Asterisks indicate the start of the next treatment line, triangles indicate those patients that are censored, and red lines indicate time on therapy. No statistical difference in treatment length was observed between resistant and sensitive patients (Wilcoxon rank sum test p-value = 0.57).

Supplementary Figure 56. Swimmer plot showing treatment length, monitoring and outcomes in OV04 patients included in the doxorubicin cohort. Bars indicate progression-free survival. Asterisks indicate the start of the next treatment line, triangles indicate those patients that are censored, and red lines indicate time on therapy. No statistical difference in treatment length was observed between resistant and sensitive patients (Wilcoxon rank sum test p-value = 0.21).

Unfortunately time on therapy information was not available for the TCGA or Hartwig cohorts so we were not able to generate swimmer plots. We did, however, generate Kaplan Meier plots for the RCT emulations (**Extended Data Figures 6-7**, reproduced below for convenience). These univariate representations reveal similar conclusions to the multivariate Cox models and we hope this provides the patient level representation of the data the reviewer is seeking.

Extended Data Figure 6. Kaplan-Meier survival curves for the taxane biomarker. Kaplan-Meier curves showing time to treatment failure differences between patients treated with taxane (Experimental) and patients treated with other standard-of-care therapies (Control) in the resistant and the sensitive arms of the different randomized control trials emulated in this study. Univariate p-values were estimated by the log-rank test. **a)** Predicted resistant patients of the relapsed ovarian cancer cohort. **b)** Predicted sensitive patients of the relapsed ovarian cancer cohort. **c)** Predicted resistant patients of the metastatic prostate cancer cohort. **d)** Predicted sensitive patients of the

metastatic prostate cancer cohort. **e)** Predicted sensitive patients of the metastatic breast cancer cohort. **f)** Predicted sensitive patients of the metastatic breast cancer cohort.

Extended Data Figure 7. Kaplan-Meier survival curves for the anthracycline biomarker. Kaplan-Meier curves showing time to treatment failure differences between patients treated with anthracycline (Experimental) and patients treated with other standard-of-care therapies (Control) in the resistant and the sensitive arms of the different randomized control trials emulated in this study. Univariate p-values were estimated by the log-rank test. **a)** Predicted resistant patients of the relapsed ovarian cancer cohort. **b)** Predicted sensitive patients of the relapsed ovarian cancer cohort. **c)** Predicted resistant patients of the metastatic breast cancer cohort.

In light of the reviewers comments, we have now removed the previously generated waterfall plots as we are unable to generate standard waterfall plots given the available data.

The manuscript has been updated to reference these figures at the appropriate places.

Reviewer #3 (Remarks to the Author)

In their revised manuscript, Thompson, Madrid, Hernando et al. present their work on predicting resistance to cytotoxic chemotherapy based on signatures of chromosomal instability. I commend the authors on the scope of the revisions and think the inclusion of the Hartwig data, the more careful phrasing and scope of the manuscript, and the enhanced focus on clinical translation has substantially improved the study. The authors have adequately addressed my previous concerns.

Small typo on l.24, it should be “many patients can...”

I congratulate the authors on the revised study, and have no further comments.

Tim Coorens

We thank the reviewer for the positive comments and previous suggestions which have improved our manuscript.

Reviewer #3 (Remarks on code availability)

Code looks to be all present and is organized well, so it's easy to follow for the reader/reviewer.

Thank you for reviewing the code.

Reviewer #4 (Remarks to the Author)

This study introduces chromosomal instability (CIN) biomarkers as a potential tool to predict resistance to key chemotherapy regimens, including platinum-, taxane-, and anthracycline-based treatments. Using retrospective analyses of real-world cohorts and emulated clinical trials, the authors demonstrate that CIN signatures effectively stratify patients likely to experience treatment failure across multiple cancer types. They also highlight the feasibility of integrating these biomarkers into clinical practice using genomic assays from tissue or liquid biopsies. The work offers a promising step toward personalized chemotherapy. However, I have some statistical concerns about the RCT emulations in the paper.

Detailed comments:

For the RCT emulations, could the authors provide more methodological details? The paper currently states: “Within these sensitive or resistant groups, patients were then retrospectively assigned to the experimental arm (treated with the chemotherapy of interest) or to the control arm (treated with an appropriate standard-of-care therapy).” However, in observational data, treatment selection bias is a significant concern. To emulate randomization effectively, it is essential to address this bias and recover the randomization process. Methods such as inverse probability weighting or propensity score matching are commonly employed to balance treatment arms and mimic randomization. Patient characteristics and clinical factors that may influence treatment assignment in real-world

data should be incorporated in the propensity scores to fully control for selection bias. Could the authors conduct such analyses to validate the findings presented?

This is an excellent suggestion by the reviewer. In the revised manuscript, we have incorporated inverse probability weighting into the Cox models to address potential treatment selection bias.

Given the evolving landscape of therapy approvals during the years in which the patients used in our study were treated (TCGA: 1992-2009; Hartwig: 2012-2020), we identified year of treatment as the most likely factor which might influence treatment selection between experimental and control arms. While other available variables such as age at diagnosis or tumour stage may have influenced treatment selection, we reasoned that they likely have a more significant direct effect on survival outcomes. Thus, we included these as covariates in the Cox proportional hazards models rather than in the inverse probability weighting to avoid double correction.

Reassuringly, following inverse probability weighting our results did not substantially change. All Cox model results have now been updated in the manuscript and reflect those with the addition of inverse probability weighting. A detailed description of the procedure has been added to **Methods page 23, reproduced below for convenience**.

Inverse probability weighting

Treatment selection may be influenced by patient characteristics and clinical factors. To effectively emulate the randomisation process in the phase III RCT and enrichment trial survival analyses, we applied inverse probability weighting to address potential systemic biases that might have influenced treatment selection.

First, we explored the available covariates in the TCGA and HMF datasets to identify those that may have a significant effect on treatment assignments. We identified the year of treatment/biopsy as the clinical covariate most likely to impact whether a patient received the chemotherapy of interest or an alternative standard of care (range of treatment years in TCGA: 1992-2009; range of biopsy years in Hartwig: 2012-2020). This was based on the fact that treatment decisions are guided by clinical guidelines, which evolve over time. While age at diagnosis and treatment line were also considered as potential factors influencing treatment selection, we reasoned that these variables might have a stronger direct impact on survival outcomes and were instead included as covariates in the Cox proportional hazards models. We also evaluated whether geography, as well as patient race, could contribute to treatment selection bias. In the TCGA cohorts, the majority of patients were from North America, with over 95% enrolled in the USA. Similarly, all patients in the HMF cohorts were recruited from the Netherlands. Based on these observations, we concluded that geography was unlikely to influence treatment selection significantly. Regarding patient race, we determined that the impact of this bias would be minimal, as the patient demographics only included 4-12% non-white individuals, split over multiple groups. Consequently, we decided to not include race as a variable in the probability weighting.

To implement the weighting in the survival analyses, we fitted a logistic regression model to examine the association between the treatment/biopsy year and the treatment arm. The

model predicted the probability of each patient belonging to the experimental arm, and thus estimated propensity scores. These scores were converted into inverse probability weights using ATT (average treatment effect for the treated) as the estimand. This process was performed using the *WeightIt* package in R. The resulting weights were then incorporated into the Cox proportional hazards models through the *weights* parameter in the *coxph* function.

At the top of Page 9, it is stated: “Statistical analysis of the cohort was performed using Cox proportional-hazards models stratifying patients by age at diagnosis (<65 or ≥65 years) and tumour stage (III, IV).” Typically, stratified analyses involve fitting separate Cox models for different age groups and tumor stages. However, I did not see multiple hazard ratios (HRs) reported for different age groups or tumor stages, and in Figure 2, the HR is also reported for the tumor stage. Could the authors clarify whether the Cox model was adjusted for age and tumor stage or if the baseline hazard function was allowed to differ by age and tumor stage? Similar phrasing appears in several parts of the paper. Please clarify and provide the rationale for the model selection.

We thank the reviewer for picking this up and agree that our descriptions were unclear in the previous version of the manuscript.

In the revised manuscript we have now carefully clarified when we are adjusting or controlling for a variable in the model (and thus reporting a Hazard Ratio), or when we are using a stratified Cox model to allow the baseline hazard to vary.

Depending on the cohort, the tumour type and distribution of age and stage, the potential effects of these variables on outcome differed. As such, the Cox models varied in their structure.

For most of the ovarian Cox models, we included an interaction with first-line PFS/TTF, however, as this is discussed in detail in **Supplementary Note 2** we did not make further changes to the text. However, for the other covariates we have revised the methods (**Methods p21-23**) description and main text to carefully articulate the model structure and rationale. A summary is provided below for convenience:

Pilot study in ovarian cancer

For all three Cox models covering platinins, taxanes and anthracyclines, we controlled for variables which may independently affect progression free survival. These included tumour stage, age at diagnosis, treatment line (for taxane), general aneuploidy (via the weighted genome instability index), and whether the patient received maintenance therapy. For all of these variables we had sufficient numbers in each category and none of the variables violated proportional hazards.

Single-arm trial emulation for platinum in ovarian

In this model, we controlled for tumour stage (III vs IV) as this may influence response to first-line treatment with platinum. While we attempted to also control for age at diagnosis, this showed evidence of non-proportional hazards. Therefore, we used a stratified Cox model, allowing the baseline hazards to vary across age groups.

RCT emulation for taxane treatment in ovarian

In these models, we were unable to control for tumour stage given the majority of cases

had stage III cancer and there were a low number of patients with stage IV cancer (45 vs 10 in resistant arm & 51 vs 6 in sensitive arm). While we attempted to also control for age at diagnosis, this showed evidence of non-proportional hazards. Therefore, we used a stratified Cox model, allowing the baseline hazards to vary across age groups.

RCT emulation for taxane treatment in ovarian

In these models, we were also unable to control for tumour stage given the majority of cases had stage III cancer and there were a low number of patients with stage IV cancer (54 vs 8 in resistant arm & 45 vs 8 in sensitive arm). While we attempted to also control for age at diagnosis, this showed evidence of non-proportional hazards. Therefore, we used a stratified Cox model, allowing the baseline hazards to vary across age groups.

Single-arm emulation for anthracycline in sarcoma

Given the limited sample number we were not able to control for age at diagnosis or tumour stage. However, given the potential for isophosphamide as a co-therapy to influence outcomes (PMID: 24618336), we controlled for this in the Cox model.

RCT emulation for taxane in prostate and breast, and for anthracycline in breast

In these models we were able to control for age at diagnosis. As these cohorts were metastatic (advanced disease), we did not control for tumour stage or subtype.

Reviewer #4 (Remarks on code availability):

The code and data were made available online, accompanied by a README file. While I successfully executed several R scripts, this required installing additional packages that were not listed in the code. Notably, packages such as ggpubr, DiagrammeR, and data.table were missing. I recommend that the authors conduct a thorough review of the code dependencies and explicitly include all required packages to ensure seamless execution.

Thank you for reviewing the code. We have updated the Supplementary Table 7 to include a list of all packages required to run the code.

Predicting resistance to chemotherapy using chromosomal instability signatures

Point-to-point response to referees for manuscript NG-A63066R3

We thank the reviewers for their constructive comments which have improved our work.

Reviewers' Comments

Reviewer #2 (Remarks to the Author)

Overall, I continue to find this manuscript to be interesting but remain concerned about the clinical utility of the CIN signature as a predictive biomarker of drug response. The authors were responsive to the prior review by including swimmers plots and KM curves where possible given the data available. However, the swimmers' plots included do not show any difference between the biomarker sensitive and resistant patients. Notably, in the Figure legend for Supplemental Figure 54 they state, "No statistical difference in treatment length was observed between resistant and sensitive patients". This would suggest to me that the data does not support the biomarker as clinical useful as the differences are unlikely to prompt clinicians to alter their treatment decisions. I agree with the plan by the editorial team to pursue further statistical review and would defer to such statistical reviewers in regards to analyzing the methodology employed and the statistical significance of the findings. In regards to their clinical significance, I suspect the impact will be low.

We thank the reviewer for their comments.

In our original wording, we mistakenly used "treatment length" instead of "time on treatment" to refer to the yellow lines in Supplemental Figure 2 (previously Supplemental Figure 54). We apologise for this. This has now been corrected. In this instance, we actually want patients to have consistent times on treatment so that this covariate does not impact our analysis of PFS. We indeed found no significant difference in the times on treatment between resistant and sensitive patients meaning this factor did not influence the survival analysis results.

We acknowledge that when looking at PFS differences between resistant and sensitive groups on the swimmer plots there is not a strong perceptible PFS difference. While this might be expected for the results of a prospective clinical trial, where covariates are tightly controlled, this is not the case when using real-world data. The additional covariates affecting PFS in the real-world setting (such as first-line platinum response) obscure strong visual differences. Unfortunately, these covariates cannot be readily visualised on a swimmer plot. The effect of these covariates places strong emphasis on using the

appropriate Cox modelling to account for them, which is what motivated us to present the results of multivariate Cox models in the main figures. We have revised the figure legends to clarify this limitation. We appreciate the opportunity to refine our presentation and ensure the findings are accurately conveyed.

Reviewer #4 (Remarks to the Author)

The authors have addressed all my comments. I have no further comments.

We thank the reviewer for the positive comment and previous suggestions, which substantially improved our manuscript.